# State Aggregation Learning from Markov Transition Data

**Yaqi Duan**
Princeton University
yaqid@princeton.edu

**Zheng Tracy Ke**
Harvard University
zke@fas.harvard.edu

**Mengdi Wang**
Princeton University
mengdiw@princeton.edu

## Abstract

State aggregation is a popular model reduction method rooted in optimal control. It reduces the complexity of engineering systems by mapping the system's states into a small number of meta-states. The choice of aggregation map often depends on the data analysts' knowledge and is largely ad hoc. In this paper, we propose a tractable algorithm that estimates the probabilistic aggregation map from the system's trajectory. We adopt a soft-aggregation model, where each meta-state has a signature raw state, called an *anchor state*. This model includes several common state aggregation models as special cases. Our proposed method is a simple two-step algorithm: The first step is spectral decomposition of empirical transition matrix, and the second step conducts a linear transformation of singular vectors to find their approximate convex hull. It outputs the aggregation distributions and disaggregation distributions for each meta-state in explicit forms, which are not obtainable by classical spectral methods. On the theoretical side, we prove sharp error bounds for estimating the aggregation and disaggregation distributions and for identifying anchor states. The analysis relies on a new entry-wise deviation bound for singular vectors of the empirical transition matrix of a Markov process, which is of independent interest and cannot be deduced from existing literature. The application of our method to Manhattan traffic data successfully generates a data-driven state aggregation map with nice interpretations.

## 1 Introduction

State aggregation is a long-existing approach for model reduction of complicated systems. It is widely used as a heuristic to reduce the complexity of control systems and reinforcement learning (RL). The earliest idea of state aggregation is to aggregate "similar" states into a small number of subsets through a partition map. However, the partition is often handpicked by practitioners based on domain-specific knowledge or exact knowledge about the dynamical system [31, 6]. Alternatively, the partition can be chosen via discretization of the state space in accordance with some priorly known similarity metric or feature functions [33]. Prior knowledge of the dynamical system is often required in order to handpick the aggregation without deteriorating its performance. There lacks a principled approach to find the best state aggregation structure from data.

In this paper, we propose a model-based approach to learning probabilistic aggregation structure. We adopt the *soft state aggregation* model, a flexible model for Markov systems. It allows one to represent each state using a membership distribution over latent variables (see Section 2 for details). Such models have been used for modeling large Markov decision processes, where the membership can be used as state features to significantly reduce its complexity [32, 36]. When the membership distributions are degenerate, it reduces to the more conventional *hard state aggregation* model, and so our method is also applicable to finding a hard partition of the state space.

The soft aggregation model is parameterized by $p$ aggregation distributions and $r$ disaggregation distributions, where $p$ is the total number of states in a Markov chain and $r$ is the number of (latent) meta-states. Each aggregation distribution contains the probabilities of transiting from one raw state to different meta-states, and each disaggregation distribution contains the probabilities of transiting from one meta-state to different raw states. Our goal is to use sample trajectories of a Markov process to estimate these aggregation/disaggregation distributions. The obtained aggregation/disaggregation distributions can be used to estimate the transition kernel, sample from the Markov process, and plug into downstream tasks in optimal control and reinforcement learning (see Section 5 for an example). In the special case when the system admits a hard aggregation, these distributions naturally produce a partition map of states.

Our method is a two-step algorithm. The first step is the same as the vanilla spectral method, where we extract the first $r$ left and right singular vectors of the empirical transition matrix. The second step is a novel linear transformation of singular vectors. The rationale of the second step is as follows: Although the left (right) singular vectors are not valid estimates of the aggregation (disaggregation) distributions, their linear span is a valid estimate of the linear span of aggregation (disaggregation) distributions. Consequently, the left (right) singular vectors differ from the targeted aggregation (disaggregation) distributions only by a linear transformation. We estimate this linear transformation by leveraging a geometrical structure associated with singular vectors. Our method requires no prior knowledge of meta-states and provides a data-driven approach to learning the aggregation map.

**Our contributions.**

1. We introduce a notion of "anchor state" for the identifiability of soft state aggregation. It creates an analog to the notions of "separable feature" in nonnegative matrix factorization [13] and "anchor word" in topic modeling [4]. The introduction of "anchor states" not only ensures model identifiability but also greatly improves the interpretability of meta-states (see Section 2 and Section 5). Interestingly, hard-aggregation indeed assumes *all* states are anchor states. Our framework instead assumes there exists *one* anchor state for each meta-state.

2. We propose an efficient method for estimating the aggregation/disaggregation distributions of a soft state aggregation model from Markov transition data. In contrast, classical spectral methods are not able to estimate these distributions directly.

3. We prove statistical error bounds for the total variation divergence between the estimated aggregation/disaggregation distributions and the ground truth. The estimation errors depend on the size of state space, number of meta-states, and mixing time of the process. We also prove a sample complexity bound for accurately recovering all anchor states. To our best knowledge, this is the first result of statistical guarantees for soft state aggregation learning.

4. At the core of our analysis is an entry-wise large deviation bound for singular vectors of the empirical transition matrix of a Markov process. This connects to the recent interests of entry-wise analysis of empirical eigenvectors [2, 23, 24, 38, 11, 14]. Such analysis is known to be challenging, and techniques that work for one type of random matrices often do not work for another. Unfortunately, our desired results cannot be deduced from any existing literature, and we have to derive everything from scratch (see Section 4). Our large-deviation bound provides a convenient technical tool for the analysis of spectral methods on Markov data, and is of independent interest.

5. We applied our method to a Manhattan taxi-trip dataset, with interesting discoveries. The estimated state aggregation model extracts meaningful traffic modes, and the output *anchor regions* capture popular landmarks of the Manhattan city, such as Times square and WTC-Exchange. A plug-in of the aggregation map in a reinforcement learning (RL) experiment, taxi-driving policy optimization, significantly improves the performance. These validate that our method is practically useful.

**Connection to literature.** While classical spectral methods have been used for aggregating states in Markov processes [34, 37], these methods do not directly estimate a state aggregation model. It was shown in [37] that spectral decomposition can reliably recover the principal subspace of a Markov transition kernel. Unfortunately, singular vectors themselves are *not* valid estimates of the aggregation/disaggregation distributions: the population quantity that singular vectors are trying to estimate is strictly different from the targeted aggregation/disaggregation distributions (see Section 3).

Our method is inspired by the connection between soft state aggregation, nonnegative matrix factorization (NMF) [26, 16], and topic modeling [7]. Our algorithm is connected to the spectral method in [22] for topic modeling. The method in [22] is a general approach about using spectral decomposition for nonnegative matrix factorization. Whether or not it can be adapted to state aggregation learning

and how accurately it estimates the soft-aggregation model was unknown. In particular, [22] worked on a topic model, where the data matrix has column-wise independence and their analysis heavily relies on this property. Unfortunately, our data matrix has column-wise dependence, so we are unable to use their techniques. We build our analysis from ground up.

There are recent works on statistical guarantees of learning a Markov model [15, 18, 28, 37, 35]. They target on estimating the transition matrix, but our focus is to estimate the aggregation/disaggregation distributions. Given an estimator of the transition matrix, how to obtain the aggregation/disaggregation distributions is non-trivial (for example, simply performing a vanilla PCA on the estimator doesn't work). To our best knowledge, our result is the first statistical guarantee for estimating the aggregation/disaggregation distributions. We can also use our estimator of aggregation/disaggregation distributions to form an estimator of the transition matrix, and it can achieve the known minimax rate for a range of parameter settings. See Section 4 for more discussions.

## 2 Soft State Aggregation Model with Anchor States

We say that a Markov chain $X_0, X_1, \ldots, X_n$ admits a *soft state aggregation* with $r$ meta-states, if there exist random variables $Z_0, Z_1, \ldots, Z_{n-1} \in \{1, 2, \ldots, r\}$ such that

$$\mathbb{P}(X_{t+1} \mid X_t) = \sum_{k=1}^{r} \mathbb{P}(Z_t = k \mid X_t) \cdot \mathbb{P}(X_{t+1} \mid Z_t = k), \tag{1}$$

for all $t$ with probability 1. Here, $\mathbb{P}(Z_t \mid X_t)$ and $\mathbb{P}(X_{t+1} \mid Z_t)$ are independent of $t$ and referred to as the *aggregation distributions* and *disaggregation distributions*, respectively. The soft state aggregation model has been discussed in various literatures (e.g., [32, 37]), where $r$ is presumably much smaller than $p$. See [5, Section 6.3.7] for a textbook review. This decomposition means that one can map the states into meta-states while preserving the system's dynamics. In the special case where each aggregation distribution is degenerate (we say a discrete distribution is *degenerate* if only one outcome is possible), it reduces to the hard state aggregation model or lumpable Markov model.

The soft state aggregation model has a matrix form. Let $\mathbf{P} \in \mathbb{R}^{p \times p}$ be the transition matrix, where $P_{ij} = \mathbb{P}(X_{t+1} = j \mid X_t = i)$. We introduce $\mathbf{U} \in \mathbb{R}^{p \times r}$ and $\mathbf{V} \in \mathbb{R}^{p \times r}$, where $U_{ik} = \mathbb{P}(Z_t = k \mid X_t = i)$ and $V_{jk} = \mathbb{P}(X_{t+1} = j \mid Z_t = k)$. Each row of $\mathbf{U}$ is an aggregation distribution, and each column of $\mathbf{V}$ is a disaggregation distribution. Then, (1) is equivalent to ($\mathbf{1}_s$: the vector of 1's)

$$\mathbf{P} = \mathbf{U}\mathbf{V}^{\top}, \qquad \text{where } \mathbf{U}\mathbf{1}_r = \mathbf{1}_p \text{ and } \mathbf{V}^{\top}\mathbf{1}_p = \mathbf{1}_r. \tag{2}$$

Here, $\mathbf{U}$ and $\mathbf{V}$ are not identifiable, unless with additional conditions. We assume that each meta-state has a signature raw state, defined either through the aggregation process or the disaggregation process.

**Definition 1** (Anchor State). *A state $i$ is called an "aggregation anchor state" of the meta-state $k$ if $U_{ik} = 1$ and $U_{is} = 0$ for all $s \neq k$. A state $j$ is called a "disaggregation anchor state" of the meta-state $k$, if $V_{jk} > 0$ and $V_{js} = 0$ for all $s \neq k$.*

An aggregation anchor state transits to only one meta-state, and a disaggregation anchor state can be transited from only one meta-state. Since (2) is indeed a nonnegative matrix factorization (NMF), the definition of anchor states are natural analogs of "separable features" in NMF [13]. They are also natural analogs of "pure documents" and "anchor words" in topic modeling. We note that in a hard state aggregation model, every state is an anchor state by default.

Throughout this paper, we take the following assumption:

**Assumption 1.** *There exists at least one disaggregation anchor state for each meta-state.*

By well-known results in NMF [13], this assumption guarantees that $\mathbf{U}$ and $\mathbf{V}$ are uniquely defined by (2), provided that $\mathbf{P}$ has a rank of $r$. Our results can be extended to the case where each meta-state has an aggregation anchor state (see the remark in Section 3). For simplicity, from now on, we call a disaggregation anchor state an *anchor state* for short.

*The introduction of anchor states not only guarantees identifiability but also enhances interpretability of the model.* This is demonstrated in an application to New York city taxi traffic data. We model the taxi traffic by a finite-state Markov chain, where each state is a pick-up/drop-off location in the city. Figure 1 illustrates the estimated soft-state aggregation model (see Section 5 for details). The

estimated anchor states coincide with notable landmarks in Manhattan, such as Times square area, the museum area on Park avenue, etc. Hence, each meta-state (whose disaggregation distribution is plotted via a heat map over Manhattan in (c)) can be nicely interpreted as a representative traffic mode with exclusive destinations (e.g., the traffic to Times square, the traffic to WTC Exchange, the traffic to museum park, etc.). In contrast, if we don't explore anchor states but simply use PCA to conduct state aggregation, the obtained meta-states have no clear association with notable landmarks, so are hard to interpret. The interpretability of our model also translates to better performance in downstream tasks in reinforcement learning (see Section 5).

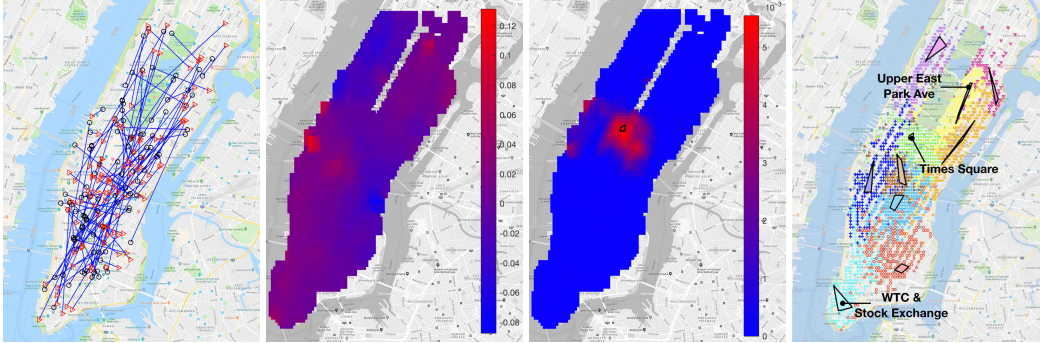

Figure 1: **Soft state aggregation learned from NYC taxi data.** Left to right: (a) Illustration of 100 taxi trips (O: pick-up location, △: drop-off location). (b) A principal component of $\mathbf{P}$ (heat map), lacking interpretability. (c) Disaggregation distribution of $\mathbf{P}$ corresponding to the Times square anchor region. (d) Ten anchor regions identified by Alg. 1, coinciding with landmarks of NYC.

## 3   An Unsupervised State Aggregation Algorithm

Given a Markov trajectory $\{X_0, X_1, \ldots, X_n\}$ from a state-transition system, let $\mathbf{N} \in \mathbb{R}^{p \times p}$ be the matrix consisting of empirical state-transition counts, i.e., $\mathbf{N}_{ij} = \sum_{t=0}^{n-1} \mathbf{1}\{X_t = i, X_{t+1} = j\}$. Our algorithm takes as input the matrix $\mathbf{N}$ and the number of meta-states $r$, and it estimates (a) the disaggregation distributions $\mathbf{V}$, (b) the aggregation distributions $\mathbf{U}$, and (c) the anchor states. See Algorithm 1. Part (a) is the core of the algorithm, which we explain below.

*Insight 1: Disaggregation distributions are linear transformations of right singular vectors.*

We consider an oracle case, where the transition matrix $\mathbf{P}$ is given and we hope to retrieve $\mathbf{V}$ from $\mathbf{P}$. Let $\mathbf{H} = [\mathbf{h}_1, \ldots, \mathbf{h}_r] \in \mathbb{R}^{p \times r}$ be the matrix containing first $r$ right singular vectors of $\mathbf{P}$. Let $\mathrm{Span}(\cdot)$ denote the column space of a matrix. By linear algebra, $\mathrm{Span}(\mathbf{H}) = \mathrm{Span}(\mathbf{P}^\top) = \mathrm{Span}(\mathbf{V}\mathbf{U}^\top) = \mathrm{Span}(\mathbf{V})$. Hence, the columns of $\mathbf{H}$ and the columns of $\mathbf{V}$ are two different bases of the same $r$-dimensional subspace. It follows immediately that there exists $\mathbf{L} \in \mathbb{R}^{r \times r}$ such that $\mathbf{H} = \mathbf{V}\mathbf{L}$. On the one hand, this indicates that singular vectors are not valid estimates of disaggregation distributions, as each singular vector $\mathbf{h}_k$ is a linear combination of multiple disaggregation distributions. On the other hand, it suggests a promising two-step procedure for recovering $\mathbf{V}$ from $\mathbf{P}$: (i) obtain the right singular vectors $\mathbf{H}$, (ii) identify the matrix $\mathbf{L}$ and retrieve $\mathbf{V} = \mathbf{H}\mathbf{L}^{-1}$.

*Insight 2: The linear transformation of $\mathbf{L}$ is estimable given the existence of anchor states.*

The estimation of $\mathbf{L}$ hinges on a geometrical structure induced by the anchor state assumption [13]: Let $\mathcal{C}$ be a simplicial cone with $r$ supporting rays, where the directions of the supporting rays are specified by the $r$ rows of the matrix $\mathbf{L}$. If $j$ is an anchor state, then the $j$-th row of $\mathbf{H}$ lies on one supporting ray of this simplicial cone. If $j$ is not an anchor state, then the $j$-th row of $\mathbf{H}$ lies in the interior of the simplicial cone. See the left panel of Figure 2 for illustration with $r = 3$. Once we identify the $r$ supporting rays of this simplicial cone, we immediately obtain the desired matrix $\mathbf{L}$.

*Insight 3: Normalization on eigenvectors is the key to estimation of $\mathbf{L}$ under noise corruption.*

In the real case where $\mathbf{N}$, instead of $\mathbf{P}$, is given, we can only obtain a noisy version of $\mathbf{H}$. With noise corruption, to estimate supporting rays of a simplicial cone is very challenging. [22] discovered that a particular row-wise normalization on $\mathbf{H}$ manages to "project" the simplicial cone to a simplex with $r$ vertices. Then, for all anchor states of one meta-state, their corresponding rows collapse to one single point in the noiseless case (and a tight cluster in the noisy case). The task reduces to estimating

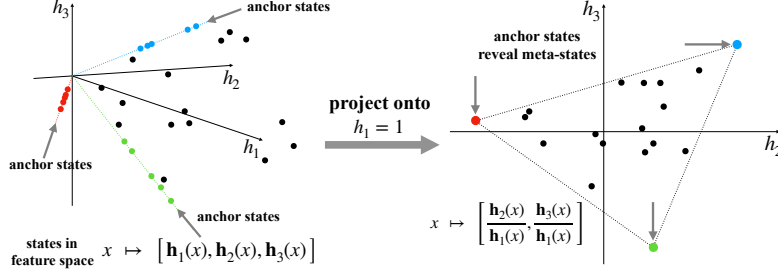

Figure 2: **Geometrical structure of anchor states**. Left: Each dot is a row of the matrix $\mathbf{H} = [\mathbf{h}_1, \mathbf{h}_2, \mathbf{h}_3]$. The data points are contained in a simplicial cone with three supporting rays. Right: Each dot is a re-scaled row of $\mathbf{H}$ by SCORE, where the first coordinate is dropped and so every row is in a plane. The data points are contained in a simplex with three vertices.

---

**Algorithm 1** Learning the Soft State Aggregation Model.

---

**Input**: empirical state-transition counts $\mathbf{N}$, number of meta-states $r$, anchor state threshold $\delta_0$

1. Estimate the matrix of disaggregation distributions $\mathbf{V}$.

   (i) Conduct SVD on $\widetilde{\mathbf{N}} = \mathbf{N}[\mathrm{diag}(\mathbf{N}^\top \mathbf{1}_p)]^{-1/2} \in \mathbb{R}^{p \times p}$, and let $\widehat{\mathbf{h}}_1, ..., \widehat{\mathbf{h}}_r$ denote the first $r$ right singular vectors. Obtain a matrix $\widehat{\mathbf{D}} = [\mathrm{diag}(\widehat{\mathbf{h}}_1)]^{-1}[\widehat{\mathbf{h}}_2, \ldots, \widehat{\mathbf{h}}_r] \in \mathbb{R}^{p \times (r-1)}$.

   (ii) Run an existing vertex finding algorithm to rows of $\widehat{\mathbf{D}}$. Let $\widehat{\mathbf{b}}_1, \ldots, \widehat{\mathbf{b}}_r$ be the output vertices. (In our numerical experiments, we use the vertex hunting algorithm in [21]).

   (iii) For $1 \le j \le p$, compute
   $$\widehat{\mathbf{w}}_j^* = \mathrm{argmin}_{\mathbf{q} \in \mathbb{R}^r} \left\| \widehat{\mathbf{d}}_j - \sum_{k=1}^r q_k \widehat{\mathbf{b}}_k \right\|^2 + \left(1 - \sum_{k=1}^r q_k\right)^2.$$

   Set the negative entries in $\widehat{\mathbf{w}}_j^*$ to zero and renormalize it to have a unit $\ell^1$-norm. The resulting vector is denoted as $\widehat{\mathbf{w}}_j$. Let $\widehat{\mathbf{W}} = [\widehat{\mathbf{w}}_1, \widehat{\mathbf{w}}_2, \ldots, \widehat{\mathbf{w}}_p]^\top \in \mathbb{R}^{p \times r}$. Obtain the matrix $[\mathrm{diag}(\widehat{\mathbf{h}}_1)][\mathrm{diag}(\mathbf{N}^\top \mathbf{1}_p)]^{1/2} \widehat{\mathbf{W}}$ and re-normalize each column of it to have a unit $\ell^1$-norm. The resulting matrix is $\widehat{\mathbf{V}}$.

2. Estimate matrix of aggregation distributions $\mathbf{U}$. Let $\widehat{\mathbf{P}} = [\mathrm{diag}(\mathbf{N}\mathbf{1}_p)]^{-1}\mathbf{N}$ be the empirical transition probability matrix. Estimate $\mathbf{U}$ by
   $$\widehat{\mathbf{U}} = \widehat{\mathbf{P}}\widehat{\mathbf{V}}(\widehat{\mathbf{V}}^\top \widehat{\mathbf{V}})^{-1},$$

3. Estimate the set of anchor states. Let $\widehat{\mathbf{w}}_j$ be as in Step (iii). Let
   $$\mathcal{A} = \left\{ 1 \le j \le p : \max_{1 \le k \le r} \widehat{\mathbf{w}}_j(k) \ge 1 - \delta_0 \right\}.$$

**Output**: estimates $\widehat{\mathbf{V}}$ and $\widehat{\mathbf{U}}$, set of anchor states $\mathcal{A}$

---

the vertices of a simplex, which is much easier to handle under noise corruption. This particular normalization is called SCORE [20]. It re-scales each row of $\mathbf{H}$ by the first coordinate of this row. After re-scaling, the first coordinate is always 1, so it is eliminated; the normalized rows then have $(r-1)$ coordinates. See the right panel of Figure 2. Once we identify the $r$ vertices of this simplex, we can use them to re-construct $\mathbf{L}$ in a closed form [22].

These insights have casted estimation of $\mathbf{V}$ to a simplex finding problem: Given data $\widehat{\mathbf{d}}_1, \ldots, \widehat{\mathbf{d}}_p \in \mathbb{R}^{r-1}$, suppose they are noisy observations of non-random vectors $\mathbf{d}_1, \ldots, \mathbf{d}_p \in \mathbb{R}^{r-1}$ (in our case, each $\mathbf{d}_j$ is a row of the matrix $\mathbf{H}$ after SCORE normalization), where these non-random vectors are contained in a simplex with $r$ vertices $\mathbf{b}_1, \ldots, \mathbf{b}_r$ with at least one $\mathbf{d}_j$ located on each vertex. We aim to estimate the vertices $\mathbf{b}_1, \ldots, \mathbf{b}_r$. This is a well-studied problem in the literature, also known as *linear unmixing* or *archetypal analysis*. There are a number of existing algorithms [8, 3, 29, 9, 21]. For example, the successive projection algorithm [3] has a complexity of $O(pr^4)$.

Algorithm 1 follows the insights above but is more sophisticated. Due to space limit, we relegate the detailed explanation to Appendix. It has a complexity of $O(p^3)$, where main cost is from SVD.

**Remark** (*The case with aggregation anchor states*). The anchor states here are the disaggregation anchor states in Defintion 1. Instead, if each meta-state has an aggregation anchor state, there is a similar geometrical structure associated with the left singular vectors. We can modify our algorithm to first use left singular vectors to estimate $\mathbf{U}$ and then estimate $\mathbf{V}$ and anchor states.

# 4 Main Statistical Results

Our main results are twofold. The first is row-wise large-deviation bounds for empirical singular vectors. The second is statistical guarantees of learning the soft state aggregation model. Throughout this section, suppose we observe a trajectory $\{X_0, X_1, \ldots, X_n\}$ from an ergodic Markov chain with $p$ states, where the transition matrix $\mathbf{P}$ satisfies (2) with $r$ meta-states. Let $\boldsymbol{\pi} \in \mathbb{R}^p$ denote the stationary distribution. Define a mixing time $\tau_* = \min\{k \geq 1 : \max_{1 \leq i \leq p} \|(\mathbf{P}^k)_{i,\cdot} - \boldsymbol{\pi}^\top\|_1 \leq 1/2\}$. We assume there are constants $c_1, C_1, \bar{C}_1, c_2, c_3, C_4 > 0$ such that the following conditions hold:

    (a) The stationary distribution $\boldsymbol{\pi}$ satisfies $c_1 p^{-1} \leq \boldsymbol{\pi}_j \leq C_1 p^{-1}$, for $j = 1, 2, \ldots, p$.
    (b) The stationary distribution on meta-states satisfies $(\mathbf{U}^\top \boldsymbol{\pi})_k \leq \bar{C}_1 r^{-1}$, for $k = 1, 2, \ldots, r$.
    (c) $\lambda_{\min}(\mathbf{U}^\top \mathbf{U}) \geq c_2 p r^{-1}$ and $\lambda_{\min}(\mathbf{V}^\top \mathbf{V}) \geq c_2 p^{-1} r$.
    (d) The first two singular values of $[\mathrm{diag}(\boldsymbol{\pi})]\mathbf{P}[\mathrm{diag}(\boldsymbol{\pi})]^{-1/2}$ satisfy $\sigma_1 - \sigma_2 \geq c_3 p^{-\frac{1}{2}}$.
    (e) The entries of the $r$-by-$r$ matrix $\mathbf{U}^\top \mathbf{P} \mathbf{V}$ satisfy $\frac{\max_{k,l}(\mathbf{U}^\top \mathbf{P} \mathbf{V})_{kl}}{\min_{k,l}(\mathbf{U}^\top \mathbf{P} \mathbf{V})_{kl}} \leq C_4$.

Conditions (a)-(b) require that the Markov chain has a balanced number of visits to each state and to each meta-state when reaching stationarity. Such conditions are often imposed for learning a Markov model [37, 28]. Condition (c) eliminates the aggregation (disaggregation) distributions from being highly collinear, so that each of them can be accurately identified from the remaining. Condition (d) is a mild eigen-gap condition, which is necessary for consistent estimation of eigenvectors from PCA [2, 22]. Condition (e) says that the meta-states are reachable from one-another and that meta-state transitions cannot be overly imbalanced.

## 4.1 Row-wise large-deviation bounds for singular vectors

At the core of the analysis of any spectral method (the classical PCA or our unsupervised algorithm) is characterization of the errors of approximating population eigenvectors by empirical eigenvectors. If we choose the loss function to be the Euclidean norm between two vectors, it reduces to deriving a bound for the spectral norm of noise matrix [12] and is often manageable. However, the Euclidean norm bound is useless for our problem: In order to obtain the total-variation bounds for estimating aggregation/disaggregation distributions, we need sharp error bounds for *each entry* of the eigenvector. Recently, there has been active research on entry-wise analysis of eigenvectors [2, 23, 24, 38, 11, 14]. Since eigenvectors depend on data matrix in a complicated and highly nonlinear form, such analysis is well-known to be challenging. More importantly, there is no universal technique that works for all problems, and such bounds are obtained in a problem-by-problem manner (e.g., for Gaussian covariance matrix [23, 24], network adjacency matrix [2, 21], and topic matrix [22]). As an addition to the nascent literature, we develop such results for transition count matrices of a Markov chain. The analysis is challenging due to that the entries of the count matrix are dependent of each other.

Recall that $\widetilde{\mathbf{N}}$ is the re-scaled transition count matrix introduced in Algorithm 1 and $\widehat{\mathbf{h}}_1, \ldots, \widehat{\mathbf{h}}_r$ are its first $r$ right singular vectors (our technique also applies to the original count matrix $\mathbf{N}$ and the empirical transition matrix $\widehat{\mathbf{P}}$). Theorem 1 and Theorem 2 deal with the leading singular vector and the remaining ones, respectively. ($\widetilde{O}$ means "bounded up to a logarithmic factor of $n, p$").

**Theorem 1** (Entry-wise perturbation bounds for $\widehat{\mathbf{h}}_1$). *Suppose the regularity conditions (a)-(e) hold. There exists a parameter $\omega \in \{\pm 1\}$ such that if $n = \widetilde{\Omega}(\tau_* p)$, then with probability at least $1 - n^{-1}$,*
$$\max_{1 \leq j \leq p} |\omega \widehat{\mathbf{h}}_1(j) - \mathbf{h}_1(j)| = \widetilde{O}\left((\sigma_2 - \sigma_1)^{-1}(1 + \sqrt{\tau_* p/n})\sqrt{\frac{\tau_*}{np}}\right).$$

**Theorem 2** (Row-wise perturbation bounds for $\widehat{\mathbf{H}}$). *Suppose the regularity conditions (a)-(e) hold. For $1 \leq s \leq t \leq r$, let $\mathbf{H}_* = [\mathbf{h}_s, \ldots, \mathbf{h}_t]$, $\widehat{\mathbf{H}}_* = [\widehat{\mathbf{h}}_s, \ldots, \widehat{\mathbf{h}}_t]$, and $\Delta_* = \min\{\sigma_{s-1} - \sigma_s, \sigma_t - \sigma_{t+1}\}$, where $\sigma_0 = +\infty$, $\sigma_{r+1} = 0$. If $n = \widetilde{\Omega}(\tau_* p)$, then with probability $1 - n^{-1}$, there is an orthogonal matrix $\boldsymbol{\Omega}_*$ such that $\max_{1 \leq j \leq p} \left\|\mathbf{e}_j^\top(\widehat{\mathbf{H}}_* \boldsymbol{\Omega}_* - \mathbf{H}_*)\right\|_2 = \widetilde{O}\left(\Delta_*^{-1}\left(1 + \sqrt{\tau_* p/nr}\right)\sqrt{\frac{\tau_* r}{np}}\right).$*

## 4.2 Statistical guarantees of soft state aggregation

We study the error of estimating $\mathbf{U}$ and $\mathbf{V}$, as well as recovering the set of anchor states. Algorithm 1 plugs in some existing algorithm for the simplex finding problem. We make the following assumption:

**Assumption 2** (Efficiency of simplex finding). *Given data $\widehat{\mathbf{d}}_1, \ldots, \widehat{\mathbf{d}}_p \in \mathbb{R}^{r-1}$, suppose they are noisy observations of non-random vectors $\mathbf{d}_1, \ldots, \mathbf{d}_p$, where these non-random vectors are contained*

*in a simplex with $r$ vertices $\mathbf{b}_1, \ldots, \mathbf{b}_r$ with at least one $\mathbf{d}_j$ located on each vertex. The simplex finding algorithm outputs $\widehat{\mathbf{b}}_1, \ldots, \widehat{\mathbf{b}}_r$ such that $\max_{1 \le k \le r} \|\widehat{\mathbf{b}}_k - \mathbf{b}_k\| \le C \max_{1 \le j \le p} \|\widehat{\mathbf{d}}_j - \widehat{\mathbf{d}}_j\|$.*

Several existing simplex finding algorithms satisfy this assumption, such as the successive projection algorithm [3], the vertex hunting algorithm [21, 22], and the algorithm of archetypal analysis [19]. Since this is not the main contribution of this paper, we refer the readers to the above references for details. In our numerical experiments, we use the vertex hunting algorithm in [21, 22].

First, we provide total-variation bounds between estimated individual aggregation/disaggregation distributions and the ground truth. Write $\mathbf{V} = [\mathbf{V}_1, \ldots, \mathbf{V}_r]$ and $\mathbf{U} = [\mathbf{u}_1, \ldots, \mathbf{u}_p]^\top$, where each $\mathbf{V}_k \in \mathbb{R}^p$ is a disaggregation distribution and each $\mathbf{u}_i \in \mathbb{R}^r$ is an aggregation distribution.

**Theorem 3** (Error bounds for estimating $\mathbf{V}$). *Suppose the regularity conditions (a)-(e) hold and Assumptions 1 and 2 are satisfied. When $n = \widetilde{\Omega}\big(\tau_* p^{\frac{3}{2}} r\big)$, with probability at least $1 - n^{-1}$, the estimate $\widehat{\mathbf{V}}$ given by Algorithm 1 satisfies $\frac{1}{r} \sum_{k=1}^{r} \big\|\widehat{\mathbf{V}}_k - \mathbf{V}_k\big\|_1 = \widetilde{O}\Big(\big(1 + p\sqrt{\tau_*/n}\big)\sqrt{\frac{\tau_* pr}{n}}\Big)$.*

**Theorem 4** (Error bounds for estimating $\mathbf{U}$). *Suppose the regularity conditions (a)-(e) hold and Assumptions 1 and 2 are satisfied. When $n = \widetilde{\Omega}\big(\tau_* p^{\frac{3}{2}} r\big)$, with probability at least $1 - n^{-1}$, the estimate $\widehat{\mathbf{U}}$ given by Algorithm 1 satisfies $\frac{1}{p} \sum_{j=1}^{p} \big\|\widehat{\mathbf{u}}_j - \mathbf{u}_j\big\|_1 = \widetilde{O}\Big(r^{\frac{3}{2}}\big(1 + p\sqrt{\tau_*/n}\big)\sqrt{\frac{\tau_* pr}{n}}\Big)$.*

Second, we provide sample complexity guarantee for the exact recovery of anchor states. To eliminate false positives, we need a condition that the non-anchor states are not too 'close' to an anchor state; this is captured by the quantity $\delta$ below. (Note $\delta_j = 0$ for anchor states $j \in \mathcal{A}^*$.)

**Theorem 5** (Exact recovery of anchor states). *Suppose the regularity conditions (a)-(e) hold and Assumptions 1 and 2 are satisfied. Let $\mathcal{A}^*$ be the set of (disaggregation) anchor states. Define $\delta_j = 1 - \max_{1 \le k \le r} \mathbb{P}_{X_0 \sim \boldsymbol{\pi}}(Z_0 = k \,|\, X_1 = j)$ and $\delta = \min_{j \notin \mathcal{A}^*} \delta_j$. Suppose the threshold $\delta_0$ in Algorithm 1 satisfies $\delta_0 = O(\delta)$. If $n = \widetilde{\Omega}\big(\delta_0^{-2} \tau_* p^{\frac{3}{2}} r\big)$, then $\mathbb{P}(\mathcal{A} = \mathcal{A}^*) \ge 1 - n^{-1}$.*

We connect our results to several lines of works in the literature. First, in the special case of $r = 1$, our problem reduces to learning a discrete distribution with $p$ outcomes, where the minimax rate of total-variation distance is $O(\sqrt{p/n})$ [17]. Our bound matches with this rate when $p = O(\sqrt{n})$. However, our problem is much harder: each row of $\mathbf{P}$ is a mixture of $r$ discrete distributions. Second, our setting is connected to the setting of learning a mixture of discrete distributions [30, 27] but is different in important ways. Those works consider learning *one* mixture distribution, and the data are $iid$ observations. Our problem is to estimate $p$ mixture distributions, which share the same basis distributions but have different mixing proportions, and our data are a single trajectory of a Markov chain. Third, our problem is connected to topic modeling [4, 22], where we may view the empirical transition profile of each raw state as a 'document'. However, in topic modeling, the documents are independent of each other, but the 'documents' here are highly dependent as they are generated from a single trajectory of a Markov chain. Last, we compare with the literature of estimating the transition matrix $\mathbf{P}$ of a Markov model. Without low-rank assumptions on $\mathbf{P}$, the minimax rate of the total variation error is $O(p/\sqrt{n})$ [35] (also, see [25] and reference therein for related settings in hidden Markov models); with a low-rank structure on $\mathbf{P}$, the minimax rate becomes $O(\sqrt{rp/n})$ [37]. To compare, we use our estimator of $(\mathbf{U}, \mathbf{V})$ to construct an estimator of $\mathbf{P}$ by $\widehat{\mathbf{P}} = \widehat{\mathbf{U}}\widehat{\mathbf{V}}^\top$. When $r$ is bounded and $p = O(\sqrt{n})$, this estimator achieves a total-variation error of $O(\sqrt{rp/n})$, which is optimal. At the same time, we want to emphasize that estimating $(\mathbf{U}, \mathbf{V})$ is a more challenging problem than estimating $\mathbf{P}$, and we are not aware of any existing theoretical results of the former.

**Simulation.** We test our method on simulations (settings are in the appendix). The results are summarized in Figure 3. It suggests: (a) the rate of convergence in Theorem 3 is confirmed by numerical evidence, and (b) our method compares favorably with existing methods on estimating $\mathbf{P}$ (to our best knowledge, there is no other method that directly estimates $\mathbf{U}$ and $\mathbf{V}$; so we instead compare the estimation on $\mathbf{P}$).

# 5    Analysis of NYC Taxi Data and Application to Reinforcement Learning

We analyze a dataset of $1.1 \times 10^7$ New York city yellow cab trips that were collected in January 2016 [1]. We treat each taxi trip as a sample transition generated by a city-wide Markov process over NYC, where the transition is from a pick-up location to some drop-off location. We discretize the map into $p = 1922$ cells so that the Markov process becomes a finite-state one.

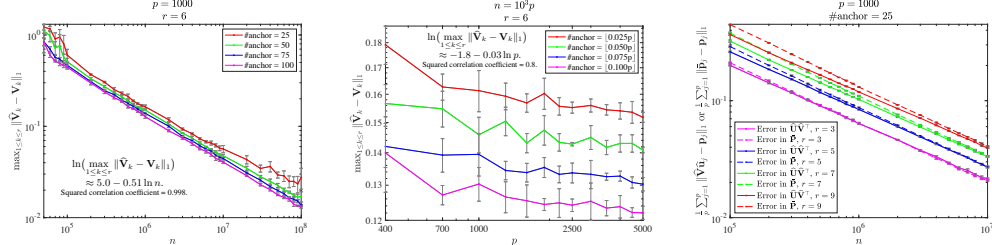

Figure 3: **Simulation Results.** Left: Total variation error on $\mathbf{V}$ ($p$ is fixed and $n$ varies). Middle: Total variation error on $\mathbf{V}$ ($p/n$ is fixed). Both panels validate the scaling of $\sqrt{p/n}$ in Theorem 3. Right: Recovery error of $\mathbf{P}$, where $\widehat{\mathbf{U}}\widehat{\mathbf{V}}^{\top}$ is our method and $\bar{\mathbf{P}}$ is the spectral estimator [37] (note: this method cannot estimate $\mathbf{U},\mathbf{V}$ ).

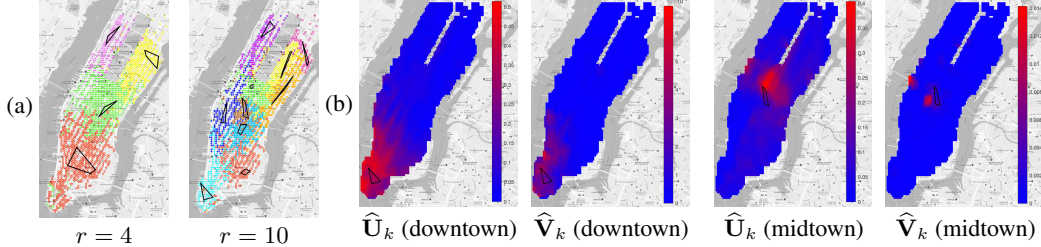

$r = 4$    $r = 10$    $\widehat{\mathbf{U}}_k$ (downtown)    $\widehat{\mathbf{V}}_k$ (downtown)    $\widehat{\mathbf{U}}_k$ (midtown)    $\widehat{\mathbf{V}}_k$ (midtown)

Figure 4: **State aggregation results.** (a) Estimated anchor regions and partition of NYC. (b) Estimated disaggregation distribution ($\widehat{\mathbf{V}}_k$) and aggregation likelihood ($\widehat{\mathbf{U}}_k$) for two meta-states.

**Anchor regions and partition.** We apply Alg. 1 to the taxi-trip data with $r = 4, 10$. The algorithm identifies sets of anchor states that are close to the vertices, as well as columns of $\widehat{\mathbf{U}}$ and $\widehat{\mathbf{V}}$ corresponding to each vertex (anchor region). We further use the estimated $\widehat{\mathbf{U}}, \widehat{\mathbf{V}}$ to find a partition of the city. Recall that in Algorithm 1, each state is projected onto a simplex, which can be represented as a convex combination of simplex's vertices (see Figure 2). For each state, we assign the state to a cluster that corresponds to largest value in the weights of convex combination. In this way, we can cluster the 1922 locations into a small number of regions. The partition results are shown in Figure 4 (a), where anchor regions are marked in each cluster.

**Estimated aggregation and disaggregation distributions.** Let $\widehat{\mathbf{U}}, \widehat{\mathbf{V}}$ be the estimated aggregation and disaggregation matrices. We use heat maps to visualize their columns. Take $r = 10$ for example. We pick two meta-states, with anchor states in the downtown and midtown areas, respectively. We plot in Figure 4 (b) the corresponding columns of $\widehat{\mathbf{U}}$ and $\widehat{\mathbf{V}}$. Each column of $\widehat{\mathbf{V}}$ is a disaggregation distribution, and each column of $\widehat{\mathbf{U}}$ can be thought of as a likelihood function for transiting to corresponding meta-states. The heat maps reveal the leading "modes" of the traffic-dynamics.

**Aggregation distributions used as features for RL.** Soft state aggregation can be used to reduce the complexity of reinforcement learning (RL) [32]. Aggregation/disaggregation distributions provide features to parameterize high-dimensional policies, in conjunction with feature-based RL methods [10, 36]. Next we experiment with using the aggregation distributions as features for RL.

Consider the taxi-driving policy optimization problem. The driver's objective is to maximize the daily revenue - a Markov decision process where the driver chooses driving directions in realtime based on locations. We compute the optimal policy using feature-based RL [10] and simulated NYC traffic. The algorithm takes as input 27 estimated aggregation distributions as *state features*. For comparison, we also use a hard partition of the city which is handpicked according to 27 NYC districts. RL using aggregation distributions as features achieves a daily revenue of $230.57, while the method using handpicked partition achieves $209.14. Figure 5 plots the optimal driving policy. This experiment suggests that (1) state aggregation learning provides features for RL automatically; (2) using aggregation distributions as features leads to better performance of RL than using handpicked features.

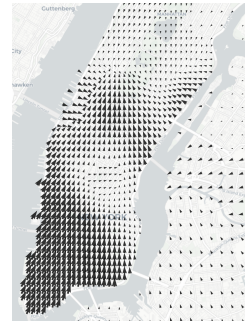

Figure 5: **The optimal driving policy** learnt by feature-based RL with estimated aggregation distributions as state features. Arrows point out the most favorable directions and the thickness is proportional to favorability of the direction.

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
