[Supplementary Material]

# Supplement for "State Aggregation Learning from Markov Transition Data"

**Yaqi Duan**
Princeton University
yaqid@princeton.edu

**Zheng Tracy Ke**
Harvard University
zke@fas.harvard.edu

**Mengdi Wang**
Princeton University
mengdiw@princeton.edu

## Contents

# A    Preliminaries

**Notations.**

In the following, we denote the frequency matrix $[\mathrm{diag}(\boldsymbol{\pi})]\mathbf{P}$ by $\mathbf{F}$. The algorithm deals with $\widetilde{\mathbf{N}} = \mathbf{N}[\mathrm{diag}(\mathbf{m})]^{-\frac{1}{2}}$ where $\mathbf{m} = \mathbf{N}^{\top}\mathbf{1}_p$. We normalize $\widetilde{\mathbf{N}}$ with $\sqrt{n}$ and define $\widehat{\mathbf{Q}} = n^{-\frac{1}{2}}\mathbf{N}[\mathrm{diag}(\mathbf{m})]^{-\frac{1}{2}}$. Martrix $\widehat{\mathbf{Q}}$ has a population counterpart $\mathbf{Q} = \mathbf{F}[\mathrm{diag}(\boldsymbol{\pi})]^{-1/2}$. The soft state aggregation assumption implies that $\mathbf{Q}$ has a singular value decomposition $\mathbf{Q} = \mathbf{G}\boldsymbol{\Sigma}\mathbf{H}^{\top}$, where $\mathbf{G} = \begin{bmatrix} \mathbf{g}_1, \mathbf{g}_2, \ldots, \mathbf{g}_r \end{bmatrix} \in \mathbb{R}^{p \times r}$, $\mathbf{H} = \begin{bmatrix} \mathbf{h}_1, \mathbf{h}_2, \ldots, \mathbf{h}_r \end{bmatrix} \in \mathbb{R}^{p \times r}$, $\boldsymbol{\Sigma} = \mathrm{diag}(\sigma_1, \sigma_2, \ldots, \sigma_r)$, $\sigma_1 \geq \sigma_2 \geq \ldots \geq \sigma_r > 0$. Analogously, suppose that $\widehat{\mathbf{Q}}$ has singular values $\widehat{\sigma}_1 \geq \widehat{\sigma}_2 \geq \ldots \geq \widehat{\sigma}_p \geq 0$. For $i = 1, 2, \ldots, p$, $\widehat{\mathbf{g}}_i$ and $\widehat{\mathbf{h}}_i$ are respectively the left and right singular vectors associated with $\widehat{\sigma}_i$.

For any $k \in \mathbb{N}$, let $\mathbb{S}^{k-1}$ denote the unit sphere in $\mathbb{R}^k$ and $\mathbf{I}_k \in \mathbb{R}^{k \times k}$ denote the identity matrix.

**Assumptions.**

In the paper, we propose the following regularity conditions:

**Assumption 1** (Regularity conditions). *There exists constants $c_1$, $C_1$, $\bar{C}_1$, $c_2$, $c_3$, $C_4 > 0$ such that*

1. *the stationary distribution $\boldsymbol{\pi}$ satisfies*
$$c_1 p^{-1} \leq \boldsymbol{\pi}_j \leq C_1 p^{-1}, \quad for\ j = 1, 2, \ldots, p; \tag{A.1}$$

2. *the stationary distribution on meta-states satisfies*
$$\left(\mathbf{U}^{\top}\boldsymbol{\pi}\right)_k \leq \bar{C}_1 r^{-1}, \quad for\ k = 1, 2, \ldots, r; \tag{A.2}$$

3. *the aggregation and disaggregation distributions satisfy*
$$\lambda_{\min}\left(\mathbf{U}^{\top}\mathbf{U}\right) \geq c_2 p r^{-1}, \ \lambda_{\min}\left(\mathbf{V}^{\top}\mathbf{V}\right) \geq c_2 p^{-1} r; \tag{A.3}$$

4. *the first and second largest singular values of $[\mathrm{diag}(\boldsymbol{\pi})]\mathbf{P}[\mathrm{diag}(\boldsymbol{\pi})]^{-1/2}$ satisfy*
$$\sigma_1 - \sigma_2 \geq c_3 p^{-\frac{1}{2}}; \tag{A.4}$$

5. *the ratio between the largest and smallest entries in an $r$-by-$r$ matrix $\mathbf{U}^{\top}\mathbf{P}\mathbf{V}$ satisfies*
$$\frac{\max_{k,l}\left(\mathbf{U}^{\top}\mathbf{P}\mathbf{V}\right)_{kl}}{\min_{k,l}\left(\mathbf{U}^{\top}\mathbf{P}\mathbf{V}\right)_{kl}} \leq C_4. \tag{A.5}$$

We can rewrite (A.3) into the following form, which is more convenient for our proofs:
$$\begin{aligned} \lambda_{\min}\left(\mathbf{U}^{\top}[\mathrm{diag}(\boldsymbol{\pi})]^2\mathbf{U}\right) &\geq c_2' p^{-1} r^{-1}, \\ \lambda_{\min}\left(\mathbf{V}^{\top}[\mathrm{diag}(\boldsymbol{\pi})]^{-1}\mathbf{V}\right) &\geq c_2' r. \end{aligned} \tag{A.6}$$

To see the equivalence between (A.3) and (A.6), we note that under assumption (A.1),
$$\mathbf{U}^{\top}\left([\mathrm{diag}(\boldsymbol{\pi})]^2 - c_1^2 p^{-2}\mathbf{I}_p\right)\mathbf{U} \succeq 0,$$

thus by (A.3),
$$\mathbf{U}^{\top}[\mathrm{diag}(\boldsymbol{\pi})]^2\mathbf{U} \succeq c_1^2 p^{-2}\mathbf{U}^{\top}\mathbf{U} \succeq c_1^2 c_2 p^{-1} r^{-1}\mathbf{I}_r.$$

Similarly,
$$\mathbf{V}^{\top}\left([\mathrm{diag}(\boldsymbol{\pi})]^{-1} - C_1^{-1}p\mathbf{I}_p\right)\mathbf{V} \succeq 0,$$

which implies
$$\mathbf{V}^{\top}[\mathrm{diag}(\boldsymbol{\pi})]^{-1}\mathbf{V} \succeq C_1^{-1}p\mathbf{V}^{\top}\mathbf{V} \succeq C_1^{-1}c_2 r\mathbf{I}_r.$$

For simplicity, we replace the parameter $c_2'$ in (A.6) by $c_2$ in the subsequent discussions.

**Preliminary lemmas.**

We list some preliminary lemmas that will be used later, of which the proofs can be found in Appendix E .

**Lemma A .1.** *Under assumptions* (A .1) *and* (A .6)*,*

$$c_2 p^{-\frac{1}{2}} \leq \sigma_r \leq \sigma_1 \leq C_1 c_1^{-\frac{1}{2}} p^{-\frac{1}{2}}. \tag{A .7}$$

**Lemma A .2.** *Under assumptions* (A .1) *and* (A .6)*, we have*

$$\left\|\mathbf{e}_j^\top \mathbf{G}\right\|_2 \leq c_2^{-\frac{1}{2}} \pi_j \sqrt{pr}, \tag{A .8}$$

$$\left\|\mathbf{e}_j^\top \mathbf{H}\right\|_2 \leq C_1 c_2^{-\frac{3}{2}} \sqrt{\pi_j r}. \tag{A .9}$$

**Lemma A .3.** *Let* $\mathbf{L} \in \mathbb{R}^{r \times r}$ *be the matrix defined by* (E .4) *and* $\mathbf{l}_1 \in \mathbb{R}^r$ *be the first column of* $\mathbf{L}$. *Under assumptions* (A .1), (A .6) *and* (A .5)*, there exist constants* $c > 0$ *and* $C > 0$ *such that*

$$cr^{-1} \leq \mathbf{l}_1(k) \leq Cr^{-1} \quad for\ k = 1, 2, \ldots, r, \qquad and \qquad c_2^{\frac{1}{2}} \sqrt{r} \leq \left\|\mathbf{L}^{-1}\right\|_2 \leq C_1^{\frac{1}{2}} c_2^{-\frac{1}{2}} \sqrt{r}.$$

**Lemma A .4.** *Under assumptions* (A .1), (A .2), (A .6) *and* (A .5)*, there exist constants* $c > 0$ *and* $C > 0$ *such that for* $j = 1, 2, \ldots, p$,

$$c\sqrt{\pi_j} \leq \mathbf{h}_1(j) \leq C\sqrt{\pi_j}, \qquad 0 \leq \mathbf{g}_1(j) \leq C\pi_j \sqrt{p}. \tag{A .10}$$

# B   Proof of Entry-wise Eigenvector Bounds

A building block of our method is to get a sharp error bound for each row of $\widehat{\mathbf{H}} = \left[\widehat{\mathbf{h}}_1, \ldots, \widehat{\mathbf{h}}_r\right]$ and each entry of $\widehat{\mathbf{h}}_1$.

## B .1   Deterministic Analysis

**Lemma B .1** (A deterministic row-wise perturbation bound for singular vectors)**.** *For* $1 \leq s \leq t \leq r$, *denote* $\mathbf{H} = \left[\mathbf{h}_s, \mathbf{h}_{s+1}, \ldots, \mathbf{h}_t\right]$ *and* $\widehat{\mathbf{H}} = \left[\widehat{\mathbf{h}}_s, \widehat{\mathbf{h}}_{s+1}, \ldots, \widehat{\mathbf{h}}_t\right]$. *Let*

$$\Delta = \min\left\{\sigma_{s-1} - \sigma_s, \sigma_t - \sigma_{t+1}, \sigma_t\right\},$$

*where* $\sigma_0 = +\infty$ *and* $\sigma_{r+1} = 0$ *for simplicity. Suppose that* $\left\|\widehat{\mathbf{Q}} - \mathbf{Q}\right\|_2 \leq \frac{\Delta}{2}$, $\Delta > 0$. *Then there exists an orthogonal matrix* $\boldsymbol{\Omega} \in \mathbb{R}^{(t-s+1)\times(t-s+1)}$ *such that*

$$
\begin{aligned}
\left\|\mathbf{e}_j^\top (\widehat{\mathbf{H}}\boldsymbol{\Omega} - \mathbf{H})\right\|_2 \leq &\frac{2}{\Delta}\left\|\boldsymbol{G}^\top(\widehat{\mathbf{Q}} - \mathbf{Q})\mathbf{e}_j\right\|_2 \\
&+ \frac{4(1+\sqrt{2})}{\Delta}\left\|\mathbf{e}_j^\top \mathbf{H}\right\|_2\left\|\widehat{\mathbf{Q}} - \mathbf{Q}\right\|_2 \\
&+ \frac{8}{\Delta^2}\left\|\widehat{\mathbf{Q}} - \mathbf{Q}\right\|_2^2,
\end{aligned}
\tag{B .1}
$$

*where* $\mathbf{H} = [\mathbf{h}_1, \mathbf{h}_2, \ldots, \mathbf{h}_r]$ *and* $\boldsymbol{G} = \left[\mathbf{g}_s, \mathbf{g}_{s+1}, \ldots, \mathbf{g}_t\right]$.

*Proof.* We apply a symmetric dilation to matrices $\widehat{\mathbf{Q}}$ and $\mathbf{Q}$, so as to relate the singular vectors $\widehat{\mathbf{H}} = \left[\widehat{\mathbf{h}}_s, \widehat{\mathbf{h}}_{s+1}, \ldots, \widehat{\mathbf{h}}_t\right]$ and $\mathbf{H} = \left[\mathbf{h}_s, \mathbf{h}_{s+1}, \ldots, \mathbf{h}_t\right]$ to the eigen vectors of some symmetric matrices. Define

$$\widehat{\mathbf{Y}} \equiv \begin{bmatrix} \mathbf{0} & \widehat{\mathbf{Q}} \\ \widehat{\mathbf{Q}}^\top & \mathbf{0} \end{bmatrix} \quad \text{and} \quad \mathbf{Y} \equiv \begin{bmatrix} \mathbf{0} & \mathbf{Q} \\ \mathbf{Q}^\top & \mathbf{0} \end{bmatrix}. \tag{B .2}$$

The symmetric matrix $\widehat{\mathbf{Y}}$ has eigen pairs $(\widehat{\sigma}_i, \widehat{\boldsymbol{\xi}}_i)$ and $(-\widehat{\sigma}_i, \widehat{\boldsymbol{\xi}}_{-i})$ with

$$\widehat{\boldsymbol{\xi}}_i = \frac{1}{\sqrt{2}}\begin{bmatrix} \widehat{\mathbf{g}}_i \\ \widehat{\mathbf{h}}_i \end{bmatrix} \quad \text{and} \quad \widehat{\boldsymbol{\xi}}_{-i} = \frac{1}{\sqrt{2}}\begin{bmatrix} \widehat{\mathbf{g}}_i \\ -\widehat{\mathbf{h}}_i \end{bmatrix}$$

for $i = 1, 2, \ldots, p$. Analogously,

$$\boldsymbol{\xi}_i = \frac{1}{\sqrt{2}} \begin{bmatrix} \mathbf{g}_i \\ \mathbf{h}_i \end{bmatrix} \quad \text{and} \quad \boldsymbol{\xi}_{-i} = \frac{1}{\sqrt{2}} \begin{bmatrix} \mathbf{g}_i \\ -\mathbf{h}_i \end{bmatrix}$$

are eigen vectors of $\mathbf{Y}$ associated with $\sigma_i$ and $-\sigma_i$ for $i = 1, 2, \ldots, r$. Define

$$\boldsymbol{\Xi} = \begin{bmatrix} \boldsymbol{\xi}_1, \boldsymbol{\xi}_2, \ldots, \boldsymbol{\xi}_r, \boldsymbol{\xi}_{-1}, \boldsymbol{\xi}_{-2}, \ldots, \boldsymbol{\xi}_{-r} \end{bmatrix},$$

then $\mathbf{Y}$ adimits an eigen decomposition

$$\mathbf{Y} = \boldsymbol{\Xi}[\mathrm{diag}(\sigma_1, \sigma_2, \ldots, \sigma_r, -\sigma_1, -\sigma_2, \ldots, -\sigma_r)]\boldsymbol{\Xi}^\top.$$

Let $\widehat{\boldsymbol{\varXi}} = \begin{bmatrix} \widehat{\boldsymbol{\xi}}_s, \ldots, \widehat{\boldsymbol{\xi}}_t \end{bmatrix}$, $\widehat{\boldsymbol{\Sigma}} = \mathrm{diag}(\widehat{\sigma}_s, \ldots, \widehat{\sigma}_t)$ and $\boldsymbol{\varXi} = \begin{bmatrix} \boldsymbol{\xi}_s, \ldots, \boldsymbol{\xi}_t \end{bmatrix}$. Based on the Davis-Kahan $\sin\theta$ theorem [1], we can estimate the difference between the subspaces spaned by the columns of $\widehat{\boldsymbol{\varXi}}$ and $\boldsymbol{\varXi}$. Denote $k = t - s + 1$. Suppose that the singular values of $\boldsymbol{\varXi}^\top \widehat{\boldsymbol{\varXi}}$ are $\sigma_1(\boldsymbol{\varXi}^\top \widehat{\boldsymbol{\varXi}}) \geq \sigma_2(\boldsymbol{\varXi}^\top \widehat{\boldsymbol{\varXi}}) \geq \ldots \geq \sigma_k(\boldsymbol{\varXi}^\top \widehat{\boldsymbol{\varXi}})$. Then we call

$$\Theta(\widehat{\boldsymbol{\varXi}}, \boldsymbol{\varXi}) = \mathrm{diag}\Big( \arccos\big(\sigma_1(\boldsymbol{\varXi}^\top \widehat{\boldsymbol{\varXi}})\big), \ldots, \arccos\big(\sigma_k(\boldsymbol{\varXi}^\top \widehat{\boldsymbol{\varXi}})\big) \Big)$$

the principal angles. According to ([1], Proposition 4.1),

$$\min_{\boldsymbol{\Omega} \in \mathbb{O}^{k \times k}} \big\| \widehat{\boldsymbol{\varXi}} \boldsymbol{\Omega} - \boldsymbol{\varXi} \big\|_2 = 2 \Big\| \sin\Big(\frac{1}{2}\Theta(\widehat{\boldsymbol{\varXi}}, \boldsymbol{\varXi})\Big) \Big\|_2. \tag{B .3}$$

Denote by $\boldsymbol{\Omega}$ the orthogonal matrix that achieves the minimum in (B .3). Since $\sin\big(\frac{\theta}{2}\big) \leq \frac{\sin\theta}{\sqrt{2}}$ for all $\theta \in [0, \frac{\pi}{2}]$, we have

$$\Big\| \sin\Big(\frac{1}{2}\Theta(\widehat{\boldsymbol{\varXi}}, \boldsymbol{\varXi})\Big) \Big\|_2 \leq \frac{\sqrt{2}}{2} \big\| \sin\big(\Theta(\widehat{\boldsymbol{\varXi}}, \boldsymbol{\varXi})\big) \big\|_2.$$

The Davis-Kahan $\sin\theta$ theorem further implies that

$$\big\| \sin\big(\Theta(\widehat{\boldsymbol{\varXi}}, \boldsymbol{\varXi})\big) \big\|_2 \leq \widehat{\Delta}^{-1} \big\| (\widehat{\mathbf{Y}} - \mathbf{Y})\boldsymbol{\varXi} \big\|_2,$$

where $\widehat{\Delta} = \min\{\widehat{\sigma}_{s-1} - \sigma_s, \sigma_t - \widehat{\sigma}_{t+1}\}$. By Weyl's inequality, $\widehat{\Delta} \geq \Delta - \big\| \widehat{\mathbf{Q}} - \mathbf{Q} \big\|_2 \geq \frac{\Delta}{2}$. Hence,

$$\big\| \widehat{\boldsymbol{\varXi}} \boldsymbol{\Omega} - \boldsymbol{\varXi} \big\|_2 \leq 2\sqrt{2}\Delta^{-1} \big\| (\widehat{\mathbf{Y}} - \mathbf{Y})\boldsymbol{\varXi} \big\|_2 \leq 2\sqrt{2}\Delta^{-1} \big\| \widehat{\mathbf{Y}} - \mathbf{Y} \big\|_2. \tag{B .4}$$

We next analyze row-wise errors $\big\| \mathbf{e}_j^\top (\widehat{\boldsymbol{\varXi}} \boldsymbol{\Omega} - \boldsymbol{\varXi}) \big\|_2$ for $j = 1, 2, \ldots, p$. Following the proof idea in [5], Lemma 3.2, we propose a matrix

$$\widetilde{\boldsymbol{\varXi}} = \begin{bmatrix} \widetilde{\boldsymbol{\xi}}_s, \widetilde{\boldsymbol{\xi}}_{s+1}, \ldots, \widetilde{\boldsymbol{\xi}}_k \end{bmatrix} \quad \text{with} \quad \widetilde{\boldsymbol{\xi}}_i = \widehat{\sigma}_i^{-1} \mathbf{Y} \widehat{\boldsymbol{\xi}}_i,$$

and decompose the row-wise error as

$$\begin{aligned}
\big\| \mathbf{e}_j^\top (\widehat{\boldsymbol{\varXi}} \boldsymbol{\Omega} - \boldsymbol{\varXi}) \big\|_2 &\leq \big\| \mathbf{e}_j^\top (\widehat{\boldsymbol{\varXi}} \boldsymbol{\Omega} - \widetilde{\boldsymbol{\varXi}} \boldsymbol{\Omega}) \big\|_2 + \big\| \mathbf{e}_j^\top (\widetilde{\boldsymbol{\varXi}} \boldsymbol{\Omega} - \boldsymbol{\varXi}) \big\|_2 \\
&= \big\| \mathbf{e}_j^\top (\widehat{\boldsymbol{\varXi}} - \widetilde{\boldsymbol{\varXi}}) \big\|_2 + \big\| \mathbf{e}_j^\top (\widetilde{\boldsymbol{\varXi}} \boldsymbol{\Omega} - \boldsymbol{\varXi}) \big\|_2.
\end{aligned} \tag{B .5}$$

By Weyl's inequality, $\min_{s \leq i \leq t} \widehat{\sigma}_i \geq \sigma_t - \big\| \widehat{\mathbf{Q}} - \mathbf{Q} \big\|_2 \geq \frac{\Delta}{2}$, thus $\big\| \widehat{\boldsymbol{\Sigma}}^{-1} \big\|_2 \leq 2\Delta^{-1}$. The first term in (B .5) satisfies

$$\begin{aligned}
\big\| \mathbf{e}_j^\top (\widehat{\boldsymbol{\varXi}} - \widetilde{\boldsymbol{\varXi}}) \big\|_2 &= \big\| \mathbf{e}_j^\top (\widehat{\mathbf{Y}} - \mathbf{Y})\widehat{\boldsymbol{\varXi}} \widehat{\boldsymbol{\Sigma}}^{-1} \big\|_2 \\
&\leq \big\| \mathbf{e}_j^\top (\widehat{\mathbf{Y}} - \mathbf{Y})\widehat{\boldsymbol{\varXi}} \big\|_2 \big\| \widehat{\boldsymbol{\Sigma}}^{-1} \big\|_2 \leq \frac{2}{\Delta} \big\| \mathbf{e}_j^\top (\widehat{\mathbf{Y}} - \mathbf{Y})\widehat{\boldsymbol{\varXi}} \big\|_2,
\end{aligned} \tag{B .6}$$

where we used $\widehat{\boldsymbol{\varXi}} = \widehat{\mathbf{Y}}\widehat{\boldsymbol{\varXi}}\widehat{\boldsymbol{\Sigma}}^{-1}$. By (B .4),

$$\begin{aligned}
\big\| \mathbf{e}_j^\top (\widehat{\mathbf{Y}} - \mathbf{Y})\widehat{\boldsymbol{\varXi}} \big\|_2 &= \big\| \mathbf{e}_j^\top (\widehat{\mathbf{Y}} - \mathbf{Y})\widehat{\boldsymbol{\varXi}} \boldsymbol{\Omega} \big\|_2 \\
&\leq \big\| \mathbf{e}_j^\top (\widehat{\mathbf{Y}} - \mathbf{Y})\boldsymbol{\varXi} \big\|_2 + \big\| \mathbf{e}_j^\top (\widehat{\mathbf{Y}} - \mathbf{Y})(\widehat{\boldsymbol{\varXi}} \boldsymbol{\Omega} - \boldsymbol{\varXi}) \big\|_2 \\
&\leq \big\| \mathbf{e}_j^\top (\widehat{\mathbf{Y}} - \mathbf{Y})\boldsymbol{\varXi} \big\|_2 + \big\| \widehat{\mathbf{Y}} - \mathbf{Y} \big\|_2 \big\| \widehat{\boldsymbol{\varXi}} \boldsymbol{\Omega} - \boldsymbol{\varXi} \big\|_2 \\
&\leq \big\| \mathbf{e}_j^\top (\widehat{\mathbf{Y}} - \mathbf{Y})\boldsymbol{\varXi} \big\|_2 + \frac{2\sqrt{2}}{\Delta} \big\| \widehat{\mathbf{Y}} - \mathbf{Y} \big\|_2^2.
\end{aligned} \tag{B .7}$$

Combining (B .6) and (B .7), we have

$$\left\|\mathbf{e}_j^\top(\widehat{\boldsymbol{\Xi}} - \widetilde{\boldsymbol{\Xi}})\right\|_2 \le \frac{2}{\Delta}\left\|\mathbf{e}_j^\top(\widehat{\mathbf{Y}} - \mathbf{Y})\boldsymbol{\Xi}\right\|_2 + \frac{4\sqrt{2}}{\Delta^2}\left\|\widehat{\mathbf{Y}} - \mathbf{Y}\right\|_2^2. \tag{B .8}$$

Considering the second term in (B .5), we find that

$$\widetilde{\boldsymbol{\Xi}} = \mathbf{Y}\widehat{\boldsymbol{\Xi}}\widehat{\boldsymbol{\Sigma}}^{-1} = (\boldsymbol{\Xi}\boldsymbol{\Xi}^\top\mathbf{Y})\widehat{\boldsymbol{\Xi}}\widehat{\boldsymbol{\Sigma}}^{-1} = \boldsymbol{\Xi}(\boldsymbol{\Xi}^\top\mathbf{Y}\widehat{\boldsymbol{\Xi}}\widehat{\boldsymbol{\Sigma}}^{-1})$$

and $\boldsymbol{\Xi} = \boldsymbol{\Xi}(\boldsymbol{\Xi}^\top\boldsymbol{\Xi})$. Therefore,

$$\begin{aligned}
\left\|\mathbf{e}_j^\top(\widetilde{\boldsymbol{\Xi}}\boldsymbol{\Omega} - \boldsymbol{\Xi})\right\|_2 &= \left\|\mathbf{e}_j^\top\boldsymbol{\Xi}(\boldsymbol{\Xi}^\top\mathbf{Y}\widehat{\boldsymbol{\Xi}}\widehat{\boldsymbol{\Sigma}}^{-1}\boldsymbol{\Omega} - \boldsymbol{\Xi}^\top\boldsymbol{\Xi})\right\|_2 \\
&\le \left\|\mathbf{e}_j^\top\boldsymbol{\Xi}\right\|_2\left\|\boldsymbol{\Xi}^\top(\mathbf{Y}\widehat{\boldsymbol{\Xi}}\widehat{\boldsymbol{\Sigma}}^{-1}\boldsymbol{\Omega} - \boldsymbol{\Xi})\right\|_2 \le \left\|\mathbf{e}_j^\top\boldsymbol{\Xi}\right\|_2\left\|\mathbf{Y}\widehat{\boldsymbol{\Xi}}\widehat{\boldsymbol{\Sigma}}^{-1}\boldsymbol{\Omega} - \boldsymbol{\Xi}\right\|_2,
\end{aligned} \tag{B .9}$$

where

$$\begin{aligned}
\left\|\mathbf{Y}\widehat{\boldsymbol{\Xi}}\widehat{\boldsymbol{\Sigma}}^{-1}\boldsymbol{\Omega} - \boldsymbol{\Xi}\right\|_2 &= \left\|(\widehat{\mathbf{Y}} - (\widehat{\mathbf{Y}} - \mathbf{Y}))\widehat{\boldsymbol{\Xi}}\widehat{\boldsymbol{\Sigma}}^{-1}\boldsymbol{\Omega} - \boldsymbol{\Xi}\right\|_2 \\
&= \left\|\widehat{\boldsymbol{\Xi}}\boldsymbol{\Omega} - (\widehat{\mathbf{Y}} - \mathbf{Y})\widehat{\boldsymbol{\Xi}}\widehat{\boldsymbol{\Sigma}}^{-1}\boldsymbol{\Omega} - \boldsymbol{\Xi}\right\|_2 \le \left\|\widehat{\boldsymbol{\Xi}}\boldsymbol{\Omega} - \boldsymbol{\Xi}\right\|_2 + \left\|(\widehat{\mathbf{Y}} - \mathbf{Y})\widehat{\boldsymbol{\Xi}}\widehat{\boldsymbol{\Sigma}}^{-1}\boldsymbol{\Omega}\right\|_2 \\
&\le \left\|\widehat{\boldsymbol{\Xi}}\boldsymbol{\Omega} - \boldsymbol{\Xi}\right\|_2 + \left\|(\widehat{\mathbf{Y}} - \mathbf{Y})\widehat{\boldsymbol{\Xi}}\right\|_2\left\|\widehat{\boldsymbol{\Sigma}}^{-1}\right\|_2 \overset{(\text{B .4})}{\le} \frac{2(1+\sqrt{2})}{\Delta}\left\|\widehat{\mathbf{Y}} - \mathbf{Y}\right\|_2.
\end{aligned}$$

Plugging (B .8) and (B .9) into (B .5), we have

$$\begin{aligned}
\left\|\mathbf{e}_j^\top(\widehat{\boldsymbol{\Xi}}\boldsymbol{\Omega} - \boldsymbol{\Xi})\right\|_2 \le{}& \frac{2}{\Delta}\left\|\mathbf{e}_j^\top(\widehat{\mathbf{Y}} - \mathbf{Y})\boldsymbol{\Xi}\right\|_2 + \frac{2(1+\sqrt{2})}{\Delta}\left\|\mathbf{e}_j^\top\boldsymbol{\Xi}\right\|_2\left\|\widehat{\mathbf{Y}} - \mathbf{Y}\right\|_2 \\
&+ \frac{4\sqrt{2}}{\Delta^2}\left\|\widehat{\mathbf{Y}} - \mathbf{Y}\right\|_2^2.
\end{aligned} \tag{B .10}$$

Recall the definitions of $\boldsymbol{\Xi}, \widehat{\mathbf{Y}}, \mathbf{Y}, \widehat{\boldsymbol{\Xi}}$ and $\boldsymbol{\Xi}$. For $j = 1, 2, \ldots, p$,

$$\begin{aligned}
\mathbf{e}_{p+j}^\top(\widehat{\boldsymbol{\Xi}}\boldsymbol{\Omega} - \boldsymbol{\Xi}) ={}& \frac{1}{\sqrt{2}}\big[\widehat{\mathbf{h}}_s(j), \widehat{\mathbf{h}}_{s+1}(j), \ldots, \widehat{\mathbf{h}}_t(j)\big]\boldsymbol{\Omega} \\
&- \frac{1}{\sqrt{2}}\big[\mathbf{h}_s(j), \mathbf{h}_{s+1}(j), \ldots, \mathbf{h}_t(j)\big] = \frac{1}{\sqrt{2}}\mathbf{e}_j^\top(\widehat{\boldsymbol{H}}\boldsymbol{\Omega} - \boldsymbol{H}), \\
\mathbf{e}_{p+j}^\top(\widehat{\mathbf{Y}} - \mathbf{Y})\boldsymbol{\Xi} ={}& \frac{1}{\sqrt{2}}\mathbf{e}_j^\top(\widehat{\mathbf{Q}} - \mathbf{Q})^\top\boldsymbol{G}, \text{ where } \boldsymbol{G} = \big[\mathbf{g}_s, \mathbf{g}_{s+1}, \ldots, \mathbf{g}_t\big], \\
&\left\|\widehat{\mathbf{Y}} - \mathbf{Y}\right\|_2 = \left\|\widehat{\mathbf{Q}} - \mathbf{Q}\right\|_2, \\
\left\|\mathbf{e}_{p+j}\boldsymbol{\Xi}\right\|_2 ={}& \left\|\frac{1}{\sqrt{2}}\big[\mathbf{h}_1(j), \ldots, \mathbf{h}_r(j), -\mathbf{h}_1(j), \ldots, -\mathbf{h}_r(j)\big]\right\|_2 = \sqrt{2}\left\|\mathbf{e}_j^\top\mathbf{H}\right\|_2, \\
&\text{where } \mathbf{H} = \big[\mathbf{h}_1, \mathbf{h}_2, \ldots, \mathbf{h}_r\big].
\end{aligned}$$

Hence, (B .10) can be reduced to (B .1).

$\square$

**Lemma B .2** (A deterministic entry-wise perturbation bound for the leading singular vector). *Let* $\delta = \sigma_1 - \sigma_2$. *Suppose that* $\left\|\widehat{\mathbf{Q}} - \mathbf{Q}\right\|_2 \le \frac{\delta}{2K}$, $K \ge 1$, $\delta > 0$. *Then there exists* $\omega \in \{\pm 1\}$ *such that*

$$\begin{aligned}
\left|\omega\widehat{\mathbf{h}}_1(j) - \mathbf{h}_1(j)\right| \le{}& \sum_{k=1}^{\lceil K/2 \rceil}\left(\sigma_1^{-2k}\left|\mathbf{e}_j^\top\mathbf{Q}^\top(\mathbf{Q}\mathbf{Q}^\top)^{k-1}(\widehat{\mathbf{Q}} - \mathbf{Q})\mathbf{h}_1\right|\right. \\
&\left. + \sigma_1^{-2k-1}\left|\mathbf{g}_1^\top(\widehat{\mathbf{Q}} - \mathbf{Q})(\mathbf{Q}^\top\mathbf{Q})^k\mathbf{e}_j\right|\right) \\
&+ |\mathbf{h}_1(j)| \cdot (2K + 1 + 2\sqrt{2})\delta^{-1}\left\|\widehat{\mathbf{Q}} - \mathbf{Q}\right\|_2 \\
&+ 4\sqrt{2}\left(\frac{2\sigma_2}{\sigma_1 + \sigma_2}\right)^K\left\|\mathbf{e}_j^\top\mathbf{H}\right\|_2\delta^{-1}\left\|\widehat{\mathbf{Q}} - \mathbf{Q}\right\|_2 \\
&+ \left(2K^2 + 8\sqrt{2}K\right)\delta^{-2}\left\|\widehat{\mathbf{Q}} - \mathbf{Q}\right\|_2^2.
\end{aligned} \tag{B .11}$$

*Proof.* In the following, we use the same notations $\widehat{\mathbf{Y}}, \mathbf{Y}, \widehat{\boldsymbol{\xi}}_i, \boldsymbol{\xi}_i, \widehat{\boldsymbol{\xi}}_{-i}$ and $\boldsymbol{\xi}_{-i}$ as in the proof of Lemma B .1, and denote $\delta = \sigma_1 - \sigma_2$. Let

$$\boldsymbol{\Sigma}_\perp = \operatorname{diag}(\sigma_2, \sigma_3, \ldots, \sigma_r, -\sigma_2, -\sigma_3, \ldots, -\sigma_r),$$
$$\boldsymbol{\Xi}_\perp = [\boldsymbol{\xi}_2, \boldsymbol{\xi}_3, \ldots, \boldsymbol{\xi}_r, \boldsymbol{\xi}_{-2}, \boldsymbol{\xi}_{-3}, \ldots, \boldsymbol{\xi}_{-r}].$$

Then one has

$$\mathbf{Y} = \sigma_1 \boldsymbol{\xi}_1 \boldsymbol{\xi}_1^\top - \sigma_1 \boldsymbol{\xi}_{-1} \boldsymbol{\xi}_{-1}^\top + \boldsymbol{\Xi}_\perp \boldsymbol{\Sigma}_\perp \boldsymbol{\Xi}_\perp^\top. \tag{B .12}$$

Inspired by power iteration, we decompose the difference $\omega \widehat{\boldsymbol{\xi}}_1 - \boldsymbol{\xi}_1$ by

$$
\begin{aligned}
& \omega \widehat{\boldsymbol{\xi}}_1 - \boldsymbol{\xi}_1 \\
=& \omega (\widehat{\boldsymbol{\xi}}_1 - \mathbf{Y} \widehat{\boldsymbol{\xi}}_1 \widehat{\sigma}_1^{-1}) + (\omega \mathbf{Y} \widehat{\boldsymbol{\xi}}_1 \widehat{\sigma}_1^{-1} - \boldsymbol{\xi}_1) \\
=& \omega (\widehat{\mathbf{Y}} - \mathbf{Y}) \widehat{\boldsymbol{\xi}}_1 \widehat{\sigma}_1^{-1} + (\omega \mathbf{Y} \widehat{\boldsymbol{\xi}}_1 \widehat{\sigma}_1^{-1} - \boldsymbol{\xi}_1) \\
=& \omega (\widehat{\mathbf{Y}} - \mathbf{Y}) \widehat{\boldsymbol{\xi}}_1 \widehat{\sigma}_1^{-1} + \omega (\mathbf{Y} \widehat{\boldsymbol{\xi}}_1 \widehat{\sigma}_1^{-1} - \mathbf{Y}^2 \widehat{\boldsymbol{\xi}}_1 \widehat{\sigma}^{-1}) + (\omega \mathbf{Y}^2 \widehat{\boldsymbol{\xi}}_1 \widehat{\sigma}_1^{-1} - \boldsymbol{\xi}_1) \\
=& \omega (\widehat{\mathbf{Y}} - \mathbf{Y}) \widehat{\boldsymbol{\xi}}_1 \widehat{\sigma}_1^{-1} + \omega \mathbf{Y} (\widehat{\mathbf{Y}} - \mathbf{Y}) \widehat{\boldsymbol{\xi}}_1 \widehat{\sigma}_1^{-2} + (\omega \mathbf{Y}^2 \widehat{\boldsymbol{\xi}}_1 \widehat{\sigma}_1^{-2} - \boldsymbol{\xi}_1) \\
=& \omega \widehat{\sigma}_1^{-1} \sum_{k=0}^{1} (\widehat{\sigma}_1^{-1} \mathbf{Y})^k (\widehat{\mathbf{Y}} - \mathbf{Y}) \widehat{\boldsymbol{\xi}}_1 + (\omega \mathbf{Y}^2 \widehat{\boldsymbol{\xi}}_1 \widehat{\sigma}_1^{-2} - \boldsymbol{\xi}_1) \\
=& \ldots \\
=& \omega \widehat{\sigma}_1^{-1} \sum_{k=0}^{K-1} (\widehat{\sigma}_1^{-1} \mathbf{Y})^k (\widehat{\mathbf{Y}} - \mathbf{Y}) \widehat{\boldsymbol{\xi}}_1 + (\omega \mathbf{Y}^K \widehat{\boldsymbol{\xi}}_1 \widehat{\sigma}_1^{-K} - \boldsymbol{\xi}_1),
\end{aligned}
\tag{B .13}
$$

where we repeatedly used the fact that $\widehat{\mathbf{Y}} \widehat{\boldsymbol{\xi}}_1 \widehat{\sigma}_1^{-1} = \widehat{\boldsymbol{\xi}}_1$.

Plugging (B .12) into the second term in (B .13) and using

$$\boldsymbol{\xi}_{-1}^\top \boldsymbol{\xi}_1 = 0, \qquad \boldsymbol{\Xi}_\perp^\top \boldsymbol{\xi}_1 = \mathbf{0}_{2r-2},$$

one obtains

$$
\begin{aligned}
\omega \mathbf{Y}^K \widehat{\boldsymbol{\xi}}_1 \widehat{\sigma}_1^{-K} - \boldsymbol{\xi}_1 =& \big((\sigma_1/\widehat{\sigma}_1)^K (\omega \boldsymbol{\xi}_1^\top \widehat{\boldsymbol{\xi}}_1) - 1\big)\boldsymbol{\xi}_1 \\
& + (-\sigma_1/\widehat{\sigma}_1)^K \boldsymbol{\xi}_{-1} \boldsymbol{\xi}_{-1}^\top (\omega \widehat{\boldsymbol{\xi}}_1 - \boldsymbol{\xi}_1) \\
& + \boldsymbol{\Xi}_\perp (\widehat{\sigma}_1^{-1} \boldsymbol{\Sigma}_\perp)^K \boldsymbol{\Xi}_\perp^\top (\omega \widehat{\boldsymbol{\xi}}_1 - \boldsymbol{\xi}_1).
\end{aligned}
$$

For a fixed $j = 1, 2, \ldots, 2p$, it follows from (B .13) that

$$
\begin{aligned}
\big|\omega \widehat{\boldsymbol{\xi}}_1(j) - \boldsymbol{\xi}(j)\big| \leq & \sigma_1^{-1} \sum_{k=1}^{K} (\sigma_1/\widehat{\sigma}_1)^k \big|\mathbf{e}_j^\top (\sigma_1^{-1} \mathbf{Y})^{k-1} (\widehat{\mathbf{Y}} - \mathbf{Y}) \widehat{\boldsymbol{\xi}}_1\big| \\
& + |\boldsymbol{\xi}_1(j)| \cdot \big|(\sigma_1/\widehat{\sigma}_1)^K (\omega \boldsymbol{\xi}_1^\top \widehat{\boldsymbol{\xi}}_1) - 1\big| \\
& + (\sigma_1/\widehat{\sigma}_1)^K |\boldsymbol{\xi}_1(j)| \cdot \big\|\omega \widehat{\boldsymbol{\xi}}_1 - \boldsymbol{\xi}_1\big\|_2 \\
& + (\sigma_2/\widehat{\sigma}_1)^K \big\|\mathbf{e}_j^\top \boldsymbol{\Xi}_\perp\big\|_2 \big\|\omega \widehat{\boldsymbol{\xi}}_1 - \boldsymbol{\xi}_1\big\|_2,
\end{aligned}
\tag{B .14}
$$

where we used $\big|\boldsymbol{\xi}_{-1}(j)\big| = \big|\boldsymbol{\xi}_1(j)\big|$.

$\omega \boldsymbol{\xi}_1^\top \widehat{\boldsymbol{\xi}}_1$ and $\big\|\omega \widehat{\boldsymbol{\xi}}_1 - \boldsymbol{\xi}_1\big\|_2$ in (B .14) represent the difference between $\widehat{\boldsymbol{\xi}}_1$ and $\boldsymbol{\xi}_1$. According to the Davis-Kahan $\sin\theta$ theorem,

$$\sin\big(\Theta(\boldsymbol{\xi}_1, \widehat{\boldsymbol{\xi}}_1)\big) \leq \delta^{-1} \big\|\widehat{\mathbf{Y}} - \mathbf{Y}\big\|_2 = \big\|\widehat{\mathbf{Q}} - \mathbf{Q}\big\|_2,$$

where $\Theta(\boldsymbol{\xi}_1, \widehat{\boldsymbol{\xi}}_1)$ denotes the principal angle between $\boldsymbol{\xi}_1$ and $\widehat{\boldsymbol{\xi}}_1$. It follows that there exists $\omega \in \{\pm 1\}$ such that

$$\omega \boldsymbol{\xi}_1^\top \widehat{\boldsymbol{\xi}}_1 = \cos\big(\Theta(\boldsymbol{\xi}_1, \widehat{\boldsymbol{\xi}}_1)\big) = \sqrt{1 - \sin\big(\Theta(\boldsymbol{\xi}_1, \widehat{\boldsymbol{\xi}}_1)\big)} \geq \sqrt{1 - \delta^{-1} \big\|\widehat{\mathbf{Q}} - \mathbf{Q}\big\|_2}. \tag{B .15}$$

Similar as (B .4), we also have

$$\big\|\omega \widehat{\boldsymbol{\xi}}_1 - \boldsymbol{\xi}_1\big\|_2 \leq 2\sqrt{2}\delta^{-1} \big\|\widehat{\mathbf{Q}} - \mathbf{Q}\big\|_2. \tag{B .16}$$

We now focus on the terms involving $\widehat{\sigma}_1$. Note that for all $K \geq 1$ and $x \in \left[0, \frac{1}{2K}\right]$.

$$\left|(1+x)^K - 1\right| \leq 2K|x|, \quad \left|(1+x)^{-K} - 1\right| \leq 2K|x|.$$

By Weyl's inequality, $\left|\widehat{\sigma}_1 - \sigma_1\right| \leq \left\|\widehat{\mathbf{Q}} - \mathbf{Q}\right\|_2$. Therefore, under the condition $\left\|\widehat{\mathbf{Q}} - \mathbf{Q}\right\|_2 \leq \frac{\delta}{2K}$,

$$(\sigma_1/\widehat{\sigma}_1)^k = \left(1 + \frac{\widehat{\sigma}_1 - \sigma_1}{\sigma_1}\right)^{-k} \leq 1 + 2k\delta^{-1}\left\|\widehat{\mathbf{Q}} - \mathbf{Q}\right\|_2 \leq 1 + 2K\delta^{-1}\left\|\widehat{\mathbf{Q}} - \mathbf{Q}\right\|_2,$$

$$\text{for } k = 1, 2, \ldots, K,$$

$$(\sigma_2/\widehat{\sigma}_1)^K \leq \left(\frac{\sigma_2}{\sigma_1 - \|\widehat{\mathbf{Q}} - \mathbf{Q}\|_2}\right)^K \leq \left(\frac{\sigma_2}{\sigma_1 - \delta/2}\right)^K \leq \left(\frac{2\sigma_2}{\sigma_1 + \sigma_2}\right)^K,$$

$$(\sigma_1/\widehat{\sigma}_1)^K \left(\omega\boldsymbol{\xi}_1^\top \widehat{\boldsymbol{\xi}}_1\right) - 1 \leq 2K\delta^{-1}\left\|\widehat{\mathbf{Q}} - \mathbf{Q}\right\|_2,$$

$$(\sigma_1/\widehat{\sigma}_1)^K \left(\omega\boldsymbol{\xi}_1^\top \widehat{\boldsymbol{\xi}}_1\right) - 1 \overset{\text{(B.15)}}{\geq} \left(1 - 2K\delta^{-1}\left\|\widehat{\mathbf{Q}} - \mathbf{Q}\right\|_2\right)\sqrt{1 - \delta^{-1}\left\|\widehat{\mathbf{Q}} - \mathbf{Q}\right\|_2} - 1$$

$$\geq \left(1 - 2K\delta^{-1}\left\|\widehat{\mathbf{Q}} - \mathbf{Q}\right\|_2\right)\left(1 - \delta^{-1}\left\|\widehat{\mathbf{Q}} - \mathbf{Q}\right\|_2\right) - 1 \geq -(2K+1)\delta^{-1}\left\|\widehat{\mathbf{Q}} - \mathbf{Q}\right\|_2.$$

$$\text{(B.17)}$$

Consider the first term in the right hand side of (B.14). For $k = 1, 2, \ldots, K$,

$$\sigma_1^{-1}\left|\mathbf{e}_j^\top \left(\sigma_1^{-1}\mathbf{Y}\right)^{k-1}(\widehat{\mathbf{Y}} - \mathbf{Y})\widehat{\boldsymbol{\xi}}_1\right|$$

$$\leq \sigma_1^{-1}\left|\mathbf{e}_j^\top \left(\sigma_1^{-1}\mathbf{Y}\right)^{k-1}(\widehat{\mathbf{Y}} - \mathbf{Y})\boldsymbol{\xi}_1\right| + \sigma_1^{-1}\left|\mathbf{e}_j^\top \left(\sigma_1^{-1}\mathbf{Y}\right)^{k-1}(\widehat{\mathbf{Y}} - \mathbf{Y})\left(\omega\widehat{\boldsymbol{\xi}}_1 - \boldsymbol{\xi}_1\right)\right|$$

$$\leq \sigma_1^{-1}\left|\mathbf{e}_j^\top \left(\sigma_1^{-1}\mathbf{Y}\right)^{k-1}(\widehat{\mathbf{Y}} - \mathbf{Y})\boldsymbol{\xi}_1\right| + \sigma^{-1}\left\|\widehat{\mathbf{Y}} - \mathbf{Y}\right\|_2\left\|\omega\widehat{\boldsymbol{\xi}}_1 - \boldsymbol{\xi}_1\right\|,$$

and by (B.16) and (B.17),

$$(\sigma_1/\widehat{\sigma}_1)^k \cdot \sigma_1^{-1}\left|\mathbf{e}_j^\top \left(\sigma_1^{-1}\mathbf{Y}\right)^{k-1}(\widehat{\mathbf{Y}} - \mathbf{Y})\widehat{\boldsymbol{\xi}}_1\right|$$

$$\leq \left(1 + 2K\delta^{-1}\left\|\widehat{\mathbf{Q}} - \mathbf{Q}\right\|_2\right) \cdot \sigma_1^{-1}\left(\left|\mathbf{e}_j^\top \left(\sigma_1^{-1}\mathbf{Y}\right)^{k-1}(\widehat{\mathbf{Y}} - \mathbf{Y})\boldsymbol{\xi}_1\right| + 2\sqrt{2}\delta^{-1}\left\|\widehat{\mathbf{Q}} - \mathbf{Q}\right\|_2^2\right)$$

$$\leq \sigma_1^{-1}\left|\mathbf{e}_j^\top \left(\sigma_1^{-1}\mathbf{Y}\right)^{k-1}(\widehat{\mathbf{Y}} - \mathbf{Y})\boldsymbol{\xi}_1\right| + \left(2K + 2\sqrt{2}\right)\delta^{-2}\left\|\widehat{\mathbf{Q}} - \mathbf{Q}\right\|_2^2 + 4\sqrt{2}\delta^{-3}\left\|\widehat{\mathbf{Q}} - \mathbf{Q}\right\|_2^3$$

$$\leq \sigma_1^{-1}\left|\mathbf{e}_j^\top \left(\sigma_1^{-1}\mathbf{Y}\right)^{k-1}(\widehat{\mathbf{Y}} - \mathbf{Y})\boldsymbol{\xi}_1\right| + \left(2K + 4\sqrt{2}\right)\delta^{-2}\left\|\widehat{\mathbf{Q}} - \mathbf{Q}\right\|_2^2,$$

$$\text{(B.18)}$$

where we used $\delta^{-1}\left\|\widehat{\mathbf{Q}} - \mathbf{Q}\right\|_2 \leq \frac{1}{2K}$.

Plugging (B.16), (B.17) and (B.18) into (B.14) gives

$$\left|\omega\widehat{\boldsymbol{\xi}}_1(j) - \boldsymbol{\xi}_1(j)\right|$$

$$\leq \sigma_1^{-1}\sum_{k=1}^{K}\left|\mathbf{e}_j^\top \left(\sigma_1^{-1}\mathbf{Y}\right)^{k-1}(\widehat{\mathbf{Y}} - \mathbf{Y})\boldsymbol{\xi}_1\right| + \left(2K^2 + 4\sqrt{2}K\right)\delta^{-2}\left\|\widehat{\mathbf{Q}} - \mathbf{Q}\right\|_2^2$$

$$+ |\boldsymbol{\xi}_1(j)| \cdot (2K+1)\delta^{-1}\left\|\widehat{\mathbf{Q}} - \mathbf{Q}\right\|_2$$

$$+ 2\sqrt{2}|\boldsymbol{\xi}_1(j)| \cdot \left(1 + 2K\delta^{-1}\left\|\widehat{\mathbf{Q}} - \mathbf{Q}\right\|_2\right)\delta^{-1}\left\|\widehat{\mathbf{Q}} - \mathbf{Q}\right\|_2$$

$$+ 2\sqrt{2}\left(\frac{2\sigma_2}{\sigma_1 + \sigma_2}\right)^K \left\|\mathbf{e}_j^\top \boldsymbol{\Xi}\right\|_2 \delta^{-1}\left\|\widehat{\mathbf{Q}} - \mathbf{Q}\right\|_2,$$

$$= \sigma_1^{-1}\sum_{k=1}^{K}\left|\mathbf{e}_j^\top \left(\sigma_1^{-1}\mathbf{Y}\right)^{k-1}(\widehat{\mathbf{Y}} - \mathbf{Y})\boldsymbol{\xi}_1\right| + |\boldsymbol{\xi}_1(j)| \cdot (2K + 1 + 2\sqrt{2})\delta^{-1}\left\|\widehat{\mathbf{Q}} - \mathbf{Q}\right\|_2$$

$$+ 2\sqrt{2}\left(\frac{2\sigma_2}{\sigma_1 + \sigma_2}\right)^K \left\|\mathbf{e}_j^\top \boldsymbol{\Xi}\right\|_2 \delta^{-1}\left\|\widehat{\mathbf{Q}} - \mathbf{Q}\right\|_2 + \left(2K^2 + 8\sqrt{2}K\right)\delta^{-2}\left\|\widehat{\mathbf{Q}} - \mathbf{Q}\right\|_2^2.$$

$$\text{(B.19)}$$

For $k = 0, 1, \ldots, K-1$, let

$$\left(\sigma_1^{-1}\mathbf{Y}\right)^k \mathbf{e}_{p+j} = \left[\begin{array}{c} \boldsymbol{v}_{1,k} \\ \boldsymbol{v}_{2,k} \end{array}\right], \quad \boldsymbol{v}_{1,k}, \boldsymbol{v}_{2,k} \in \mathbb{R}^p.$$

Since

$$\mathbf{Y}^k = \begin{cases} \begin{bmatrix} \mathbf{0} & \mathbf{Q}(\mathbf{Q}^\top\mathbf{Q})^{\frac{k-1}{2}} \\ \mathbf{Q}^\top(\mathbf{Q}\mathbf{Q}^\top)^{\frac{k-1}{2}} & \mathbf{0} \end{bmatrix}, \\ \qquad\qquad \text{if } k \text{ is odd,} \\[1em] \begin{bmatrix} (\mathbf{Q}\mathbf{Q}^\top)^{\frac{k}{2}} & \mathbf{0} \\ \mathbf{0} & (\mathbf{Q}^\top\mathbf{Q})^{\frac{k}{2}} \end{bmatrix}, \\ \qquad\qquad \text{if } k \text{ is even,} \end{cases}$$

$$\boldsymbol{v}_{1,k} = \sigma_1^{-k}\mathbf{Q}(\mathbf{Q}^\top\mathbf{Q})^{\frac{k-1}{2}}\mathbf{e}_j, \ \boldsymbol{v}_{2,k} = \mathbf{0}, \quad \text{when } k \text{ is odd,}$$

$$\boldsymbol{v}_{1,k} = \mathbf{0}, \ \boldsymbol{v}_{2,k} = \sigma_1^{-k}(\mathbf{Q}^\top\mathbf{Q})^{\frac{k}{2}}\mathbf{e}_j, \qquad \text{when } k \text{ is even.}$$

Recall the definitions of $\widehat{\mathbf{Y}}$, $\mathbf{Y}$, $\boldsymbol{\Xi}$ and $\boldsymbol{\xi}_1$. For $j = 1, 2, \ldots, p$, we reduce (B .19) into (B .11). $\qquad\square$

## B .2 Markov Chain Concentration Inequalities

**Lemma B .3** (Lemma 7 in [8], Markov chain concentration inequality). *Suppose $\mathbf{P} \in \mathbb{R}^{p\times p}$ is an ergodic Markov chain transition matrix on $p$ states $\{1, \cdots, p\}$. $\mathbf{P}$ is with invariant distribution $\boldsymbol{\pi}$ and the Markov mixing time*

$$\tau(\varepsilon) = \min\left\{k : \max_{1\le i\le p} \frac{1}{2}\|\mathbf{e}_i^\top\mathbf{P}^k - \boldsymbol{\pi}^\top\|_1 \le \varepsilon\right\}.$$

*Recall the frequency matrix is $\mathbf{F} = [\mathrm{diag}(\boldsymbol{\pi})]\mathbf{P}$. Give a Markov trajectory with $(n+1)$ observable states $X = \{X_0, X_1, \cdots, X_n\}$ from any initial state. Let $\tau_* = \tau(1/4)$. For any constant $c_0 > 0$, there exists a constant $C > 0$ such that if $n \ge C\tau_* p\log^2(n)$, then*

$$\mathbb{P}\left(\|n^{-1}\mathbf{N} - \mathbf{F}\|_2 \ge Cn^{-1/2}\sqrt{\pi_{\max}\tau_*\log^2(n)}\right) \le n^{-c_0}, \tag{B .20}$$

$$\mathbb{P}\left(\|n^{-1}\mathbf{m} - \boldsymbol{\pi}\|_\infty \ge Cn^{-1/2}\sqrt{\pi_{\max}\tau_*\log^2(n)}\right) \le n^{-c_0}. \tag{B .21}$$

**Lemma B .4** (Row-wise Markov chain concentration inequality). *Suppose $\boldsymbol{\eta} \in \mathbb{R}^p$ is a fixed unit vector and $\mathbf{K} \in \mathbb{O}^{p\times s}$ is a fixed orthonormal matrix satisfying*

$$\max_{1\le i\le p}\|\mathbf{e}_i^\top\mathbf{K}\|_2 \le \gamma\sqrt{s/p}.$$

*Under the same setting as in Lemma B .3, for any $c_0 > 0$, there exists $C > 0$ such that if $n \ge Cp\tau_*\log^2(n)$, then*

$$\begin{aligned} \mathbb{P}\left(\|\boldsymbol{\eta}^\top(n^{-1}\widehat{\mathbf{N}} - \mathbf{F})[\mathrm{diag}(\boldsymbol{\pi})]^{-\frac{1}{2}}\mathbf{K}\|_2 \ge Cn^{-1/2}\sqrt{\tau_*(\gamma^2 s/p)\log^2(n)}\right) \le n^{-c_0}, \\ \mathbb{P}\left(\|\mathbf{K}^\top(n^{-1}\mathbf{N} - \mathbf{F})[\mathrm{diag}(\boldsymbol{\pi})]^{-\frac{1}{2}}\boldsymbol{\eta}\|_2 \ge Cn^{-1/2}\sqrt{\tau_*(\gamma^2 s/p)\log^2(n)}\right) \le n^{-c_0}. \end{aligned} \tag{B .22}$$

*Proof.* We follow the proof of Lemma 8 in [8]. For any $t > 0$, let

$$\alpha = \tau\left(\pi_{\min} \wedge \frac{\sqrt{c_1}t}{2\gamma\sqrt{s}}\right) + 1,$$

and $n_0 = \lfloor n/\alpha\rfloor$. Without loss of generality, assume $n$ is a multiple of $\alpha$. By definition,

$$n^{-1}\mathbf{K}^\top\mathbf{N}[\mathrm{diag}(\boldsymbol{\pi})]^{-\frac{1}{2}}\boldsymbol{\eta} = \frac{1}{n}\sum_{l=1}^{\alpha}\sum_{k=1}^{n_0}\mathbf{T}_k^{(l)},$$

$$\mathbf{T}_k^{(l)} = \mathbf{K}^\top\mathbf{e}_{X_{k\alpha+l-1}}\mathbf{e}_{X_{k\alpha+l}}^\top[\mathrm{diag}(\boldsymbol{\pi})]^{-\frac{1}{2}}\boldsymbol{\eta},$$

and

$$n^{-1}\boldsymbol{\eta}^\top\mathbf{N}[\mathrm{diag}(\boldsymbol{\pi})]^{-\frac{1}{2}}\mathbf{K} = \frac{1}{n}\sum_{l=1}^{\alpha}\sum_{k=1}^{n_0}\mathbf{S}_k^{(l)},$$

$$\mathbf{S}_k^{(l)} = \boldsymbol{\eta}^\top \mathbf{e}_{X_{k\alpha+l-1}} \mathbf{e}_{X_{k\alpha+l}}^\top [\mathrm{diag}(\boldsymbol{\pi})]^{-\frac{1}{2}} \mathbf{K}.$$

We introduce the "thin" sequences as

$$\widetilde{\mathbf{T}}_k^{(l)} = \mathbf{T}_k^{(l)} - \mathbb{E}\big[\mathbf{T}_k^{(l)}\big|X_{(k-1)\alpha+l}\big], \qquad \widetilde{\mathbf{S}}_k^{(l)} = \mathbf{S}_k^{(l)} - \mathbb{E}\big[\mathbf{S}_k^{(l)}\big|X_{(k-1)\alpha+l}\big],$$

for $l = 1, \ldots, \alpha$, $k = 1, \ldots, n_0$, and apply the matrix Freedman's inequality [7] to derive concentration inequalities of partial sum sequences $\sum_{k=1}^{n_0} \widetilde{\mathbf{T}}_k^{(l)}$ and $\sum_{k=1}^{n_0} \widetilde{\mathbf{S}}_k^{(l)}$ for a fixed $l$.

Consider the predictable quadratic variation processes of the martingales $\left\{\sum_{k=1}^t \widetilde{\mathbf{T}}_k^{(l)}\right\}_{t=1}^{n_0}$ and $\left\{\sum_{k=1}^t \widetilde{\mathbf{S}}_k^{(l)}\right\}_{t=1}^{n_0}$:

$$\mathbf{X}_{1,t}^{(l)} = \sum_{k=1}^t \mathbb{E}\Big[\widetilde{\mathbf{T}}_k^{(l)}\big(\widetilde{\mathbf{T}}_k^{(l)}\big)^\top \Big| X_{(k-1)\alpha+l}\Big], \quad \mathbf{X}_{2,t}^{(l)} = \sum_{k=1}^t \mathbb{E}\Big[\big(\widetilde{\mathbf{T}}_k^{(l)}\big)^\top \widetilde{\mathbf{T}}_k^{(l)} \Big| X_{(k-1)\alpha+l}\Big],$$

and

$$\mathbf{Z}_{1,t}^{(l)} = \sum_{k=1}^t \mathbb{E}\Big[\widetilde{\mathbf{S}}_k^{(l)}\big(\widetilde{\mathbf{S}}_k^{(l)}\big)^\top \Big| X_{(k-1)\alpha+l}\Big], \quad \mathbf{Z}_{2,t}^{(l)} = \sum_{k=1}^t \mathbb{E}\Big[\big(\widetilde{\mathbf{S}}_k^{(l)}\big)^\top \widetilde{\mathbf{S}}_k^{(l)} \Big| X_{(k-1)\alpha+l}\Big],$$

for $t = 1, 2, \ldots, n_0$. Since

$$\big\|\mathbf{X}_{1,t}^{(l)}\big\|_2 \le \sum_{k=1}^t \Big\|\mathbb{E}\Big[\widetilde{\mathbf{T}}_k^{(l)}\big(\widetilde{\mathbf{T}}_k^{(l)}\big)^\top \Big| X_{(t-1)\alpha+l}\Big]\Big\|_2 \le \sum_{k=1}^t \mathbb{E}\Big[\Big\|\widetilde{\mathbf{T}}_k^{(l)}\big(\widetilde{\mathbf{T}}_k^{(l)}\big)^\top\Big\|_2 \Big| X_{(k-1)\alpha+l}\Big]$$

$$= \sum_{k=1}^t \mathbb{E}\Big[\big\|\widetilde{\mathbf{T}}_k^{(l)}\big\|_2^2 \Big| X_{(k-1)\alpha+l}\Big] = \mathbf{X}_{2,t}^{(l)},$$

and $\big\|\mathbf{Z}_{2,t}^{(l)}\big\|_2 \le \mathbf{Z}_{1,t}^{(l)}$ for the same reason, we only need to focus on $\mathbf{X}_{2,t}^{(l)}$ and $\mathbf{Z}_{1,t}^{(l)}$.

Denote $\widetilde{\boldsymbol{\pi}}^{(k,l)} \equiv (\mathbf{P}^{\alpha-1})^\top \mathbf{e}_{X_{(k-1)\alpha+l-1}}$. By the definition of mixing time $\alpha$,

$$\max_{1 \le s \le p} \frac{1}{2}\big\|\mathbf{e}_s^\top \mathbf{P}^{\alpha-1} - \boldsymbol{\pi}^\top\big\|_1 \le \pi_{\min} \wedge \frac{\sqrt{c_1 t}}{2\gamma\sqrt{s}}.$$

Hence, $\widetilde{\pi}_i^{(k,l)} \le \pi_i + \pi_{\min} \le 2\pi_i$ for $i = 1, \ldots, p$. We have

$$\mathbb{E}\Big[\big(\widetilde{\mathbf{T}}_k^{(l)}\big)^\top \widetilde{\mathbf{T}}_k^{(l)} \Big| X_{(k-1)\alpha+l}\Big] \le \mathbb{E}\Big[\big\|\mathbf{T}_k^{(l)}\big\|_2^2 \Big| X_{(k-1)\alpha+l}\Big]$$

$$= \mathbb{E}\Big[\big\|\mathbf{e}_{X_{k\alpha+l-1}}^\top \mathbf{K}\big\|_2^2 \big(\mathbf{e}_{X_{k\alpha+l}}^\top [\mathrm{diag}(\boldsymbol{\pi})]^{-\frac{1}{2}}\boldsymbol{\eta}\big)^2 \Big| X_{(k-1)\alpha+l}\Big]$$

$$= \sum_{i=1}^p \sum_{j=1}^p \mathbb{P}\big[X_{k\alpha+l-1} = i\big|X_{(k-1)\alpha+l}\big] \cdot \mathbb{P}\big[X_{k\alpha+l} = j\big|X_{k\alpha+l-1} = i\big] \big\|\mathbf{e}_i^\top \mathbf{K}\big\|_2^2 \pi_j^{-1}\eta_j^2$$

$$= \sum_{i=1}^p \sum_{j=1}^p \widetilde{\pi}_i^{(k,l)} P_{ij} \big\|\mathbf{e}_i^\top \mathbf{K}\big\|_2^2 \pi_j^{-1}\eta_j^2 \le \sum_{j=1}^p 2\Big(\sum_{i=1}^p \pi_i P_{ij}\Big)\pi_j^{-1}\eta_j^2 \cdot \gamma^2 sp^{-1}$$

$$= \sum_{j=1}^p 2\eta_j^2 \cdot \gamma^2 sp^{-1} = 2\gamma^2 sp^{-1}, \quad \text{for } k = 1, \ldots, n_0,$$

where we used $\boldsymbol{\pi}^\top \mathbf{P} = \boldsymbol{\pi}^\top$, $\sum_{i=1}^p \pi_i P_{ij} = \pi_j$. Similarly,

$$\mathbb{E}\Big[\widetilde{\mathbf{S}}_k^{(l)}\big(\widetilde{\mathbf{S}}_k^{(l)}\big)^\top\Big|X_{(k-1)\alpha+l}\Big] \leq \mathbb{E}\Big[\big\|\mathbf{S}_k^{(l)}\big\|_2^2\Big|X_{(k-1)\alpha+l}\Big]$$

$$=\mathbb{E}\Big[\big(\boldsymbol{\eta}^\top \mathbf{e}_{X_{k\alpha+l-1}}\big)^2 \big\|\mathbf{e}_{X_{k\alpha+l}}^\top [\mathrm{diag}(\boldsymbol{\pi})]^{-\frac{1}{2}}\mathbf{K}\big\|_2^2\Big|X_{(k-1)\alpha+l}\Big]$$

$$=\sum_{i=1}^p \sum_{j=1}^p \mathbb{P}\big[X_{k\alpha+l-1}=i\big|X_{(k-1)\alpha+l}\big]\cdot \mathbb{P}\big[X_{k\alpha+l}=j\big|X_{k\alpha+l-1}=i\big]\big\|\mathbf{e}_j^\top\mathbf{K}\big\|_2^2 \pi_j^{-1}\eta_i^2$$

$$=\sum_{i=1}^p \sum_{j=1}^p \widetilde{\pi}_i^{(k,l)} P_{ij}\big\|\mathbf{e}_j^\top\mathbf{K}\big\|_2^2 \pi_j^{-1}\eta_i^2 \leq \sum_{i=1}^p \sum_{j=1}^p 2(\pi_i/\pi_j)P_{ij}\eta_i^2 \cdot \gamma^2 sp^{-1}$$

$$\leq 2C_1 c_1^{-1}\sum_{i=1}^p \eta_i^2 \sum_{j=1}^p P_{ij}\cdot \gamma^2 sp^{-1} = 2C_1 c_1^{-1}\gamma^2 sp^{-1}, \qquad \text{for } k=1,\ldots,n_0.$$

Note that

$$\big\|\mathbf{T}_k^{(l)}\big\|_2 \leq \max_{1\leq i\leq p}\pi_{\min}^{-\frac{1}{2}}\big\|\mathbf{e}_i^\top\mathbf{K}\big\|_2 \leq \gamma\sqrt{s/(\pi_{\min}p)} \leq c_1^{-\frac{1}{2}}\gamma\sqrt{s},$$

$$\big\|\mathbb{E}\big[\mathbf{T}_k^{(l)}\big|X_{(k-1)\alpha+l}\big]\big\|_2 \leq \mathbb{E}\big[\big\|\mathbf{T}_k^{(l)}\big\|_2\big|X_{(k-1)\alpha+l}\big] \leq c_1^{-\frac{1}{2}}\gamma\sqrt{s}.$$

Therefore, $\big\|\widetilde{\mathbf{T}}_k^{(l)}\big\|_2 \leq 2c_1^{-\frac{1}{2}}\gamma\sqrt{s}$. Similarly, we have $\big\|\widetilde{\mathbf{S}}_k^{(l)}\big\|_2 \leq 2c_1^{-\frac{1}{2}}\gamma\sqrt{s}$. The matrix Freedman's inequality [7] implies

$$\mathbb{P}\left(\Big\|n_0^{-1}\sum_{k=1}^{n_0}\widetilde{\mathbf{T}}_k^{(l)}\Big\|_2 \geq \frac{t}{2}\right) \leq (s+1)\exp\left(-\frac{n_0^2 t^2/8}{2n_0\gamma^2 sp^{-1}+n_0 c_1^{-\frac{1}{2}}\gamma t\sqrt{s}/3}\right)$$

and

$$\mathbb{P}\left(\Big\|n_0^{-1}\sum_{k=1}^{n_0}\widetilde{\mathbf{S}}_k^{(l)}\Big\|_2 \geq \frac{t}{2}\right) \leq (s+1)\exp\left(-\frac{n_0^2 t^2/8}{2n_0 C_1 c_1^{-1}\gamma^2 sp^{-1}+n_0 c_1^{-\frac{1}{2}}\gamma t\sqrt{s}/3}\right).$$

By union bound,

$$\mathbb{P}\left(\Big\|n^{-1}\mathbf{K}^\top\mathbf{N}[\mathrm{diag}(\boldsymbol{\pi})]^{-\frac{1}{2}}\boldsymbol{\eta}-n^{-1}\sum_{l=1}^\alpha\sum_{k=1}^{n_0}\mathbb{E}\big[\mathbf{T}_k^{(l)}\big|X_{(k-1)\alpha+l}\big]\Big\|_2 \geq \frac{t}{2}\right)$$

$$=\mathbb{P}\left(\Big\|n^{-1}\sum_{l=1}^\alpha\sum_{k=1}^{n_0}\widetilde{\mathbf{T}}_k^{(l)}\Big\|_2 \geq \frac{t}{2}\right) \leq \sum_{l=1}^\alpha \mathbb{P}\left(\Big\|n_0^{-1}\sum_{k=1}^{n_0}\widetilde{\mathbf{T}}_k^{(l)}\Big\|_2 \geq \frac{t}{2}\right) \qquad\text{(B.23)}$$

$$\leq \alpha(s+1)\exp\left(-\frac{n_0 t^2/8}{2\gamma^2 sp^{-1}+c_1^{-\frac{1}{2}}\gamma t\sqrt{s}/3}\right).$$

Similarly,

$$\mathbb{P}\left(\Big\|n^{-1}\boldsymbol{\eta}^\top\mathbf{N}[\mathrm{diag}(\boldsymbol{\pi})]^{-\frac{1}{2}}\mathbf{K}-n^{-1}\sum_{l=1}^\alpha\sum_{k=1}^{n_0}\mathbb{E}\big[\mathbf{S}_k^{(l)}\big|X_{(k-1)\alpha+l}\big]\Big\|_2 \geq \frac{t}{2}\right)$$

$$\qquad\qquad\qquad\qquad\qquad\qquad\qquad\qquad\qquad\qquad\qquad\qquad\text{(B.24)}$$

$$\leq \alpha(s+1)\exp\left(-\frac{n_0 t^2/8}{2C_1 c_1^{-1}\gamma^2 sp^{-1}+c_1^{-\frac{1}{2}}\gamma t\sqrt{s}/3}\right).$$

We next analyze the differences

$$n^{-1}\sum_{l=1}^\alpha\sum_{k=1}^{n_0}\mathbb{E}\big[\mathbf{T}_k^{(l)}\big|X_{(k-1)\alpha+l}\big] - \mathbf{K}^\top\mathbf{F}[\mathrm{diag}(\boldsymbol{\pi})]^{-\frac{1}{2}}\boldsymbol{\eta}$$

and

$$n^{-1}\sum_{l=1}^\alpha\sum_{k=1}^{n_0}\mathbb{E}\big[\mathbf{S}_k^{(l)}\big|X_{(k-1)\alpha+l}\big] - \boldsymbol{\eta}^\top\mathbf{F}[\mathrm{diag}(\boldsymbol{\pi})]^{-\frac{1}{2}}\mathbf{K}.$$

Since

$$\mathbb{E}\big[\mathbf{T}_k^{(l)}\big|X_{(k-1)\alpha+l}\big] = \sum_{i=1}^{p}\sum_{j=1}^{p}\widetilde{\pi}_i^{(k,l)}P_{ij}\mathbf{K}^\top\mathbf{e}_i\mathbf{e}_j^\top[\mathrm{diag}(\boldsymbol{\pi})]^{-\frac{1}{2}}\boldsymbol{\eta}$$

$$= \sum_{i=1}^{p}\sum_{j=1}^{p}\widetilde{\pi}_i^{(k,l)}P_{ij}\mathbf{K}^\top\mathbf{e}_i\pi_j^{-\frac{1}{2}}\eta_j,$$

$$\mathbf{K}^\top\mathbf{F}[\mathrm{diag}(\boldsymbol{\pi})]^{-\frac{1}{2}}\boldsymbol{\eta} = \sum_{i=1}^{p}\sum_{j=1}^{p}\mathbf{K}^\top\big(\pi_i\mathbf{e}_i\mathbf{e}_i^\top\big)\mathbf{P}(\mathbf{e}_j\mathbf{e}_j^\top)[\mathrm{diag}(\boldsymbol{\pi})]^{-\frac{1}{2}}\boldsymbol{\eta}$$

$$= \sum_{i=1}^{p}\sum_{j=1}^{p}\pi_iP_{ij}\mathbf{K}^\top\mathbf{e}_i\pi_j^{-\frac{1}{2}}\eta_j,$$

it follows that

$$\Big\|\mathbb{E}\big[\mathbf{T}_k^{(l)}\big|X_{(k-1)\alpha+l}\big] - \mathbf{K}^\top\mathbf{F}[\mathrm{diag}(\boldsymbol{\pi})]^{-\frac{1}{2}}\boldsymbol{\eta}\Big\|_2$$

$$= \Big\|\sum_{i=1}^{p}\sum_{j=1}^{p}(\widetilde{\pi}_i^{(k,l)} - \pi_i)P_{ij}\mathbf{K}^\top\mathbf{e}_i\pi_j^{-\frac{1}{2}}\eta_j\Big\|_2$$

$$\leq \sum_{i=1}^{p}\sum_{j=1}^{p}\big|\widetilde{\pi}_i^{(k,l)} - \pi_i\big|P_{ij}\pi_j^{-\frac{1}{2}}\eta_j\big\|\mathbf{e}_i^\top\mathbf{K}\big\|_2$$

$$\leq \sum_{i=1}^{p}\big|\widetilde{\pi}_i^{(k,l)} - \pi_i\big|\sum_{j=1}^{p}P_{ij}\pi_{\min}^{-\frac{1}{2}}\cdot\gamma\sqrt{s/p}$$

$$\leq \big\|\widetilde{\boldsymbol{\pi}}^{(k,l)} - \boldsymbol{\pi}\big\|_1\cdot c_1^{-\frac{1}{2}}\gamma\sqrt{s} \leq \frac{t}{2}.$$

Therefore,

$$\Big\|n^{-1}\sum_{l=1}^{\alpha}\sum_{k=1}^{n_0}\mathbb{E}\big[\mathbf{T}_k^{(l)}\big|X_{(k-1)\alpha+l}\big] - \mathbf{K}^\top\mathbf{F}[\mathrm{diag}(\boldsymbol{\pi})]^{-\frac{1}{2}}\boldsymbol{\eta}\Big\|_2$$

$$\leq n^{-1}\sum_{l=1}^{\alpha}\sum_{k=1}^{n_0}\Big\|\mathbb{E}\big[\mathbf{T}_k^{(l)}\big|X_{(k-1)\alpha+l}\big] - \mathbf{K}^\top\mathbf{F}[\mathrm{diag}(\boldsymbol{\pi})]^{-\frac{1}{2}}\boldsymbol{\eta}\Big\|_2 \leq \frac{t}{2}, \tag{B.25}$$

Symmetrically, we have

$$\Big\|n^{-1}\sum_{l=1}^{\alpha}\sum_{k=1}^{n_0}\mathbb{E}\big[\mathbf{S}_k^{(l)}\big|X_{(k-1)\alpha+l}\big] - \boldsymbol{\eta}^\top\mathbf{F}[\mathrm{diag}(\boldsymbol{\pi})]^{-\frac{1}{2}}\mathbf{K}\Big\|_2 \leq \frac{t}{2}. \tag{B.26}$$

Combining (B.23) and (B.25), (B.24) and (B.26) yields

$$\mathbb{P}\Big(\big\|n^{-1}\mathbf{K}^\top\mathbf{N}[\mathrm{diag}(\boldsymbol{\pi})]^{-\frac{1}{2}}\boldsymbol{\eta} - \mathbf{K}^\top\mathbf{F}[\mathrm{diag}(\boldsymbol{\pi})]^{-\frac{1}{2}}\boldsymbol{\eta}\big\|_2 \geq t\Big)$$

$$\leq \alpha(s+1)\exp\left(-\frac{nt^2/(8\alpha)}{2\gamma^2sp^{-1} + c_1^{-\frac{1}{2}}\gamma t\sqrt{s}/3}\right),$$

and

$$\mathbb{P}\Big(\big\|n^{-1}\boldsymbol{\eta}^\top\mathbf{N}[\mathrm{diag}(\boldsymbol{\pi})]^{-\frac{1}{2}}\mathbf{K} - \boldsymbol{\eta}^\top\mathbf{F}[\mathrm{diag}(\boldsymbol{\pi})]^{-\frac{1}{2}}\mathbf{K}\big\|_2 \geq t\Big)$$

$$\leq \alpha(s+1)\exp\left(-\frac{nt^2/(8\alpha)}{2C_1c_1^{-1}\gamma^2sp^{-1} + c_1^{-\frac{1}{2}}\gamma t\sqrt{s}/3}\right).$$

For a fixed $c_0 > 0$, there exists a constant $C', C'' > 0$ such that, by taking

$$t = C'n^{-\frac{1}{2}}\sqrt{\gamma^2sp^{-1}\alpha\log(n)} + C'n^{-1}\sqrt{s}\alpha\log(n),$$

$$n \geq C''\big(\alpha p\log(n) \vee \alpha s\big),$$

one has

$$\mathbb{P}\Big(\big\|n^{-1}\mathbf{K}^{\top}\mathbf{N}[\mathrm{diag}(\boldsymbol{\pi})]^{-\frac{1}{2}}\boldsymbol{\eta} - \mathbf{K}^{\top}\mathbf{F}[\mathrm{diag}(\boldsymbol{\pi})]^{-\frac{1}{2}}\boldsymbol{\eta}\big\|_2$$

$$\geq C'\big(1 + (C'')^{-\frac{1}{2}}\big)n^{-\frac{1}{2}}\sqrt{\gamma^2 s p^{-1}\alpha \log(n)}\Big) \leq n^{-c_0}.$$

According to Lemma 5 in [8],

$$\alpha \leq -\tau_* \log_2\Big(\frac{\pi_{\min}}{2} \wedge \frac{\sqrt{c_1}t}{4\gamma\sqrt{s}}\Big),$$

where $\tau_* = \tau(1/4)$. Fix $C'$, when $C''$ is sufficiently large,

$$n^{-2} \leq \frac{\sqrt{c_1}}{4\gamma}C'n^{-1} \leq \frac{\sqrt{c_1}t}{4\gamma\sqrt{s}} \leq \frac{\pi_{\min}}{2}.$$

In this case, $\alpha \leq 2\tau_* \log_2(n)$. Therefore, there exists a constant $C > 0$ such that when $n \geq Cp\tau_* \log^2(n)$,

$$\mathbb{P}\Big(\big\|n^{-1}\mathbf{K}^{\top}\mathbf{N}[\mathrm{diag}(\boldsymbol{\pi})]^{-\frac{1}{2}}\boldsymbol{\eta} - \mathbf{K}^{\top}\mathbf{F}[\mathrm{diag}(\boldsymbol{\pi})]^{-\frac{1}{2}}\boldsymbol{\eta}\big\|_2$$

$$\leq Cn^{-\frac{1}{2}}\sqrt{\gamma^2 s p^{-1}\tau_* \log^2(n)}\Big) \leq n^{-c_0}. \tag{B.27}$$

Following the same analysis, one also has

$$\mathbb{P}\Big(\big\|n^{-1}\boldsymbol{\eta}^{\top}\mathbf{N}[\mathrm{diag}(\boldsymbol{\pi})]^{-\frac{1}{2}}\mathbf{K} - \boldsymbol{\eta}^{\top}\mathbf{F}[\mathrm{diag}(\boldsymbol{\pi})]^{-\frac{1}{2}}\mathbf{K}\big\|_2$$

$$\leq \bar{C}n^{-\frac{1}{2}}\sqrt{\gamma^2 s p^{-1}\tau_* \log^2(n)}\Big) \leq n^{-c_0}, \tag{B.28}$$

when $n \geq \bar{C}p\tau_* \log^2(n)$ for some sufficiently large $\bar{C} > 0$. $\qquad\square$

## B.3  Adapting Markov Chain Concentration Inequalities to Our Settings

**Corollary B.1.** *Under assumption (A.1), for any $c_0 > 0$, there exists a constant $C > 0$, such that if $n \geq C\tau_* p \log^2(n)$, then*

$$\mathbb{P}\Big(\big\|\widehat{\mathbf{Q}} - \mathbf{Q}\big\|_2 \geq Cn^{-1/2}\sqrt{\tau_* \log^2(n)}\Big) \leq n^{-c_0}. \tag{B.29}$$

*Proof.* Based on Lemma B.3 and the union bound, we know that for any fixed $c_0 > 0$, there exists a constant $C_0 > 0$ such that with probability at least $1 - 2n^{-c_0}$,

$$\big\|n^{-1}\mathbf{N} - \mathbf{F}\big\|_2 \leq C_0 n^{-1/2}\sqrt{\pi_{\max}\tau_* \log^2(n)}$$

$$\text{and } \big\|n^{-1}\mathbf{m} - \boldsymbol{\pi}\big\|_\infty \leq C_0 n^{-1/2}\sqrt{\pi_{\max}\tau_* \log^2(n)} \tag{B.30}$$

hold simultaneously for $n \geq C_0\tau_* p \log^2(n)$.

We find that

$$\big\|\widehat{\mathbf{Q}} - \mathbf{Q}\big\|_2 = \big\|(n^{-1}\mathbf{N})[\mathrm{diag}(n^{-1}\mathbf{m})]^{-\frac{1}{2}} - \mathbf{F}[\mathrm{diag}(\boldsymbol{\pi})]^{-\frac{1}{2}}\big\|_2$$

$$\leq \big\|(n^{-1}\mathbf{N} - \mathbf{F})[\mathrm{diag}(\boldsymbol{\pi})]^{-\frac{1}{2}}\big\|_2 + \big\|\big((n^{-1}\mathbf{N})[\mathrm{diag}(n^{-1}\mathbf{m})]^{-\frac{1}{2}} - [\mathrm{diag}(\boldsymbol{\pi})]^{-\frac{1}{2}}\big)\big\|_2. \tag{B.31}$$

Under condition (B.30), the first term in (B.31) satisfies

$$\big\|(n^{-1}\mathbf{N} - \mathbf{F})[\mathrm{diag}(\boldsymbol{\pi})]^{-\frac{1}{2}}\big\|_2 \leq \pi_{\min}^{-\frac{1}{2}}\big\|n^{-1}\mathbf{N} - \mathbf{F}\big\|_2$$

$$\leq C_0 n^{-1/2}\sqrt{\frac{\pi_{\max}}{\pi_{\min}}\tau_* \log^2(n)} \leq C_0 C_1^{\frac{1}{2}}c_1^{-\frac{1}{2}}n^{-1/2}\sqrt{\tau_* \log^2(n)}. \tag{B.32}$$

We decompose the second term in (B.31) into

$$\big\|(n^{-1}\mathbf{N})\big([\mathrm{diag}(n^{-1}\mathbf{m})]^{-\frac{1}{2}} - [\mathrm{diag}(\boldsymbol{\pi})]^{-\frac{1}{2}}\big)\big\|_2$$

$$\leq \big\|n^{-1}\mathbf{N}\big\|\big\|[\mathrm{diag}(n^{-1}\mathbf{m})]^{-\frac{1}{2}} - [\mathrm{diag}(\boldsymbol{\pi})]^{-\frac{1}{2}}\big\|_2$$

$$\leq \big(\|\mathbf{F}\|_2 + \big\|n^{-1}\mathbf{N} - \mathbf{F}\big\|_2\big) \cdot \max_{1 \leq j \leq p}\big|(n^{-1}m_j)^{-\frac{1}{2}} - \pi_j^{-\frac{1}{2}}\big|.$$

Here, $\|\mathbf{F}\|_2 \leq \sqrt{\pi_{\max}}$ by (E .3). Notice that $\left|x^{-1/2} - 1\right| \leq |x - 1|$ for any $x \geq \frac{1}{2}$. Taking $n \geq 4C_0^2 C_1 c_1^{-2} \tau_* p \log^2(n)$, we have

$$
\begin{aligned}
\left|(n^{-1}m_j)^{-\frac{1}{2}} - \pi_j^{-\frac{1}{2}}\right| &= \pi_j^{-\frac{1}{2}}\left|\left(\pi_j^{-1}n^{-1}m_j\right)^{-\frac{1}{2}} - 1\right| \\
&\leq \pi_j^{-\frac{1}{2}}\left|\pi_j^{-1}n^{-1}m_j - 1\right| \leq \pi_j^{-\frac{3}{2}}\|n^{-1}\mathbf{m} - \boldsymbol{\pi}\|_\infty
\end{aligned}
\tag{B .33}
$$

for $j = 1, 2, \ldots, p$. It follows from condition (B .30) that

$$
\begin{aligned}
\max_{1\leq j\leq p}\left|(n^{-1}m_j)^{-\frac{1}{2}} - \pi_j^{-\frac{1}{2}}\right| &\leq \pi_{\min}^{-\frac{3}{2}} \cdot C_0 n^{-1/2}\sqrt{\pi_{\max}\tau_* \log^2(n)} \\
&\leq C_0 C_1^{\frac{1}{2}} c_1^{-\frac{3}{2}} n^{-1/2} p\sqrt{\tau_* \log^2(n)}.
\end{aligned}
$$

When $n \geq C_0 \tau_* p \log^2(n)$, condition (B .30) also implies

$$
\begin{aligned}
\left\|n^{-1}\mathbf{N} - \mathbf{F}\right\|_2 &\leq C_0\left(C_0\tau_* p\log^2(n)\right)^{-\frac{1}{2}}\sqrt{\pi_{\max}\tau_* \log^2(n)} \\
&\leq \sqrt{C_0\pi_{\max}p^{-1}} \leq C_0^{\frac{1}{2}}C_1^{\frac{1}{2}}p^{-1}.
\end{aligned}
$$

The second term in (B .31) then satisfies

$$
\begin{aligned}
&\left\|(n^{-1}\mathbf{N})\left([\operatorname{diag}(n^{-1}\mathbf{m})]^{-\frac{1}{2}} - [\operatorname{diag}(\boldsymbol{\pi})]^{-\frac{1}{2}}\right)\right\|_2 \\
&\leq (C_0^{\frac{1}{2}} + C_1^{\frac{1}{2}})C_0 C_1 c_1^{-\frac{3}{2}} n^{-1/2}\sqrt{\tau_* \log^2(n)}.
\end{aligned}
\tag{B .34}
$$

Plugging (B .32) and (B .34) in (B .31), we can conclude that, when $n \geq C_0\tau_* p \log^2(n)$,

$$
\mathbb{P}\left(\left\|\widehat{\mathbf{Q}} - \mathbf{Q}\right\|_2 \geq \widetilde{C}n^{-1/2}\sqrt{\tau \log^2(n)}\right) \leq 2n^{-c_0}
$$

for some constant $\widetilde{C} > 0$. $\qquad\square$

**Corollary B .2.** *Under assumptions* (A .1) *and* (A .6), *for any* $c_0 > 0$, *there exists a constant* $C > 0$, *such that if* $n \geq C\tau_* p \log^2(n)$, *then*

$$
\mathbb{P}\left(\left\|\mathbf{G}^\top(\widehat{\mathbf{Q}} - \mathbf{Q})\mathbf{e}_j\right\|_2 \geq Cn^{-1/2}\sqrt{\tau_*(r/p) \log^2(n)}\right) \leq n^{-c_0}.
\tag{B .35}
$$

*Proof.* Recall that we have proved $\left\|\mathbf{e}_j^\top\mathbf{G}\right\|_2 \leq c_2^{-\frac{1}{2}}\pi_j\sqrt{pr}$ in Lemma A .2. Let $\gamma = C_1 c_2^{-\frac{1}{2}}$. Then

$$
\max_{1\leq j\leq p}\left\|\mathbf{e}_j^\top\mathbf{G}\right\|_2 \leq \gamma\sqrt{r/p}.
$$

$\mathbf{e}_j$ and $\mathbf{G}$ satisfy the conditions in Lemma B .4. For a fixed $c_0 > 0$, Lemma B .3 and B .4 imply that there exists a constant $C_0 > 0$ such that when $n \geq C_0\tau_* p \log^2(n)$, by union bound, with probability at least $1 - 2n^{-c_0}$,

$$
\|n^{-1}\mathbf{m} - \boldsymbol{\pi}\|_\infty \leq C_0 n^{-1/2}\sqrt{\pi_{\max}\tau_* \log^2(n)}
\tag{B .36}
$$

and

$$
\max_{1\leq j\leq p}\left\|\mathbf{G}^\top(n^{-1}\mathbf{N} - \mathbf{F})[\operatorname{diag}(\boldsymbol{\pi})]^{-\frac{1}{2}}\mathbf{e}_j\right\|_2 \leq C_0 n^{-1/2}\sqrt{\tau_*(\gamma^2 r/p) \log^2(n)}.
\tag{B .37}
$$

Note that

$$
\begin{aligned}
\left\|\mathbf{G}^\top(\widehat{\mathbf{Q}} - \mathbf{Q})\mathbf{e}_j\right\|_2 &= \left\|\mathbf{G}^\top\left((n^{-1}\mathbf{N})[\operatorname{diag}(n^{-1}\mathbf{m})]^{-\frac{1}{2}} - \mathbf{F}[\operatorname{diag}(\boldsymbol{\pi})]^{-\frac{1}{2}}\right)\mathbf{e}_j\right\|_2 \\
&\leq \left\|\mathbf{G}^\top(n^{-1}\mathbf{N} - \mathbf{F})[\operatorname{diag}(\boldsymbol{\pi})]^{-\frac{1}{2}}\mathbf{e}_j\right\|_2 + \left|(n^{-1}m_j)^{-\frac{1}{2}} - \pi_j^{-\frac{1}{2}}\right| \cdot \left\|\mathbf{G}^\top(n^{-1}\mathbf{N})\mathbf{e}_j\right\|_2.
\end{aligned}
\tag{B .38}
$$

The inequality (B .37) provides an upper bound for the first term in (B .38). In (B .33), there is an estimate of $\left|(n^{-1}m_j)^{-\frac{1}{2}} - \pi_j^{-\frac{1}{2}}\right|$. We only need to analyze $\left\|\mathbf{G}^\top(n^{-1}\mathbf{N})\mathbf{e}_j\right\|_2$ in the following.

We find that

$$\left\|\mathbf{G}^\top \mathbf{F} \mathbf{e}_j\right\|_2 = \left\|\mathbf{G}^\top \mathbf{Q}[\mathrm{diag}(\boldsymbol{\pi})]^{\frac{1}{2}} \mathbf{e}_j\right\|_2 = \sqrt{\pi_j}\left\|\mathbf{\Sigma}\mathbf{H}^\top \mathbf{e}_j\right\|_2$$
$$\leq \sqrt{\pi_j}\sigma_1\left\|\mathbf{e}_j^\top \mathbf{H}\right\|_2 \leq C_1^2 c_1^{-\frac{1}{2}} c_2^{-\frac{3}{2}} \pi_j \sqrt{r/p}, \tag{B.39}$$

where we used (A.9) and (A.7).

Additionally, when $n \geq C_0\tau_* p \log^2(n)$, (B.37) shows that

$$\left\|\mathbf{G}^\top (n^{-1}\mathbf{N} - \mathbf{F})\mathbf{e}_j\right\|_2 = \sqrt{\pi_j}\left\|\mathbf{G}^\top (n^{-1}\mathbf{N} - \mathbf{F})[\mathrm{diag}(\boldsymbol{\pi})]^{-\frac{1}{2}}\mathbf{e}_j\right\|_2$$
$$\leq \sqrt{\pi_j} C_0 \left(C_0\tau_* p \log^2(n)\right)^{-\frac{1}{2}} \sqrt{\tau_*(\gamma^2 r/p)\log^2(n)} \tag{B.40}$$
$$= C_0^{\frac{1}{2}} \gamma p^{-1} \sqrt{\pi_j r} \leq C_0^{\frac{1}{2}} C_1^{\frac{1}{2}} c_1^{-\frac{1}{2}} \gamma p^{-\frac{3}{2}} r^{\frac{1}{2}}.$$

Combining (B.39) and (B.40) gives

$$\left\|\mathbf{G}^\top (n^{-1}\mathbf{N})\mathbf{e}_j\right\|_2 \leq \left\|\mathbf{G}^\top \mathbf{F}\mathbf{e}_j\right\|_2 + \left\|\mathbf{G}^\top (n^{-1}\mathbf{N} - \mathbf{F})\mathbf{e}_j\right\|_2 \leq \widetilde{C} p^{-\frac{3}{2}} r^{\frac{1}{2}} \tag{B.41}$$

for some constant $\widetilde{C} > 0$.

Plugging (B.36), (B.37), (B.33) and (B.41) into (B.38), we complete the proof of Corollary B.2.

$$\square$$

**Corollary B.3.** *Suppose $\boldsymbol{\eta} \in \mathbb{R}^p$ is a nonnegative unit vector. Under assumptions* (A.1) *and* (A.5), *if $\mathbf{h}_1^\top \boldsymbol{\eta} \leq \beta p^{-\frac{1}{2}}$ for some $\beta > 0$, then for any $c_0 > 0$, there exists a constant $C > 0$, such that when $n \geq C\tau_* p \log^2(n)$,*

$$\mathbb{P}\left(\left|\mathbf{g}_1^\top (\widehat{\mathbf{Q}} - \mathbf{Q})\boldsymbol{\eta}\right| \geq C n^{-1/2}\sqrt{\tau_* p^{-1}\log^2(n)}\right) \leq n^{-c_0}. \tag{B.42}$$

*Symmetrically, if $\boldsymbol{\eta}^\top \mathbf{g}_1 \leq \beta p^{-1}$ then for any $c_0 > 0$, there exists a constant $C > 0$, such that when $n \geq C\tau_* p \log^2(n)$,*

$$\mathbb{P}\left(\left|\boldsymbol{\eta}^\top (\widehat{\mathbf{Q}} - \mathbf{Q})\mathbf{h}_1\right| \geq C n^{-1/2}\sqrt{\tau_* p^{-1}\log^2(n)}\right) \leq n^{-c_0}. \tag{B.43}$$

*Proof.* In Lemma A.4, we proved that $0 < \mathbf{h}_1(j) \leq C^\# \sqrt{\pi_j}$ and $0 \leq \mathbf{g}_1(j) \leq c^\# \pi_j \sqrt{p}$ for $j = 1, 2, \ldots, p$ and some $c^\#, C^\# > 0$. Let $\gamma = C_1^{\frac{1}{2}} C^\# \vee C_1 c^\#$, then $\left|\mathbf{h}_1(j)\right| \leq \gamma p^{-\frac{1}{2}}$ and $\left|\mathbf{g}_1(j)\right| \leq \gamma p^{-\frac{1}{2}}$. Lemma B.3 and B.4 show that for any fixed $c_0 > 0$, there exists a constant $C_0 > 0$ such that when $n \geq C_0\tau_* p \log^2(n)$, with probability at least $1 - 2n^{-c_0}$,

$$\left|\mathbf{g}_1^\top (n^{-1}\mathbf{N} - \mathbf{F})[\mathrm{diag}(\boldsymbol{\pi})]^{-\frac{1}{2}}\boldsymbol{\eta}\right| \leq C_0 n^{-1/2}\sqrt{\tau_* \gamma^2 p^{-1}\log^2(n)}, \tag{B.44}$$

$$\left|\boldsymbol{\eta}^\top (n^{-1}\mathbf{N} - \mathbf{F})[\mathrm{diag}(\boldsymbol{\pi})]^{-\frac{1}{2}}\mathbf{h}_1\right| \leq C_0 n^{-1/2}\sqrt{\tau_* \gamma^2 p^{-1}\log^2(n)}, \tag{B.45}$$

$$\left\|n^{-1}\mathbf{m} - \boldsymbol{\pi}\right\|_\infty \leq C_0 n^{-1/2}\sqrt{\pi_{\max}\tau_* \log^2(n)}. \tag{B.46}$$

Note that

$$\left|\mathbf{g}_1^\top (\widehat{\mathbf{Q}} - \mathbf{Q})\boldsymbol{\eta}\right| \leq \left|\mathbf{g}_1^\top (n^{-1}\mathbf{N} - \mathbf{F})[\mathrm{diag}(\boldsymbol{\pi})]^{-\frac{1}{2}}\boldsymbol{\eta}\right|$$
$$+ \left|\mathbf{g}_1^\top (n^{-1}\mathbf{N})\left([\mathrm{diag}(n^{-1}\mathbf{m})]^{-\frac{1}{2}} - [\mathrm{diag}(\boldsymbol{\pi})]^{-\frac{1}{2}}\right)\boldsymbol{\eta}\right|. \tag{B.47}$$

Since $\mathbf{g}_1, \mathbf{N}, \boldsymbol{\eta} \geq 0$,

$$\left|\mathbf{g}_1^\top (n^{-1}\mathbf{N})\left([\mathrm{diag}(n^{-1}\mathbf{m})]^{-\frac{1}{2}} - [\mathrm{diag}(\boldsymbol{\pi})]^{-\frac{1}{2}}\right)\boldsymbol{\eta}\right|$$
$$\leq \left|\mathbf{g}_1^\top (n^{-1}\mathbf{N})[\mathrm{diag}(\boldsymbol{\pi})]^{-\frac{1}{2}}\boldsymbol{\eta}\right| \cdot \max_{1 \leq i \leq p}\left|(n^{-1}m_i/\pi_i)^{-\frac{1}{2}} - 1\right| \tag{B.48}$$
$$\leq \left(\left|\mathbf{g}_1^\top (n^{-1}\mathbf{N} - \mathbf{F})[\mathrm{diag}(\boldsymbol{\pi})]^{-\frac{1}{2}}\boldsymbol{\eta}\right| + \mathbf{g}_1^\top \mathbf{Q}\boldsymbol{\eta}\right) \cdot \max_{1 \leq i \leq p}\left|\sqrt{n^{-1}m_i/\pi_i} - 1\right|.$$

By (B .44), when $n \geq C_0 \tau_* p \log^2(n)$,

$$
\begin{aligned}
&\left| \mathbf{g}_1^\top (n^{-1}\mathbf{N} - \mathbf{F})[\mathrm{diag}(\boldsymbol{\pi})]^{-\frac{1}{2}} \boldsymbol{\eta} \right| \\
&\leq C_0 \big( C_0 \tau_* p \log^2(n) \big)^{-\frac{1}{2}} \sqrt{\tau_* \gamma^2 p^{-1} \log^2(n)} \leq C_0^{\frac{1}{2}} \gamma p^{-1}.
\end{aligned}
\tag{B .49}
$$

In addition,

$$
\mathbf{g}_1^\top \mathbf{Q} \boldsymbol{\eta} = \sigma_1 \mathbf{h}_1^\top \boldsymbol{\eta} \leq C_1 c_1^{-\frac{1}{2}} \beta p^{-1},
\tag{B .50}
$$

where we used (A .7) and condition $\mathbf{h}_1^\top \boldsymbol{\eta} \leq \beta p^{-\frac{1}{2}}$. According to (B .33), we have

$$
\max_{1 \leq i \leq p} \left| \sqrt{n^{-1} m_i / \pi_i} - 1 \right| \leq \pi_{\min}^{-1} \big\| n^{-1} \mathbf{m}_j - \boldsymbol{\pi} \big\|_\infty.
\tag{B .51}
$$

Combining (B .47) - (B .51) gives (B .42). We can prove (B .43) in the same way.

$\square$

## B .4   Plugging Concentration Inequalities into Deterministic Bounds

**Theorem B .1** (Theorem 2 in paper). *Let* $\mathbf{H}^* = \big[\mathbf{h}_2, \mathbf{h}_3, \ldots, \mathbf{h}_r\big]$ *and* $\widehat{\mathbf{H}}^* = \big[\widehat{\mathbf{h}}_2, \widehat{\mathbf{h}}_3, \ldots, \widehat{\mathbf{h}}_r\big]$. *Under assumptions* (A .1), (A .6) *and* (A .4), *for a fixed* $c_0 > 0$, *there exists a constant* $C > 0$ *and an orthogonal matrix* $\boldsymbol{\Omega}_* \in \mathbb{O}^{(r-1) \times (r-1)}$, *such that if* $n \geq C \tau_* p \log^2(n)$,

$$
\begin{aligned}
\mathbb{P} \Bigg( &\max_{1 \leq j \leq p} \big\| \mathbf{e}_j^\top \big( \widehat{\mathbf{H}}^* \boldsymbol{\Omega}^* - \mathbf{H}^* \big) \big\|_2 \\
&\geq C \Big( n^{-1/2} \sqrt{\tau_* r \log^2(n)} + n^{-1} \tau_* p \log^2(n) \Big) \Bigg) \leq n^{-c_0}.
\end{aligned}
\tag{B .52}
$$

*Proof.* According to Lemma B .1, $\widehat{\mathbf{H}}^*$ has a deterministic row-wise perturbation bound

$$
\begin{aligned}
\big\| \mathbf{e}_j^\top (\widehat{\mathbf{H}}_* \boldsymbol{\Omega}_* - \mathbf{H}_*) \big\|_2 \leq &\frac{2}{\Delta^*} \big\| \mathbf{G}_*^\top (\widehat{\mathbf{Q}} - \mathbf{Q}) \mathbf{e}_j \big\|_2 + \frac{4(1 + \sqrt{2})}{\Delta^*} \big\| \mathbf{e}_j^\top \mathbf{H} \big\|_2 \big\| \widehat{\mathbf{Q}} - \mathbf{Q} \big\|_2 \\
&+ \frac{8}{(\Delta^*)^2} \big\| \widehat{\mathbf{Q}} - \mathbf{Q} \big\|_2^2
\end{aligned}
\tag{B .53}
$$

Here, $\mathbf{G}^* = \big[\mathbf{g}_2, \mathbf{g}_3, \ldots, \mathbf{g}_r\big]$ and $\Delta^* = \min\{\sigma_1 - \sigma_2, \sigma_r\}$. We note that $\big\| \mathbf{G}_*^\top (\widehat{\mathbf{Q}} - \mathbf{Q}) \mathbf{e}_j \big\|_2 \leq \big\| \mathbf{G}^\top (\widehat{\mathbf{Q}} - \mathbf{Q}) \mathbf{e}_j \big\|_2$. Based on Corollary B .1 and B .2, for a fixed $c_0 > 0$, there exists a constant $C_0 > 0$ such that when $n \geq C_0 \tau_* p \log^2(n)$, by union bound, with probability at least $1 - 2n^{-c_0}$,

$$
\big\| \widehat{\mathbf{Q}} - \mathbf{Q} \big\|_2 \leq C_0 n^{-1/2} \sqrt{\tau_* \log^2(n)},
\tag{B .54}
$$

$$
\big\| \mathbf{G}^\top (\widehat{\mathbf{Q}} - \mathbf{Q}) \mathbf{e}_j \big\|_2 \leq C_0 n^{-1/2} \sqrt{\tau_* (r/p) \log^2(n)},
\tag{B .55}
$$

for $j = 1, 2, \ldots, p$. Recall that $\sigma_1 - \sigma_2 \geq c_3 p^{-\frac{1}{2}}$ by assumption (A .4), and $\sigma_r \geq c_2 p^{-\frac{1}{2}}$ by Lemma A .1. We have

$$
\Delta^* \geq (c_2 \wedge c_3) p^{-\frac{1}{2}}.
\tag{B .56}
$$

Plugging (B .54), (B .55), (A .8) and (B .56) into (B .53), we complete the proof of Theorem B .1.

$\square$

**Theorem B .2** (Theorem 1 in paper). *Under assumptions* (A .1), (A .4) *and* (A .5), *for any constant* $c_0 > 0$, *there exists a constant* $C > 0$ *and* $\omega \in \{\pm 1\}$ *such that if* $n \geq C \tau_* p \big(\log^2(r) \vee 1\big) \log^2(n)$,

$$
\begin{aligned}
\mathbb{P} \Bigg( \max_{1 \leq j \leq p} \big| \omega \widehat{\mathbf{h}}_1(j) - \mathbf{h}_1(j) \big| \geq &C \Big( n^{-1/2} \sqrt{\tau_* \big(\log^2(r) \vee 1\big) \log^2(n)} \\
&+ n^{-1} \tau_* p \big(\log^2(r) \vee 1\big) \log^2(n) \Big) \Bigg) \leq n^{-c_0}.
\end{aligned}
$$

*Proof.* In Lemma B .2, we take

$$K = \left\lceil \left[ 2\log\left(\frac{\sigma_1 + \sigma_2}{2\sigma_2}\right) \right]^{-1} \log(r) \right\rceil \vee 1,$$

then

$$\left(\frac{2\sigma_2}{\sigma_1 + \sigma_2}\right)^K \leq r^{-\frac{1}{2}}.$$

Fix $c_0 > 0$. According to Corollary B .1, there exists a constant $C_0 \geq c_3^2$ such that when $n \geq C_0 \tau_* p \log^2(n)$, with probability at least $1 - n^{-c_0}$,

$$\left\| \widehat{\mathbf{Q}} - \mathbf{Q} \right\|_2 \leq C_0 n^{-1/2} \sqrt{\tau_* \log^2(n)}. \tag{B .57}$$

We further take $n \geq C_0^2 c_3^{-2} K^2 \tau_* p \log^2(n)$, then

$$\left\| \widehat{\mathbf{Q}} - \mathbf{Q} \right\|_2 \leq \frac{\sigma_2 - \sigma_1}{2K}.$$

By Lemma B .2, there exists $\omega \in \{\pm 1\}$ such that

$$\left| \omega \widehat{\mathbf{h}}_1(j) - \mathbf{h}_1(j) \right|$$
$$\leq \sum_{k=1}^{\lceil K/2 \rceil} \left( \sigma_1^{-2k} \left| \mathbf{e}_j^\top \mathbf{Q}^\top (\mathbf{Q}\mathbf{Q}^\top)^{k-1} (\widehat{\mathbf{Q}} - \mathbf{Q}) \mathbf{h}_1 \right| + \sigma_1^{-2k-1} \left| \mathbf{g}_1^\top (\widehat{\mathbf{Q}} - \mathbf{Q}) (\mathbf{Q}^\top \mathbf{Q})^k \mathbf{e}_j \right| \right)$$
$$+ |\mathbf{h}_1(j)| \cdot (2K + 1 + 2\sqrt{2}) \delta^{-1} \left\| \widehat{\mathbf{Q}} - \mathbf{Q} \right\|_2 + 4\sqrt{2} r^{-\frac{1}{2}} \left\| \mathbf{e}_j^\top \mathbf{H} \right\|_2 \delta^{-1} \left\| \widehat{\mathbf{Q}} - \mathbf{Q} \right\|_2 \tag{B .58}$$
$$+ \left( 2K^2 + 8\sqrt{2}K \right) \delta^{-2} \left\| \widehat{\mathbf{Q}} - \mathbf{Q} \right\|_2^2.$$

Note that for $k = 1, 2, \ldots, \lceil K/2 \rceil$,

$$\sigma_1^{-2k+1} \mathbf{e}_j^\top \mathbf{Q}^\top (\mathbf{Q}\mathbf{Q}^\top)^{k-1} \mathbf{g}_1 = \sigma_1^{-2k+1} \mathbf{e}_j^\top \mathbf{Q}^\top (\mathbf{Q}\mathbf{Q}^\top)^{k-2} \mathbf{Q}\mathbf{Q}^\top \mathbf{g}_1$$
$$= \sigma_1^{-2k+2} \mathbf{e}_j^\top \mathbf{Q}^\top (\mathbf{Q}\mathbf{Q}^\top)^{k-2} \mathbf{Q} \mathbf{h}_1 = \sigma_1^{-2(k-1)+1} \mathbf{e}_j^\top \mathbf{Q}^\top (\mathbf{Q}\mathbf{Q}^\top)^{k-2} \mathbf{g}_1$$
$$= \ldots$$
$$= \sigma_1^{-1} \mathbf{e}_j^\top \mathbf{Q}^\top \mathbf{g}_1 = \mathbf{e}_j^\top \mathbf{h}_1 = \mathbf{h}_1(j),$$

where we used $\mathbf{Q}^\top \mathbf{g}_1 = \sigma_1 \mathbf{h}_1$ and $\mathbf{Q}\mathbf{h}_1 = \sigma \mathbf{g}_1$ iteratively. Similarly,

$$\sigma_1^{-2k} \mathbf{h}_1^\top (\mathbf{Q}^\top \mathbf{Q})^k \mathbf{e}_j = \mathbf{g}_1(j).$$

Therefore, according to Lemmat A .4, there exists some constant $C^\# > 0$ such that $\boldsymbol{v}_{1,2k-1} = \sigma_1^{-2k+1} \mathbf{Q}(\mathbf{Q}^\top \mathbf{Q})^{k-1} \mathbf{e}_j$ satisfies

$$\boldsymbol{v}_{1,2k-1} \geq 0, \quad \|\boldsymbol{v}_{1,2k-1}\|_2 = 1 \quad \text{and} \quad \boldsymbol{v}_{1,2k-1}^\top \mathbf{g}_1 = \mathbf{h}_1(j) \leq C^\# p^{-\frac{1}{2}},$$

and $\boldsymbol{v}_{2,2k} = \sigma_1^{-2k} (\mathbf{Q}^\top \mathbf{Q})^k \mathbf{e}_j$ satisfies

$$\boldsymbol{v}_{2,2k} \geq 0, \quad \|\boldsymbol{v}_{2,2k}\|_2 = 1 \quad \text{and} \quad \mathbf{h}_1^\top \boldsymbol{v}_{2,2k} = \mathbf{g}_1(j) \leq C^\# p^{-\frac{1}{2}}.$$

According to Corollary B .3, there exists a constant $\widetilde{C}_0 \geq C_0^2 c_3^{-2}$ such that when $n \geq \widetilde{C}_0 K^2 \tau_* p \log^2(n)$, with probability at least $1 - 2\lceil K/2 \rceil n^{-c_0}$,

$$\left| \boldsymbol{v}_{1,2k-1}^\top (\widehat{\mathbf{Q}} - \mathbf{Q}) \mathbf{h}_1 \right| \leq \widetilde{C}_0 n^{-1/2} \sqrt{\tau_* p^{-1} \log^2(n)}, \tag{B .59}$$

$$\left| \mathbf{g}_1^\top (\widehat{\mathbf{Q}} - \mathbf{Q}) \boldsymbol{v}_{2,2k} \right| \leq \widetilde{C}_0 n^{-1/2} \sqrt{\tau_* p^{-1} \log^2(n)}, \tag{B .60}$$

for $k = 1, 2, \ldots, \lceil K/2 \rceil$.

Plugging (B .57), (B .59), (B .60) into (B .58) and using Lemma A .2 and A .4, we have

$$\left| \omega \widehat{\mathbf{h}}_1(j) - \mathbf{h}_1(j) \right| \leq \widetilde{C} \left( K n^{-1/2} \sqrt{\tau_* \log^2(n)} + K^2 n^{-1} \tau_* p \log^2(n) \right)$$

for some sufficiently large $\widetilde{C} > 0$. Notice that $K$ has order $\log(r)$, we complete the proof of Theorem B .2. □

## C  Proof of Statistical Guarantees

Define

$$err = \min_{\omega \in \{\pm 1\}} \max_{1 \le j \le p} \left| \omega \widehat{\mathbf{h}}_1(j) - \mathbf{h}_1(j) \right|, \tag{C.1}$$

$$Err = \min_{\mathbf{\Omega} \in \mathbb{O}^{(r-1)\times(r-1)}} \max_{1 \le j \le p} \left\| \mathbf{\Omega} \widehat{\mathbf{d}}_j - \mathbf{d}_j \right\|_2. \tag{C.2}$$

### C.1  SCORE Normalization

The bound for $err$ is shown in Theorem B.2. It remains to estimate $Err$.

**Theorem C.1.** *Under assumptions* (A.1) - (A.5), *for any $c_0 > 0$, there exists a constant $C > 0$ and an orthogonal matrix $\mathbf{\Omega} \in \mathbb{O}^{(r-1)\times(r-1)}$, such that if $n \ge C\tau_* p^{\frac{3}{2}} \left( \log^2(r) \vee 1 \right) \log^2(n)$,*

$$\mathbb{P}\left( \max_{1 \le j \le p} \left\| \mathbf{\Omega} \widehat{\mathbf{d}}_j - \mathbf{d}_j \right\|_2 \ge C \left( n^{-1/2} \sqrt{\tau_* pr \left( \log^2(r) \vee 1 \right) \log^2(n)} \right.\right.$$

$$\left.\left. + n^{-1} \tau_* p^{\frac{3}{2}} r^{\frac{1}{2}} \left( \log^2(r) \vee 1 \right) \log^2(n) \right) \right) \le n^{-c_0}.$$

*Proof.* By definition,

$$\widehat{\mathbf{d}}_j = \left[ \widehat{\mathbf{h}}_1(j) \right]^{-1} \widehat{\mathbf{H}}_*^\top \mathbf{e}_j, \quad \text{for } j = 1, 2, \ldots, p,$$

where $\widehat{\mathbf{H}}_* = \left[ \widehat{\mathbf{h}}_2, \widehat{\mathbf{h}}_3, \ldots, \widehat{\mathbf{h}}_r \right]$. According to Theorem B.1 and B.2, for a fixed $c_0 > 0$, there exists a constant $C_0 \ge 1$, $\omega \in \{\pm 1\}$ and $\mathbf{\Omega}_* \in \mathbb{R}^{(r-1)\times(r-1)}$ such that when $n \ge C_0 \tau_* p \left( \log^2(r) \vee 1 \right) \log^2(n)$, with probability at least $1 - 2n^{-c_0}$,

$$\left| \omega \widehat{\mathbf{h}}_1(j) - \mathbf{h}_1(j) \right| \le C_0 \left( n^{-1/2} \sqrt{\tau_* \left( \log^2(r) \vee 1 \right) \log^2(n)} \right.$$

$$\left. + n^{-1} \tau_* p \left( \log^2(r) \vee 1 \right) \log^2(n) \right), \tag{C.3}$$

$$\left\| \mathbf{e}_j^\top \left( \widehat{\mathbf{H}}^* \mathbf{\Omega}^* - \mathbf{H}^* \right) \right\|_2 \le C_0 \left( n^{-1/2} \sqrt{\tau_* r \log^2(n)} + n^{-1} \tau_* p \log^2(n) \right), \tag{C.4}$$

for $j = 1, 2, \ldots, p$.

Define $\mathbf{\Omega} = \omega \mathbf{\Omega}_*^\top$. We find that

$$\begin{aligned}
& \left\| \mathbf{\Omega} \widehat{\mathbf{d}}_j - \mathbf{d}_j \right\|_2 \\
= & \left\| \left[ \omega \widehat{\mathbf{h}}_1(j) \right]^{-1} \mathbf{\Omega}_*^\top \widehat{\mathbf{H}}_*^\top \mathbf{e}_j - \left[ \mathbf{h}_1(j) \right]^{-1} \mathbf{H}_*^\top \mathbf{e}_j \right\|_2 \\
\le & \left| \mathbf{h}_1(j) \right|^{-1} \left\| \mathbf{e}_j^\top \left( \widehat{\mathbf{H}}_* \mathbf{\Omega}_* - \mathbf{H}_* \right) \right\|_2 \\
& + \left| \left[ \omega \widehat{\mathbf{h}}_1(j) \right]^{-1} - \left[ \mathbf{h}_1(j) \right]^{-1} \right| \cdot \left\| \mathbf{e}_j^\top \mathbf{H}_* \right\|_2 \\
& + \left| \left[ \omega \widehat{\mathbf{h}}_1(j) \right]^{-1} - \left[ \mathbf{h}_1(j) \right]^{-1} \right| \cdot \left\| \mathbf{e}_j^\top \left( \widehat{\mathbf{H}}_* \mathbf{\Omega}_* - \mathbf{H}_* \right) \right\|_2.
\end{aligned} \tag{C.5}$$

Since $\left| x^{-1} - 1 \right| \le 2|x - 1|$ for $x \ge \frac{1}{2}$, if we take $n \ge C^{\#} \tau_* p^{\frac{3}{2}} \left( \log^2(r) \vee 1 \right) \log^2(n)$ for some large enough $C^{\#} > 0$, then $\left| \omega \widehat{\mathbf{h}}_1(j) - \mathbf{h}_1(j) \right| \le \frac{1}{2} \mathbf{h}_1(j)$,

$$\left| \left[ \omega \widehat{\mathbf{h}}_1(j) \right]^{-1} - \left[ \mathbf{h}_1(j) \right]^{-1} \right| = \left[ \mathbf{h}_1(j) \right]^{-1} \left| \left[ \frac{\omega \widehat{\mathbf{h}}_1(j)}{\mathbf{h}_1(j)} \right]^{-1} - 1 \right|$$

$$\le 2 \left[ \mathbf{h}_1(j) \right]^{-1} \left| \frac{\omega \widehat{\mathbf{h}}_1(j)}{\mathbf{h}_1(j)} - 1 \right| = 2 \left[ \mathbf{h}_1(j) \right]^{-2} \left| \omega \widehat{\mathbf{h}}_1(j) - \mathbf{h}_1(j) \right|.$$

Using the fact in Lemma A .2 and A .4 that $\mathbf{h}_1(j) \geq c^{\#}\sqrt{\pi_j}$ for some $c^{\#} > 0$ and $\left\|\mathbf{e}_j^\top \mathbf{H}_*\right\|_2 \leq \left\|\mathbf{e}_j^\top \mathbf{H}\right\|_2 \leq C_1 c_2^{-\frac{3}{2}}\sqrt{\pi_j r}$, we reduce (C .5) into

$$\left\|\mathbf{\Omega}\widehat{\mathbf{d}}_j - \mathbf{d}_j\right\|_2 \leq \widetilde{C}\Big(n^{-1/2}\sqrt{\tau_* pr\left(\log^2(r) \vee 1\right)\log^2(n)}$$
$$+ n^{-1}\tau_* p^{\frac{3}{2}}r^{\frac{1}{2}}\left(\log^2(r) \vee 1\right)\log^2(n)\Big),$$

where $\widetilde{C} > 0$ is a constant.

$\square$

## C .2  Vertex Hunting

The estimated vertices $\{\widehat{\mathbf{b}}_1, \widehat{\mathbf{b}}_2, \ldots, \widehat{\mathbf{b}}_r\}$ solves the following optimization problem:

$$\operatorname*{minimize}_{\{\widehat{\mathbf{b}}_1,\ldots,\widehat{\mathbf{b}}_r\} \subseteq \{\widehat{\mathbf{a}}_1,\ldots,\widehat{\mathbf{a}}_p\}} \quad \max_{1 \leq j \leq p} \left\|\widehat{\mathbf{d}}_j - \mathcal{P}_{\mathcal{S}}\widehat{\mathbf{d}}_j\right\|_2,$$

where $\mathcal{S}$ is the convex hull of $\widehat{\mathbf{b}}_1, \widehat{\mathbf{b}}_2, \ldots, \widehat{\mathbf{b}}_r$ and $\mathcal{P}$ is the projection operator induced by Euclidean norm. One can refer to [5] for further details of vertex hunting algorithms.

**Theorem C .2** (Vertex hunting). *Suppose that for each meta-state, there exists at least one anchor state. Then there exist constants $\alpha, \alpha' > 0$ such that if $Err \leq \alpha'\sqrt{r}$,*

$$\max_{1 \leq k \leq r}\left\|\mathbf{\Omega}\widehat{\mathbf{b}}_k - \mathbf{b}_k\right\|_2 \leq \alpha \cdot Err,$$

*where $\mathbf{\Omega}$ is the orthogonal matrix that achieves the minimum in the definition of $Err$.*

*Proof.* Inspired by the proof of Lemma 3.1 in [5], we define a mapping $\mathcal{R}$ that maps a weight vector in the standard simplex $\mathcal{S}_{r-1} \subseteq \mathbb{R}^r$ to a vector in the simplex $\mathcal{S}(\mathbf{b}_1, \mathbf{b}_2, \ldots, \mathbf{b}_r)$:

$$\mathbf{w} \overset{\mathcal{R}}{\longmapsto} [\mathbf{b}_1, \mathbf{b}_2, \ldots, \mathbf{b}_r]\mathbf{w}.$$

To begin with, we prove that the mapping $\mathcal{R}$ has the following properties:

1. $\mathcal{R}\mathbf{w}_j = \mathbf{d}_j$ for $j = 1, 2, \ldots, p$.

2. There exist constants $C^\star > 0$, $c^\star > 0$ such that for any two weight vectors $\mathbf{w}, \mathbf{w}' \in \mathcal{S}_{r-1}$,
$$c^\star\sqrt{r}\left\|\mathbf{w} - \mathbf{w}'\right\|_2 \leq \left\|\mathcal{R}\mathbf{w} - \mathcal{R}\mathbf{w}'\right\|_2 \leq C^\star\sqrt{r}\left\|\mathbf{w} - \mathbf{w}'\right\|_2. \tag{C .6}$$

3. $\mathcal{R}$ is a bijection.

1. & 3. are obvious, we only need to show 2. Note that

$$\begin{pmatrix} 0 \\ \mathcal{R}\widetilde{\mathbf{w}} \end{pmatrix} - \begin{pmatrix} 0 \\ \mathcal{R}\widetilde{\mathbf{w}}' \end{pmatrix} = \begin{pmatrix} 1 & \cdots & 1 \\ \mathbf{b}_1 & \cdots & \mathbf{b}_r \end{pmatrix}(\mathbf{w} - \mathbf{w}').$$

According to Lemma A .3, there exist constants $c^\star > 0$ and $C^\star > 0$ such that $\mathbf{B} = \begin{pmatrix} 1 & \cdots & 1 \\ \mathbf{b}_1 & \cdots & \mathbf{b}_r \end{pmatrix} = \mathbf{L}^\top[\operatorname{diag}(\mathbf{l}_1)]^{-1}$ satisfies

$$\sigma_1(\mathbf{B}) \leq \|\mathbf{L}\|_2\big(\min_{1 \leq k \leq r} \mathbf{l}_1(k)\big)^{-1} \leq C^\star\sqrt{r},$$

$$\sigma_r(\mathbf{B}) \geq \left\|\mathbf{L}^{-1}\right\|_2^{-1}\big(\max_{1 \leq k \leq r} \mathbf{l}_1(k)\big)^{-1} \geq c^\star\sqrt{r}.$$

Hence,
$$c^\star\sqrt{r}\left\|\mathbf{w} - \mathbf{w}'\right\|_2 \leq \left\|\mathcal{R}\mathbf{w} - \mathcal{R}\mathbf{w}'\right\|_2 \leq C^\star\sqrt{r}\left\|\mathbf{w} - \mathbf{w}'\right\|_2.$$

It what follows, we first show that

$$\max_{1 \leq j \leq p} \operatorname{dist}\left(\widehat{\mathbf{d}}_j, \mathcal{S}\big(\widehat{\mathbf{b}}_1, \widehat{\mathbf{b}}_2, \ldots, \widehat{\mathbf{b}}_r\big)\right) \leq 2Err. \tag{C .7}$$

Here, $\text{dist}$ denotes the distance function yielded by the Euclidean norm. Let $a_k \in [p]$ denote an anchor state associated with meta-state $k$. For $j = 1, 2, \ldots, p$,

$$\mathbf{d}_j = \sum_{k=1}^{r} \mathbf{w}_j(k)\mathbf{b}_k = \sum_{k=1}^{r} \mathbf{w}_j(k)\mathbf{d}_{a_k}.$$

Let $\boldsymbol{\Omega}$ be the orthogonal matrix that achieves the minimum in the definition of $Err$. Then

$$\left\| \widehat{\mathbf{d}}_j - \sum_{k=1}^{r} \mathbf{w}_j(k)\widehat{\mathbf{d}}_{a_k} \right\|_2 \leq \left\| \boldsymbol{\Omega}\widehat{\mathbf{d}}_j - \mathbf{d}_j \right\|_2 + \sum_{k=1}^{r} \mathbf{w}_j(k)\left\| \boldsymbol{\Omega}\widehat{\mathbf{d}}_{a_k} - \mathbf{d}_{a_k} \right\|_2 \leq 2Err.$$

In other words,

$$\max_{1 \leq j \leq p} \text{dist}\left( \widehat{\mathbf{d}}_j, \mathcal{S}\left( \widehat{\mathbf{d}}_{a_1}, \widehat{\mathbf{d}}_{a_2}, \ldots, \widehat{\mathbf{d}}_{a_r} \right) \right) \leq 2Err.$$

Our algorithm guarantees that

$$\max_{1 \leq j \leq p} \text{dist}\left( \widehat{\mathbf{d}}_j, \mathcal{S}\left( \widehat{\mathbf{b}}_1, \widehat{\mathbf{b}}_2, \ldots, \widehat{\mathbf{b}}_r \right) \right) \leq \max_{1 \leq j \leq p} \text{dist}\left( \widehat{\mathbf{d}}_j, \mathcal{S}\left( \widehat{\mathbf{d}}_{a_1}, \widehat{\mathbf{d}}_{a_2}, \ldots, \widehat{\mathbf{d}}_{a_r} \right) \right),$$

therefore, we have (C .7).

Let $j_k \in [p]$ be the index such that $\widehat{\mathbf{b}}_k = \widehat{\mathbf{d}}_{j_k}$. We next consider

$$1 - \max_{1 \leq l \leq r} \mathbf{w}_{j_l}(k)$$

for $k = 1, 2, \ldots, r$. For any $\boldsymbol{\eta} \in \mathcal{S}_{r-1}$,

$$1 - \max_{1 \leq l \leq r} \mathbf{w}_{j_l}(k) \leq 1 - \sum_{l=1}^{r} \eta_l \mathbf{w}_{j_l}(k) \leq \left\| \mathbf{e}_k - \sum_{l=1}^{r} \eta_l \mathbf{w}_{j_l} \right\|_2,$$

According to property 1, $\mathcal{R}\mathbf{w}_{j_l} = \mathbf{d}_{j_l}$, $\mathcal{R}\mathbf{e}_k = \mathbf{b}_k = \mathbf{d}_{a_k}$,

$$\left\| \mathbf{e}_k - \sum_{l=1}^{r} \eta_l \mathbf{w}_{j_l} \right\|_2 = \left\| \mathbf{w}_{a_k} - \sum_{l=1}^{r} \eta_l \mathbf{w}_{j_l} \right\|_2$$

$$\overset{(C .6)}{\leq} (c^\star)^{-1} r^{-\frac{1}{2}} \left\| \mathcal{R}\mathbf{w}_{a_k} - \sum_{l=1}^{r} \eta_l \mathcal{R}\mathbf{w}_{j_l} \right\|_2 = (c^\star)^{-1} r^{-\frac{1}{2}} \left\| \mathbf{d}_{a_k} - \sum_{l=1}^{r} \eta_l \mathbf{d}_{j_l} \right\|_2$$

$$\leq (c^\star)^{-1} r^{-\frac{1}{2}} \left( \left\| \widehat{\mathbf{d}}_{a_k} - \sum_{l=1}^{r} \eta_l \widehat{\mathbf{d}}_{j_l} \right\|_2 + 2Err \right)$$

$$= (c^\star)^{-1} r^{-\frac{1}{2}} \left( \left\| \widehat{\mathbf{d}}_{a_k} - \sum_{l=1}^{r} \eta_l \widehat{\mathbf{b}}_l \right\|_2 + 2Err \right).$$

we have

$$1 - \max_{1 \leq l \leq r} \mathbf{w}_{j_l}(k) \leq (c^\star)^{-1} r^{-\frac{1}{2}} \text{dist}\left( \widehat{\mathbf{d}}_{a_k}, \mathcal{S}\left( \widehat{\mathbf{b}}_1, \widehat{\mathbf{b}}_2, \ldots, \widehat{\mathbf{b}}_k \right) \right) + 2(c^\star)^{-1} r^{-\frac{1}{2}} Err.$$

It follows from (C .7) that

$$1 - \max_{1 \leq l \leq r} \mathbf{w}_{j_l}(k) \leq 4(c^\star)^{-1} r^{-\frac{1}{2}} Err. \tag{C .8}$$

When $Err < 8^{-1} c^\star \sqrt{r}$, for each $k$, there is only one $l$ that attains the maximum in (C .8). Based on (C .8), for $k = 1, 2, \ldots, r$

$$\min_{1 \leq l \leq r} \left\| \boldsymbol{\Omega}\widehat{\mathbf{b}}_l - \mathbf{b}_k \right\|_2 = \min_{1 \leq l \leq r} \left\| \boldsymbol{\Omega}\widehat{\mathbf{d}}_{j_l} - \mathbf{b}_k \right\|_2 \leq \min_{1 \leq l \leq r} \left\| \mathbf{d}_{j_l} - \mathbf{b}_k \right\|_2 + Err$$

$$= \min_{1 \leq l \leq r} \left\| \mathcal{R}\mathbf{w}_{j_l} - \mathcal{R}\mathbf{e}_k \right\|_2 + Err \overset{(C .6)}{\leq} C^\star \sqrt{r} \min_{1 \leq l \leq r} \left\| \mathbf{w}_{j_l} - \mathbf{e}_k \right\|_2 + Err$$

$$\leq C^\star \sqrt{r} \min_{1 \leq l \leq r} \left\| \mathbf{w}_{j_l} - \mathbf{e}_k \right\|_1 + Err \leq C^\star \sqrt{r} \min_{1 \leq l \leq r} 2\left( 1 - \mathbf{w}_{j_l}(k) \right) + Err$$

$$\leq 2C^\star \sqrt{r} \left( 1 - \max_{1 \leq l \leq r} \mathbf{w}_{j_l}(k) \right) + Err \leq \left( 8C^\star (c^\star)^{-1} + 1 \right) Err.$$

$\square$

## C .3  Error Decomposition

**Theorem C .3.** *Under assumptions* (A .1) - (A .5), *there exist constants $c^* > 0$ and $C^* > 0$ such that if $Err < c^*$,*

$$\max_{1 \leq j \leq p} \left\| \widehat{\mathbf{w}}_j - \mathbf{w}_j \right\|_1 \leq C^* \cdot Err. \tag{C .9}$$

*If we further have $\left\| n^{-1}\mathbf{m} - \boldsymbol{\pi} \right\|_\infty \leq c_1 p^{-1}$, then*

$$\frac{1}{r} \sum_{k=1}^{p} \left\| (\widehat{\mathbf{V}} - \mathbf{V})\mathbf{e}_k \right\|_1 \leq C^* \left( Err + \sqrt{p} \cdot err + p \left\| n^{-1}\mathbf{m} - \boldsymbol{\pi} \right\|_\infty \right). \tag{C .10}$$

*Proof.* Suppose that $\omega \in \{\pm 1\}$ and $\boldsymbol{\Omega} \in \mathbb{O}^{(r-1) \times (r-1)}$ achieve the minima in definitions (C .1) and (C .2).

We first focus on the differences between $\widehat{\mathbf{w}}_j^*$ and $\mathbf{w}_j$ for $j = 1, 2, \ldots, p$. Note that

$$\widehat{\mathbf{w}}_j^* = \widehat{\mathbf{B}}^{-1} \begin{pmatrix} 1 \\ \boldsymbol{\Omega}\widehat{\mathbf{d}}_j \end{pmatrix}, \quad \mathbf{w}_j = \mathbf{B}^{-1} \begin{pmatrix} 1 \\ \mathbf{d}_j \end{pmatrix},$$

where

$$\widehat{\mathbf{B}} = \begin{pmatrix} 1 & \cdots & 1 \\ \boldsymbol{\Omega}\widehat{\mathbf{b}}_1 & \cdots & \boldsymbol{\Omega}\widehat{\mathbf{b}}_r \end{pmatrix},$$

$$\mathbf{B} = \begin{pmatrix} 1 & \cdots & 1 \\ \mathbf{b}_1 & \cdots & \mathbf{b}_r \end{pmatrix} = \mathbf{L}^\top [\mathrm{diag}(\mathbf{l}_1)]^{-1},$$

$$\begin{pmatrix} 1 \\ \mathbf{d}_j \end{pmatrix} = [\mathbf{h}_1(j)]^{-1} \mathbf{H}^\top \mathbf{e}_j.$$

We have

$$\begin{aligned}
\left\| \widehat{\mathbf{w}}_j^* - \mathbf{w}_j \right\|_2 &\leq \left\| \widehat{\mathbf{B}}^{-1} \right\|_2 \left\| \begin{pmatrix} 1 \\ \boldsymbol{\Omega}\widehat{\mathbf{d}}_j \end{pmatrix} - \begin{pmatrix} 1 \\ \mathbf{d}_j \end{pmatrix} \right\|_2 + \left\| \widehat{\mathbf{B}}^{-1} \begin{pmatrix} 1 \\ \mathbf{d}_j \end{pmatrix} - \mathbf{w}_j \right\|_2 \\
&= \left\| \widehat{\mathbf{B}}^{-1} \right\|_2 \left\| \boldsymbol{\Omega}\widehat{\mathbf{d}}_j - \mathbf{d}_j \right\|_2 + \left\| \widehat{\mathbf{B}}^{-1}\mathbf{B}\mathbf{w}_j - \mathbf{w}_j \right\|_2 \\
&\leq \left\| \widehat{\mathbf{B}}^{-1} \right\|_2 \left\| \boldsymbol{\Omega}\widehat{\mathbf{d}}_j - \mathbf{d}_j \right\|_2 + \left\| \widehat{\mathbf{B}}^{-1} \right\|_2 \left\| (\mathbf{B} - \widehat{\mathbf{B}})\mathbf{w}_j \right\|_2,
\end{aligned} \tag{C .11}$$

where

$$\left\| \boldsymbol{\Omega}\widehat{\mathbf{d}}_j - \mathbf{d}_j \right\|_2 \leq Err,$$

and

$$\begin{aligned}
\left\| (\widehat{\mathbf{B}} - \mathbf{B})\mathbf{w}_j \right\|_2 &= \left\| \sum_{k=1}^{r} \mathbf{w}_j(k)\left( \boldsymbol{\Omega}\widehat{\mathbf{b}}_k - \mathbf{b}_k \right) \right\|_2 \\
&\leq \sum_{k=1}^{r} \mathbf{w}_j(k) \left\| \boldsymbol{\Omega}\widehat{\mathbf{b}}_k - \mathbf{b}_k \right\|_2 \leq \max_{1 \leq k \leq r} \left\| \boldsymbol{\Omega}\widehat{\mathbf{b}}_k - \mathbf{b}_k \right\|_2.
\end{aligned} \tag{C .12}$$

Here we used $\mathbf{w}_j \geq 0$ and $\sum_{k=1}^{r} \mathbf{w}_j(k) = 1$.

We now derive an upper bound for $\left\| \widehat{\mathbf{B}}^{-1} \right\|_1$. According to Lemma A .3,

$$\left\| \mathbf{B}^{-1} \right\|_2 = \left\| [\mathrm{diag}(\mathbf{l}_1)]\mathbf{L}^{-\top} \right\|_2 \leq \left\| \mathbf{l}_1 \right\|_\infty \left\| \mathbf{L}^{-1} \right\|_2 \leq \widetilde{C} r^{-\frac{1}{2}}$$

for some constant $\widetilde{C} > 0$. In addition,

$$\begin{aligned}
\left\| \widehat{\mathbf{B}}^{-1} - \mathbf{B}^{-1} \right\|_2 &= \left\| \widehat{\mathbf{B}}^{-1}(\widehat{\mathbf{B}} - \mathbf{B})\mathbf{B}^{-1} \right\|_2 \leq \left\| \widehat{\mathbf{B}}^{-1} \right\|_2 \left\| \mathbf{B}^{-1} \right\|_2 \left\| \widehat{\mathbf{B}} - \mathbf{B} \right\|_2 \\
&\leq \left( \left\| \mathbf{B}^{-1} \right\|_2 + \left\| \widehat{\mathbf{B}}^{-1} - \mathbf{B}^{-1} \right\|_2 \right) \left\| \mathbf{B}^{-1} \right\|_2 \left\| \widehat{\mathbf{B}} - \mathbf{B} \right\|_2.
\end{aligned} \tag{C .13}$$

Lemma C .2 shows that if $Err \leq \alpha' \sqrt{r}$,

$$\max_{1 \leq k \leq r} \left\| \boldsymbol{\Omega}\widehat{\mathbf{b}}_k - \mathbf{b}_k \right\|_2 \leq \alpha \cdot Err,$$

therefore,

$$\left\|\widehat{\mathbf{B}} - \mathbf{B}\right\|_2 \le \sqrt{r} \max_{1 \le k \le r} \left\|\mathbf{\Omega}\widehat{\mathbf{b}}_k - \mathbf{b}_k\right\|_2 \le \alpha\sqrt{r} \cdot Err.$$

The inequality (C .13) can be reduced to

$$\left\|\widehat{\mathbf{B}}^{-1} - \mathbf{B}^{-1}\right\|_2 \le \left(\alpha\widetilde{C} \cdot Err\right)\left\|\widehat{\mathbf{B}}^{-1} - \mathbf{B}^{-1}\right\|_2 + \alpha\widetilde{C}^2 r^{-\frac{1}{2}} \cdot Err.$$

Under condition $Err \le \frac{1}{2}\alpha^{-1}\widetilde{C}^{-1}$, we further have

$$\begin{aligned}
\left\|\widehat{\mathbf{B}}^{-1} - \mathbf{B}^{-1}\right\|_2 &\le 2\alpha\widetilde{C}^2 r^{-\frac{1}{2}} \cdot Err, \\
\left\|\widehat{\mathbf{B}}^{-1}\right\|_2 &\le \left\|\mathbf{B}^{-1}\right\|_2 + \left\|\widehat{\mathbf{B}}^{-1} - \mathbf{B}^{-1}\right\|_2 \le 2\widetilde{C}r^{-\frac{1}{2}}.
\end{aligned} \tag{C .14}$$

Plugging (C .14) into (C .11) gives

$$\left\|\widehat{\mathbf{w}}_j^* - \mathbf{w}_j\right\|_2 \le \widetilde{C}'r^{-\frac{1}{2}} \cdot Err \tag{C .15}$$

for some constant $\widetilde{C}' > 0$.

Recall that

$$\widehat{\mathbf{W}} = [\widehat{\mathbf{w}}_1, \dots, \widehat{\mathbf{w}}_p]^\top$$

with

$$\widehat{\mathbf{w}}_j = \left\|\left[\widehat{\mathbf{w}}_j^*\right]_+\right\|_1^{-1}\left[\widehat{\mathbf{w}}_j^*\right]_+.$$

We have

$$\begin{aligned}
\left\|\widehat{\mathbf{w}}_j - \mathbf{w}_j\right\|_1 &\le \left\|\widehat{\mathbf{w}}_j - \left[\widehat{\mathbf{w}}_j^*\right]_+\right\|_1 + \left\|\left[\widehat{\mathbf{w}}_j^*\right]_+ - \mathbf{w}_j\right\|_1 \\
&\le \left|\left\|\left[\widehat{\mathbf{w}}_j^*\right]_+\right\|_1 - 1\right| + \left\|\left[\widehat{\mathbf{w}}_j^*\right]_+ - \mathbf{w}_j\right\|_1 \\
&\le 2\left\|\left[\widehat{\mathbf{w}}_j^*\right]_+ - \mathbf{w}_j\right\|_1 \le 2\left\|\widehat{\mathbf{w}}_j^* - \mathbf{w}_j\right\|_1 \\
&\le 2\sqrt{r}\left\|\widehat{\mathbf{w}}_j^* - \mathbf{w}_j\right\|_2 \le 2\widetilde{C}' \cdot Err.
\end{aligned} \tag{C .16}$$

Define

$$\widehat{\mathbf{V}}^\circ = [\text{diag}(\omega\widehat{\mathbf{h}}_1)][\text{diag}(n^{-1}\mathbf{m})]^{\frac{1}{2}}\widehat{\mathbf{W}},$$

and

$$\mathbf{V}^\circ = [\text{diag}(\mathbf{h}_1)][\text{diag}(\boldsymbol{\pi})]^{\frac{1}{2}}\mathbf{W} = \mathbf{V}[\text{diag}(\mathbf{l}_1)].$$

We calculate the row-wise $\ell_1$-distance between $\widehat{\mathbf{V}}^\circ$ and $\mathbf{V}^\circ$,

$$\begin{aligned}
\left\|\mathbf{e}_j^\top\left(\widehat{\mathbf{V}}^\circ - \mathbf{V}^\circ\right)\right\|_1 &= \left\|\omega\widehat{\mathbf{h}}_1(j)\sqrt{n^{-1}m_j} \cdot \widehat{\mathbf{w}}_j - \mathbf{h}_1(j)\sqrt{\pi_j} \cdot \mathbf{w}_j\right\|_1 \\
&\le \left|\omega\widehat{\mathbf{h}}_1(j) - \mathbf{h}_1(j)\right| \cdot \sqrt{n^{-1}m_j} \cdot \left\|\widehat{\mathbf{w}}_j\right\|_1 \\
&\quad + \left|\mathbf{h}_1(j)\right| \cdot \left|\sqrt{n^{-1}m_j} - \sqrt{\pi_j}\right| \cdot \left\|\widehat{\mathbf{w}}_j\right\|_1 \\
&\quad + \left|\mathbf{h}_1(j)\right| \cdot \sqrt{\pi_j} \cdot \left\|\widehat{\mathbf{w}}_j - \mathbf{w}_j\right\|_1.
\end{aligned} \tag{C .17}$$

Because $|\sqrt{x} - 1| \le |x - 1|$ for $x \ge 0$,

$$\begin{aligned}
\left|\sqrt{n^{-1}m_j} - \sqrt{\pi_j}\right| &= \sqrt{\pi_j}\left|\sqrt{n^{-1}m_j/\pi_j} - 1\right| \\
&\le \sqrt{\pi_j}\left|(n^{-1}m_j/\pi_j) - 1\right| = \pi_j^{-\frac{1}{2}}\left|n^{-1}m_j - \pi_j\right|.
\end{aligned}$$

Under the condition $\left\|n^{-1}\mathbf{m} - \boldsymbol{\pi}\right\|_\infty \le c_1 p^{-1}$,

$$\sqrt{n^{-1}m_j} \le \sqrt{\pi_j} + \left|\sqrt{n^{-1}m_j} - \sqrt{\pi_j}\right| \le 2\sqrt{\pi_j}.$$

We apply $\left\|\widehat{\mathbf{w}}_j\right\|_1 = 1$, $\left|\omega\widehat{\mathbf{h}}_1(j) - \mathbf{h}_1(j)\right| \le err$, (A .10) and (C .16) to (C .17), and obtain

$$\left\|\mathbf{e}_j^\top\left(\widehat{\mathbf{V}}^\circ - \mathbf{V}^\circ\right)\right\|_1 \le 2\sqrt{\pi_j} \cdot err + C^\#\left\|n^{-1}\mathbf{m} - \boldsymbol{\pi}\right\|_\infty + 2\widetilde{C}'C^\#\pi_j \cdot Err. \tag{C .18}$$

Here, $C^{\#} > 0$ is the constant in Lemma A .4.

In the last step, we normalize each column of $\widehat{\mathbf{V}}^{\circ}$ to get $\widehat{\mathbf{V}}$. We find that for $k = 1, 2, \ldots, r$,

$$
\begin{aligned}
\left\|(\widehat{\mathbf{V}} - \mathbf{V})\mathbf{e}_k\right\|_1 &= \left\|\left\|\widehat{\mathbf{V}}^{\circ}\mathbf{e}_k\right\|_1^{-1}\widehat{\mathbf{V}}^{\circ}\mathbf{e}_k - \left\|\mathbf{V}^{\circ}\mathbf{e}_k\right\|_1^{-1}\mathbf{V}^{\circ}\mathbf{e}_k\right\|_1 \\
&\leq \left|\left\|\widehat{\mathbf{V}}^{\circ}\mathbf{e}_k\right\|_1^{-1} - \left\|\mathbf{V}^{\circ}\mathbf{e}_k\right\|_1^{-1}\right| \cdot \left\|\widehat{\mathbf{V}}^{\circ}\mathbf{e}_k\right\|_1 + \left\|\mathbf{V}^{\circ}\mathbf{e}_k\right\|_1^{-1} \cdot \left\|(\widehat{\mathbf{V}}^{\circ} - \mathbf{V}^{\circ})\mathbf{e}_k\right\|_1 \\
&= \left|\left\|\widehat{\mathbf{V}}^{\circ}\mathbf{e}_k\right\|_1 - \left\|\mathbf{V}^{\circ}\mathbf{e}_k\right\|_1\right| \cdot \left\|\mathbf{V}^{\circ}\mathbf{e}_k\right\|_1^{-1} + \left\|\mathbf{V}^{\circ}\mathbf{e}_k\right\|_1^{-1} \cdot \left\|(\widehat{\mathbf{V}}^{\circ} - \mathbf{V}^{\circ})\mathbf{e}_k\right\|_1 \\
&\leq \left\|\widehat{\mathbf{V}}^{\circ}\mathbf{e}_k - \mathbf{V}^{\circ}\mathbf{e}_k\right\|_1 \cdot \left\|\mathbf{V}^{\circ}\mathbf{e}_k\right\|_1^{-1} + \left\|\mathbf{V}^{\circ}\mathbf{e}_k\right\|_1^{-1} \cdot \left\|(\widehat{\mathbf{V}}^{\circ} - \mathbf{V}^{\circ})\mathbf{e}_k\right\|_1 \\
&= 2\left\|\mathbf{V}^{\circ}\mathbf{e}_k\right\|_1^{-1} \cdot \left\|(\widehat{\mathbf{V}}^{\circ} - \mathbf{V}^{\circ})\mathbf{e}_k\right\|_1.
\end{aligned}
$$

Since by Lemma A .3, $\left\|\mathbf{V}^{\circ}\mathbf{e}_k\right\|_1 = \mathbf{l}_1(k) \geq c^{\#}r^{-1}$ for some constant $c^{\#} > 0$,

$$
\left\|(\widehat{\mathbf{V}} - \mathbf{V})\mathbf{e}_k\right\|_1 \leq 2(c^{\#})^{-1}r \cdot \left\|(\widehat{\mathbf{V}}^{\circ} - \mathbf{V}^{\circ})\mathbf{e}_k\right\|_1.
$$

We further derive from (C .18) that

$$
\begin{aligned}
\frac{1}{r}\sum_{k=1}^{r}\left\|(\widehat{\mathbf{V}} - \mathbf{V})\mathbf{e}_k\right\|_1 &\leq 2(c^{\#})^{-1}\sum_{k=1}^{r}\left\|(\widehat{\mathbf{V}}^{\circ} - \mathbf{V}^{\circ})\mathbf{e}_k\right\|_1 \\
&= 2(c^{\#})^{-1}\sum_{j=1}^{p}\left\|\mathbf{e}_j^{\top}(\widehat{\mathbf{V}}^{\circ} - \mathbf{V}^{\circ})\right\|_1 \leq \widetilde{C}''\left(\sqrt{p}\cdot err + p\left\|n^{-1}\mathbf{m} - \boldsymbol{\pi}\right\|_{\infty} + Err\right)
\end{aligned}
$$

for some constant $\widetilde{C}'' > 0$. $\qquad\square$

## C .4 Main Results

**Theorem C .4** (Statistical error bounds for $\mathbf{V}$, Theorem 3 in paper). *Under assumptions* (A .1) - (A .5), *for any $c_0 > 0$, there exists a constant $C > 0$ such that if $n \geq C\tau_* p^{\frac{3}{2}}r\left(\log^2(r) \vee 1\right)\log^2(n)$, then*

$$
\begin{aligned}
\mathbb{P}\Bigg(\frac{1}{r}\sum_{k=1}^{r}\left\|(\widehat{\mathbf{V}} - \mathbf{V})\mathbf{e}_k\right\|_1 \geq C\Big(&n^{-1/2}\sqrt{\tau_* pr\left(\log^2(r) \vee 1\right)\log^2(n)} \\
&+ n^{-1}\tau_* p^{\frac{3}{2}}r^{\frac{1}{2}}\left(\log^2(r) \vee 1\right)\log^2(n)\Big)\Bigg) \leq n^{-c_0}.
\end{aligned}
$$

*Proof.* According to Lemma B .3, B .2 and C .1, for a fixed $c_0 > 0$, there exists a constant $C_0 > 0$, $\omega \in \{\pm 1\}$ and $\boldsymbol{\Omega} \in \mathbb{O}^{(r-1)\times(r-1)}$ such that, when $n \geq C_0\tau_* p^{\frac{3}{2}}\left(\log^2(r) \vee 1\right)\log^2(n)$, with probability at least $1 - 3n^{-c_0}$,

$$
\begin{aligned}
Err &\leq \max_{1 \leq j \leq p}\left\|\boldsymbol{\Omega}\widehat{\mathbf{d}}_j - \mathbf{d}_j\right\|_2 \\
&\leq C_0\left(n^{-1/2}\sqrt{\tau_* pr\left(\log^2(r) \vee 1\right)\log^2(n)} + n^{-1}\tau_* p^{\frac{3}{2}}r^{\frac{1}{2}}\left(\log^2(r) \vee 1\right)\log^2(n)\right),
\end{aligned} \tag{C .19}
$$

$$
\begin{aligned}
err &\leq \max_{1 \leq j \leq p}\left|\omega\widehat{\mathbf{h}}_1(j) - \mathbf{h}_1(j)\right| \\
&\leq C_0\left(n^{-1/2}\sqrt{\tau_*\left(\log^2(r) \vee 1\right)\log^2(n)} + n^{-1}\tau_* p\left(\log^2(r) \vee 1\right)\log^2(n)\right),
\end{aligned} \tag{C .20}
$$

$$
\left\|n^{-1}\mathbf{m} - \boldsymbol{\pi}\right\|_{\infty} \leq C_0 n^{-1/2}\sqrt{\tau_* p^{-1}\log^2(n)}. \tag{C .21}
$$

We can take $\widetilde{C}_0 \geq C_0$ such that when $n \geq \widetilde{C}_0\tau_* p^{\frac{3}{2}}r\left(\log^2(r) \vee 1\right)\log^2(n)$,

$$
Err \leq c^*, \qquad \left\|n^{-1}\mathbf{m} - \boldsymbol{\pi}\right\|_{\infty} \leq c^*p^{-1}.
$$

Here, $c^*$ is the constant in Theorem C .3. Then plugging (C .19), (C .20) and (C .21) into (C .10), we complete the proof.

$\qquad\square$

**Theorem C .5** (Statistical error bounds for $\mathbf{U}$, Theorem 4 in paper). *Under assumptions* (A .1) -
(A .5), *for any $c_0 > 0$, there exists a constant $C > 0$ such that if $n \geq C\tau_* p^{\frac{3}{2}} r \left(\log^2(r) \vee 1\right) \log^2(n)$,
then*

$$
\mathbb{P}\bigg(\frac{1}{p}\sum_{j=1}^{p}\big\|\mathbf{e}_j^\top(\widehat{\mathbf{U}} - \mathbf{U})\big\|_1 \geq Cr^{\frac{3}{2}}\Big(n^{-1/2}\sqrt{\tau_* pr\left(\log^2(r) \vee 1\right)\log^2(n)}
$$
$$
+ n^{-1}\tau_* p^{\frac{3}{2}} r^{\frac{1}{2}}\left(\log^2(r) \vee 1\right)\log^2(n)\Big)\bigg) \leq n^{-c_0}.
\tag{C .22}
$$

**Remark.** Write for short

$$
\widetilde{err}_n = C\Big(n^{-1/2}\sqrt{\tau_* pr\left(\log^2(r) \vee 1\right)\log^2(n)} + n^{-1}\tau_* p^{\frac{3}{2}} r^{\frac{1}{2}}\left(\log^2(r) \vee 1\right)\log^2(n)\Big).
$$

Below are a few alternative expressions for (C .22):

With probability at least $1 - n^{-c_0}$,

- $p^{-\frac{1}{2}}\big\|\widehat{\mathbf{U}} - \mathbf{U}\big\|_2 \leq \sqrt{r} \cdot err_n$,

- $\sqrt{p^{-1}\sum_{j=1}^{p}\big\|\widehat{\mathbf{u}}_j - \mathbf{u}_j\big\|_2^2} \leq r \cdot err_n$,

- $p^{-1}\sum_{j=1}^{p}\big\|\widehat{\mathbf{u}}_j - \mathbf{u}_j\big\|_1 \leq r^{\frac{3}{2}} \cdot err_n$.

*Proof.* By definition,

$$
\widehat{\mathbf{U}} = [\mathrm{diag}(n^{-1}\mathbf{m})]^{-1}(n^{-1}\widehat{\mathbf{N}})\widehat{\mathbf{V}}(\widehat{\mathbf{V}}^\top\widehat{\mathbf{V}})^{-1},
$$
$$
\mathbf{U} = [\mathrm{diag}(\boldsymbol{\pi})]^{-1}\mathbf{F}\mathbf{V}(\mathbf{V}^\top\mathbf{V})^{-1}.
$$

We need some preparations. First, consider the diagonal matrix $[\mathrm{diag}(n^{-1}\mathbf{m})]^{-1}$. The assumption
(A .1) guarantees $\pi_j \geq c_1 p^{-1}$. By (B .21), with probability $1 - n^{-c_0}$, $\|n^{-1}\mathbf{m} - \boldsymbol{\pi}\|_\infty \leq Cp^{-1}r^{-\frac{1}{2}} \cdot$
$\widetilde{err}_n$. It follows that $n^{-1}m_j \geq \pi_j - \|n^{-1}\mathbf{m} - \boldsymbol{\pi}\|_\infty \geq c_1 p^{-1}/2$. Therefore,

$$
\big\|[\mathrm{diag}(\boldsymbol{\pi})]^{-1}\big\|_2 \leq Cp, \quad \big\|[\mathrm{diag}(n^{-1}\widehat{\mathbf{N}})]^{-1}\big\|_2 \leq Cp,
\tag{C .23}
$$

and

$$
\big\|[\mathrm{diag}(n^{-1}\widehat{\mathbf{N}})]^{-1} - [\mathrm{diag}(\boldsymbol{\pi})]^{-1}\big\|_2
$$
$$
\leq\big\|[\mathrm{diag}(\tfrac{1}{n}\widehat{\mathbf{N}})]^{-1}\big\|_2\big\|\mathrm{diag}(\tfrac{1}{n}\widehat{\mathbf{N}}) - \mathrm{diag}(\boldsymbol{\pi})\big\|_2\big\|[\mathrm{diag}(\boldsymbol{\pi})]^{-1}\big\|
$$
$$
\leq Cpr^{-\frac{1}{2}} \cdot \widetilde{err}_n.
\tag{C .24}
$$

Second, consider the matrix $n^{-1}\widehat{\mathbf{N}}$. By (B .21), with probability $1 - n^{-c_0}$,

$$
\|n^{-1}\widehat{\mathbf{N}} - \mathbf{F}\|_2 \leq Cp^{-1}r^{-\frac{1}{2}} \cdot \widetilde{err}_n.
\tag{C .25}
$$

Additionally, by (E .3), $\|\mathbf{F}\|_2 \leq Cp^{-1}$. Combining it with (C .25) gives

$$
\|\mathbf{F}\|_2 \leq Cp^{-1}, \qquad \|n^{-1}\widehat{\mathbf{N}}\|_2 \leq Cp^{-1}.
\tag{C .26}
$$

Next, consider the matrix $\widehat{\mathbf{V}}$. By (E .9) and the assumption (A .1), $\|\mathbf{e}_j^\top\mathbf{V}\|_1 \leq Cp^{-1}r$ for all $1 \leq j \leq$
$p$. It follows that $\|\mathbf{V}\|_1 = \max_{1 \leq k \leq r}\|\mathbf{V}\mathbf{e}_k\|_1 = 1$, and $\|\mathbf{V}\|_\infty = \max_{1 \leq j \leq p}\|\mathbf{e}_j^\top\mathbf{V}\|_1 \leq Cp^{-1}r$.
As a result,

$$
\|\mathbf{V}\|_2 \leq \sqrt{\|\mathbf{V}\|_1\|\mathbf{V}\|_\infty} \leq Cp^{-\frac{1}{2}}r^{\frac{1}{2}}.
\tag{C .27}
$$

By Theorem C .4, with probability $1 - n^{-c_0}$, $\sum_{k=1}^{r}\|(\widehat{\mathbf{V}} - \mathbf{V})\mathbf{e}_k\|_1 \leq Cr \cdot \widetilde{err}_n$. It immediately
gives

$$
\|\widehat{\mathbf{V}} - \mathbf{V}\|_1 = \max_{1 \leq k \leq r}\|(\widehat{\mathbf{V}} - \mathbf{V})\mathbf{e}_k\|_1 \leq Cr \cdot \widetilde{err}_n.
\tag{C .28}
$$

We then bound $\|\widehat{\mathbf{V}} - \mathbf{V}\|_\infty$. Let $\mathbf{V}^\circ$ and $\widehat{\mathbf{V}}^\circ$ be the same as in (C .18). We have seen in (C .18) that, with probability $1 - n^{-c_0}$, $\|\mathbf{e}_j^\top (\widehat{\mathbf{V}}^\circ - \mathbf{V}^\circ)\|_1 \leq Cp^{-1}\widetilde{err}_n$. It follows that

$$\sum_{k=1}^{r}\big\|\widehat{\mathbf{V}}^\circ\mathbf{e}_k - \mathbf{V}^\circ\mathbf{e}_k\big\|_1 = \sum_{j=1}^{p}\|\mathbf{e}_j^\top(\widehat{\mathbf{V}}^\circ - \mathbf{V}^\circ)\|_1 \leq C\widetilde{err}_n. \tag{C .29}$$

Additionally, by Lemma A .3, $\|\mathbf{V}^\circ\mathbf{e}_k\|_1 = \mathbf{l}_1(k) \geq C^{-1}r^{-1}$; as a result, $\|\widehat{\mathbf{V}}^\circ\mathbf{e}_k\|_1 \geq \|\mathbf{V}^\circ\mathbf{e}_k\|_1 - C\widetilde{err}_n \geq C^{-1}r^{-1}$. It is seen that, for each $1 \leq j \leq p$,

$$\begin{aligned}
\big|\mathbf{e}_j^\top(\widehat{\mathbf{V}} - \mathbf{V})\mathbf{e}_k\big| &= \Big|\|\widehat{\mathbf{V}}^\circ\mathbf{e}_k\|_1^{-1}\mathbf{e}_j^\top\widehat{\mathbf{V}}^\circ\mathbf{e}_k - \|\mathbf{V}^\circ\mathbf{e}_k\|_1^{-1}\mathbf{e}_j^\top\mathbf{V}^\circ\mathbf{e}_k\Big| \\
&\leq\Big|\|\widehat{\mathbf{V}}^\circ\mathbf{e}_k\|_1^{-1} - \|\mathbf{V}^\circ\mathbf{e}_k\|_1^{-1}\Big|\cdot\big|\mathbf{e}_j^\top\widehat{\mathbf{V}}^\circ\mathbf{e}_k\big| + \|\mathbf{V}^\circ\mathbf{e}_k\|_1^{-1}\cdot\big|\mathbf{e}_j^\top(\widehat{\mathbf{V}}^\circ - \mathbf{V}^\circ)\mathbf{e}_k\big| \\
&=\|\mathbf{V}^\circ\mathbf{e}_k\|_1^{-1}\Big|\|\widehat{\mathbf{V}}^\circ\mathbf{e}_k\|_1 - \|\mathbf{V}^\circ\mathbf{e}_k\|_1\Big|\cdot\|\widehat{\mathbf{V}}^\circ\mathbf{e}_k\|_1^{-1}\big|\mathbf{e}_j^\top\widehat{\mathbf{V}}^\circ\mathbf{e}_k\big|, \\
&\quad + \|\mathbf{V}^\circ\mathbf{e}_k\|_1^{-1}\cdot\big|\mathbf{e}_j^\top(\widehat{\mathbf{V}}^\circ - \mathbf{V}^\circ)\mathbf{e}_k\big| \\
&\leq Cr\cdot\big\|\widehat{\mathbf{V}}^\circ\mathbf{e}_k - \mathbf{V}^\circ\mathbf{e}_k\big\|_1\cdot Cr\cdot\big\|\mathbf{e}_j^\top\widehat{\mathbf{V}}^\circ\big\|_1 + Cr\cdot\big|\mathbf{e}_j^\top(\widehat{\mathbf{V}}^\circ - \mathbf{V}^\circ)\mathbf{e}_k\big|.
\end{aligned}$$

Summing over $k$ on both sides gives

$$\begin{aligned}
\big\|\mathbf{e}_j^\top(\widehat{\mathbf{V}} - \mathbf{V})\big\|_1 &\leq Cr^2\big\|\mathbf{e}_j^\top\widehat{\mathbf{V}}^\circ\big\|_1\cdot\sum_{k=1}^{r}\big\|\widehat{\mathbf{V}}^\circ\mathbf{e}_k - \mathbf{V}^\circ\mathbf{e}_k\big\|_1 + Cr\cdot\big\|\mathbf{e}_j^\top(\widehat{\mathbf{V}}^\circ - \mathbf{V}^\circ)\big\|_1 \\
&\leq Cr^2\cdot\big\|\mathbf{e}_j^\top\widehat{\mathbf{V}}^\circ\big\|_1\cdot C\widetilde{err}_n + Cr\cdot p^{-1}\widetilde{err}_n,
\end{aligned}$$

where the last inequality is from (C .18) and (C .29). Since $\mathbf{V}^\circ = \mathbf{V}[\mathrm{diag}(\mathbf{l}_1)]$, we have $\|\mathbf{e}_j^\top\mathbf{V}^\circ\|_1 \leq \|\mathbf{e}_j^\top\mathbf{V}\|_1\|\mathbf{l}_1\|_\infty$. By Lemma A .3, $\|\mathbf{l}\|_\infty \leq Cr^{-1}$; by (E .9) and the assumption (A .1), $\|\mathbf{e}_j^\top\mathbf{V}\|_1 \leq Cp^{-1}r$. Hence, $\|\mathbf{e}_j^\top\mathbf{V}^\circ\|_1 \leq Cp^{-1}$. Plugging it into the above inequality gives

$$\big\|\mathbf{e}_j^\top(\widehat{\mathbf{V}} - \mathbf{V})\big\|_1 \leq Cp^{-1}r^2\cdot\widetilde{err}_n.$$

It follows that
$$\|\widehat{\mathbf{V}} - \mathbf{V}\|_\infty \leq Cp^{-1}r^2\cdot\widetilde{err}_n. \tag{C .30}$$

Combing (C .28) and (C .30) gives

$$\|\widehat{\mathbf{V}} - \mathbf{V}\|_2 \leq \sqrt{\|\widehat{\mathbf{V}} - \mathbf{V}\|_1\|\widehat{\mathbf{V}} - \mathbf{V}\|_\infty} \leq Cp^{-\frac{1}{2}}r^{\frac{3}{2}}\cdot\widetilde{err}_n. \tag{C .31}$$

Last, we study the matrix $(\widehat{\mathbf{V}}^\top\widehat{\mathbf{V}})^{-1}$. Since $(\widehat{\mathbf{V}}^\top\widehat{\mathbf{V}} - \mathbf{V}^\top\mathbf{V})$ is a symmetric matrix,

$$\begin{aligned}
\|\widehat{\mathbf{V}}^\top\widehat{\mathbf{V}} - \mathbf{V}^\top\mathbf{V}\|_2 &\leq \|\widehat{\mathbf{V}}^\top\widehat{\mathbf{V}} - \mathbf{V}^\top\mathbf{V}\|_1 \\
&\leq\|\mathbf{V}^\top(\widehat{\mathbf{V}} - \mathbf{V})\|_1 + \|(\widehat{\mathbf{V}} - \mathbf{V})^\top\mathbf{V}\|_1 + \|(\widehat{\mathbf{V}} - \mathbf{V})^\top(\widehat{\mathbf{V}} - \mathbf{V})\|_1 \\
&\leq\|\mathbf{V}\|_\infty\|\widehat{\mathbf{V}} - \mathbf{V}\|_1 + \|\widehat{\mathbf{V}} - \mathbf{V}\|_\infty\|\mathbf{V}\|_1 + \|\widehat{\mathbf{V}} - \mathbf{V}\|_\infty\|\widehat{\mathbf{V}} - \mathbf{V}\|_1 \\
&\leq(Cp^{-1}r)(Cr\cdot\widetilde{err}_n) + (Cp^{-1}r^2\cdot\widetilde{err}_n)\cdot 1 + (Cr\cdot\widetilde{err}_n)(Cp^{-1}r^2\cdot\widetilde{err}_n) \\
&\leq Cp^{-1}r^2\cdot\widetilde{err}_n.
\end{aligned}$$

By the assumption (A .6), $\lambda_{\min}(\mathbf{V}^\top\mathbf{V}) \geq C^{-1}p^{-1}r$. It further implies that $\lambda_{\min}(\widehat{\mathbf{V}}^\top\widehat{\mathbf{V}}) \geq \lambda_{\min}(\mathbf{V}^\top\mathbf{V}) - \|\widehat{\mathbf{V}}^\top\widehat{\mathbf{V}} - \mathbf{V}^\top\mathbf{V}\|_2 \geq C^{-1}p^{-1}r$. In other words,

$$\big\|(\mathbf{V}^\top\mathbf{V})^{-1}\big\|_2 \leq Cpr^{-1}, \quad \big\|(\widehat{\mathbf{V}}^\top\widehat{\mathbf{V}})^{-1}\big\|_2 \leq Cpr^{-1}. \tag{C .32}$$

Furthermore,

$$\begin{aligned}
&\big\|(\mathbf{V}^\top\mathbf{V})^{-1} - (\widehat{\mathbf{V}}^\top\widehat{\mathbf{V}})^{-1}\big\|_2 \\
&\leq\big\|(\widehat{\mathbf{V}}^\top\widehat{\mathbf{V}})^{-1}\big\|_2\|\widehat{\mathbf{V}}^\top\widehat{\mathbf{V}} - \mathbf{V}^\top\mathbf{V}\|_2\big\|(\mathbf{V}^\top\mathbf{V})^{-1}\big\|_2 \\
&\leq Cp\cdot\widetilde{err}_n. \tag{C .33}
\end{aligned}$$

We now proceed to proving the claim. Using the triangular inequality,

$$\|\widehat{\mathbf{U}} - \mathbf{U}\|_2$$
$$\leq \left\|[\mathrm{diag}(n^{-1}\widehat{\mathbf{N}})]^{-1} - [\mathrm{diag}(\boldsymbol{\pi})]^{-1}\right\|_2 \|n^{-1}\widehat{\mathbf{N}}\|_2 \|\widehat{\mathbf{V}}\|_2 \|(\widehat{\mathbf{V}}^\top \widehat{\mathbf{V}})^{-1}\|_2$$
$$+ \left\|[\mathrm{diag}(\boldsymbol{\pi})]^{-1}\right\|_2 \|n^{-1}\widehat{\mathbf{N}} - \mathbf{F}\|_2 \|\widehat{\mathbf{V}}\|_2 \|(\widehat{\mathbf{V}}^\top \widehat{\mathbf{V}})^{-1}\|_2$$
$$+ \left\|[\mathrm{diag}(\boldsymbol{\pi})]^{-1}\right\|_2 \|\mathbf{F}\|_2 \|\widehat{\mathbf{V}} - \mathbf{V}\|_2 \|(\widehat{\mathbf{V}}^\top \widehat{\mathbf{V}})^{-1}\|_2$$
$$+ \left\|[\mathrm{diag}(\boldsymbol{\pi})]^{-1}\right\|_2 \|\mathbf{F}\|_2 \|\mathbf{V}\|_2 \|(\widehat{\mathbf{V}}^\top \widehat{\mathbf{V}})^{-1} - (\mathbf{V}^\top \mathbf{V})^{-1}\|_2$$
$$\leq C(pr^{-\frac{1}{2}}\widetilde{err}_n) \cdot p^{-1} \cdot p^{-\frac{1}{2}}r^{\frac{1}{2}} \cdot pr^{-1}$$
$$+ Cp \cdot (p^{-1}r^{-\frac{1}{2}}\widetilde{err}_n) \cdot p^{-\frac{1}{2}}r^{\frac{1}{2}} \cdot pr^{-1}$$
$$+ Cp \cdot p^{-1} \cdot (p^{-\frac{1}{2}}r^{\frac{3}{2}}\widetilde{err}_n) \cdot pr^{-1}$$
$$+ Cp \cdot p^{-1} \cdot p^{-\frac{1}{2}}r^{\frac{1}{2}} \cdot (p\,\widetilde{err}_n)$$
$$\leq C\sqrt{pr} \cdot \widetilde{err}_n.$$

It follows that

$$p^{-1}\sum_{j=1}^{p} \|\mathbf{e}_j^\top (\widehat{\mathbf{U}} - \mathbf{U})\|_2^2 = p^{-1}\|\widehat{\mathbf{U}} - \mathbf{U}\|_F^2 \leq p^{-1} \cdot (2r) \cdot \|\widehat{\mathbf{U}} - \mathbf{U}\|_2^2 \leq Cr^2 \cdot \widetilde{err}_n.$$

$\square$

**Theorem C .6** (Recovery of anchor states, Theorem 5 in paper). *Let $\mathcal{N}$ be the set of non-anchor states. Denote*

$$\phi = \max_{j\in\mathcal{N}} \max_{1\leq k\leq r} \mathbb{P}_{X_0\sim\boldsymbol{\pi}}\big(Z_t = k \mid X_{t+1} = j\big).$$

*Under conditions (A .1) - (A .5), for a fixed $c_0 > 0$, there exist constants $c > 0$ and $C > 0$ such that when $\delta_0 \leq c(1 - \phi)$ and*

$$n \geq C\delta_0^{-2}\tau_* p^{\frac{3}{2}} r\big(\log^2(r) \vee 1\big)\log^2(n),$$

*we can successfully identify the anchor states with probability at least $1 - n^{-c_0}$.*

*Proof.* Define

$$\boldsymbol{\zeta}_j = \pi_j^{-1}\big[\mathrm{diag}(\mathbf{U}^\top\boldsymbol{\pi})\big]\mathbf{V}^\top\mathbf{e}_j, \quad j = 1, 2, \ldots, p.$$

Then for $k = 1, 2, \ldots, r$,

$$\boldsymbol{\zeta}_j(k) = \mathbb{P}_{X_0\sim\boldsymbol{\pi}}\big(Z_t = k \mid X_{t+1} = j\big).$$

We first present a useful fact that, for any two vectors $\mathbf{x}, \mathbf{x}'$ in the $r$-dimensional standard simplex $\mathcal{S}_{r-1}$, by triangle inequality,

$$\big\|\mathbf{x}' - \mathbf{x}\big\|_1 = \big|\mathbf{x}'(k) - \mathbf{x}(k)\big| + \sum_{l\neq k}\big|\mathbf{x}'(l) - \mathbf{x}(l)\big|$$
$$\geq \big|\mathbf{x}'(k) - \mathbf{x}(k)\big| + \Big|\sum_{l\neq k}\big(\mathbf{x}'(l) - \mathbf{x}(l)\big)\Big| = 2\big|\mathbf{x}'(k) - \mathbf{x}(k)\big|, \quad k = 1, 2, \ldots, r.$$
(C .34)

If $\mathbf{x} = \mathbf{e}_k$ then the equality holds. According to (C .34), the parameter $\phi$ in the theorem can be equivalently defined as

$$1 - \phi = \min_{j\in\mathcal{N}} \min_{1\leq k\leq r}\big(1 - \boldsymbol{\zeta}_j(k)\big) = \frac{1}{2}\min_{j\in\mathcal{N}} \min_{1\leq k\leq r}\big\|\boldsymbol{\zeta}_j - \mathbf{e}_k\big\|_1.$$

Denote

$$\mathbf{A} = \big[\mathrm{diag}(\mathbf{U}^\top\boldsymbol{\pi})\big]\big[\mathrm{diag}(\mathbf{l}_1)\big]^{-1}.$$

Since

$$\mathbf{w}_j = \pi_j^{-\frac{1}{2}}[\mathbf{h}_1(j)]^{-1}[\mathrm{diag}(\mathbf{l}_1)]\mathbf{V}^\top\mathbf{e}_j,$$

we have $\boldsymbol{\zeta}_j = \|\mathbf{Aw}_j\|_1^{-1} \mathbf{Aw}_j$ for $j = 1, 2, \ldots, p$. By (E.8), Lemma A.3 and assumption (A.2), there exist constants $\widetilde{c} > 0$ and $\widetilde{C} > 0$ such that the diagonal entries of $\mathbf{A}$ satisfies

$$\widetilde{c} \le \left(\mathbf{U}^\top \boldsymbol{\pi}\right)_k \left[\mathbf{l}_1(k)\right]^{-1} \le \widetilde{C}, \quad \text{for } k = 1, 2, \ldots, r.$$

We first derive a lower bound for $\|\mathbf{w}_j - \mathbf{e}_k\|_1$, $j \in \mathcal{N}$, using $\phi$. For any $\mathbf{w}, \mathbf{w}' \in \mathcal{S}_{r-1}$, let $\boldsymbol{\zeta} = \|\mathbf{Aw}\|_1^{-1} \mathbf{Aw}$, $\boldsymbol{\zeta}' = \|\mathbf{Aw}'\|_1^{-1} \mathbf{Aw}'$. Because $\mathbf{w}, \mathbf{w}' \in \mathcal{S}_{r-1}$,

$$\|\mathbf{Aw}\|_1 \ge \widetilde{c}, \qquad \|\mathbf{Aw}'\|_1 \ge \widetilde{c}, \qquad \|\mathbf{A}(\mathbf{w} - \mathbf{w}')\|_1 \le \widetilde{C}\|\mathbf{w} - \mathbf{w}'\|_1.$$

We find that

$$
\begin{aligned}
\|\boldsymbol{\zeta} - \boldsymbol{\zeta}'\|_1 &= \left\|\|\mathbf{Aw}\|_1^{-1}\mathbf{Aw} - \|\mathbf{Aw}'\|_1^{-1}\mathbf{Aw}'\right\|_1 \\
&\le \|\mathbf{Aw}\|_1^{-1}\|\mathbf{A}(\mathbf{w} - \mathbf{w}')\|_1 + \left|\|\mathbf{Aw}\|_1^{-1} - \|\mathbf{Aw}'\|_1^{-1}\right| \cdot \|\mathbf{Aw}'\|_1
\end{aligned}
$$

Here, the second term

$$
\begin{aligned}
\left|\|\mathbf{Aw}\|_1^{-1} - \|\mathbf{Aw}'\|_1^{-1}\right| \cdot \|\mathbf{Aw}'\|_1 &\le \left|\|\mathbf{Aw}\|_1 - \|\mathbf{Aw}\|_1\right| \cdot \|\mathbf{Aw}\|_1^{-1} \\
&\le \|\mathbf{Aw}\|_1^{-1}\|\mathbf{A}(\mathbf{w} - \mathbf{w}')\|_1.
\end{aligned}
$$

Therefore,

$$\|\boldsymbol{\zeta} - \boldsymbol{\zeta}'\|_1 \le 2\|\mathbf{Aw}\|_1^{-1}\|\mathbf{A}(\mathbf{w} - \mathbf{w}')\|_1 \le 2\widetilde{C}\widetilde{c}^{-1}\|\mathbf{w} - \mathbf{w}'\|_1, \tag{C.35}$$

and

$$\min_{j \in \mathcal{N}} \min_{1 \le k \le r} \|\mathbf{w}_j - \mathbf{e}_k\|_1 \ge \frac{1}{2}\widetilde{C}^{-1}\widetilde{c} \min_{j \in \mathcal{N}} \min_{1 \le k \le r} \|\boldsymbol{\zeta}_j - \mathbf{e}_k\|_1 = \widetilde{C}^{-1}\widetilde{c}(1 - \phi). \tag{C.36}$$

We now consider the perturbation bound for $\widehat{\mathbf{w}}_j$. According to Lemma C.1, for a fixed $c_0 > 0$, there exists a constant $C_0 > 0$ such that if $n \ge C_0 \tau_* p^{\frac{3}{2}} \left(\log^2(r) \vee 1\right) \log^2(n)$, with probability at least $1 - n^{-c_0}$,

$$
\begin{aligned}
Err \le C_0 \Big(&n^{-1/2}\sqrt{\tau_* pr\left(\log^2(r) \vee 1\right)\log^2(n)} \\
&+ n^{-1}\tau_* p^{\frac{3}{2}}r^{\frac{1}{2}}\left(\log^2(r) \vee 1\right)\log^2(n)\Big).
\end{aligned}
$$

We further take $n > (c^*)^{-2}(C_0^{\frac{1}{2}} + 1)^{-1}C_0\tau_* p^{\frac{3}{2}}r\left(\log^2(r) \vee 1\right)\log^2(n)$, then $Err < c^*$, where $c^*$ is the constant in Theorem C.3. Theorem C.3 then implies that

$$\max_{1 \le j \le p}\|\widehat{\mathbf{w}}_j - \mathbf{w}_j\|_1 \le C^* \cdot Err.$$

There exists a constant $\widetilde{C}_0 \ge (c^*)^{-2}(C_0^{\frac{1}{2}} + 1)^{-1}C_0$ such that when

$$n \ge \widetilde{C}_0 \delta_0^{-2}\tau_* p^{\frac{3}{2}}r\left(\log^2(r) \vee 1\right)\log^2(n),$$

we have

$$\max_{1 \le j \le p}\|\widehat{\mathbf{w}}_j - \mathbf{w}_j\|_1 \le 2\delta_0. \tag{C.37}$$

Suppose that $j$ is an anchor state for meta-state $k$. Then $\mathbf{w}_j = \mathbf{e}_k$. Under (C.37), we use (C.34) and obtain

$$\widehat{\mathbf{w}}_j(k) = 1 - \left(1 - \widehat{\mathbf{w}}_j(k)\right) = 1 - \frac{1}{2}\|\widehat{\mathbf{w}}_j - \mathbf{w}_j\|_1 \ge 1 - \delta_0.$$

Consider the case where $j \in \mathcal{N}$. Suppose that $\delta_0 \le 4^{-1}\widetilde{C}^{-1}\widetilde{c}(1 - \phi)$. Then by (C.36), for $k = 1, 2, \ldots, r$,

$$1 - \mathbf{w}_j(k) = \frac{1}{2}\|\mathbf{w}_j - \mathbf{e}_k\|_1 \ge \frac{1}{2}\widetilde{C}^{-1}\widetilde{c}(1 - \phi) \ge 2\delta_0.$$

It follows from (C.37) that

$$
\begin{aligned}
\widehat{\mathbf{w}}_j(k) &\le 1 - \left(1 - \mathbf{w}_j(k)\right) + \left|\widehat{\mathbf{w}}_j(k) - \mathbf{w}_j(k)\right| \\
&\le 1 - 2\delta_0 + \frac{1}{2}\|\widehat{\mathbf{w}}_j - \mathbf{w}_j\|_1 \le 1 - \delta_0.
\end{aligned}
$$

$\square$

# D  Explanation of Main Algorithm

In this section, we explain the rationale of Algorithm 1, especially for:

- Why the SCORE normalization [3] produces a simplex geometry.
- How the simplex geometry is used for estimating $\mathbf{V}$.

Without loss of genrality, we assume that all entires of the stationary distribution $\boldsymbol{\pi} \in \mathbb{R}^p$ are positive. The normalized data matrix

$$\widetilde{\mathbf{N}} \approx n^{\frac{1}{2}} \mathrm{diag}(\boldsymbol{\pi}) \mathbf{P} [\mathrm{diag}(\boldsymbol{\pi})]^{-1/2} \equiv n^{\frac{1}{2}} \mathbf{Q}. \tag{D.1}$$

The matrix $\mathbf{Q}$ can be viewed as the "signal" part of $\widetilde{\mathbf{N}}$. Let $\mathbf{h}_1, \ldots, \mathbf{h}_r$ be the right singular vectors of $\mathrm{diag}(\boldsymbol{\pi}) \mathbf{P} [\mathrm{diag}(\boldsymbol{\pi})]^{-1/2}$. They can be viewed as the population counterpart of $\widehat{\mathbf{h}}_1, \ldots, \widehat{\mathbf{h}}_r$. We define a population counterpart of the matrix $\widehat{\mathbf{D}}$ produced by SCORE:

$$\mathbf{D} = [\mathrm{diag}(\mathbf{h}_1)]^{-1}[\mathbf{h}_2, \ldots, \mathbf{h}_r] = [\mathbf{d}_1, \mathbf{d}_2, \ldots, \mathbf{d}_p]^\top. \tag{D.2}$$

From now on, we pretend that the matrix $\mathbf{Q}$ is directly given and study the geometric structures associated with the singular vectors and the SCORE matrix $\mathbf{D}$.

## D.1  The Simplex Geometry and Explanation of Steps of Algorithm 1

When $\mathbf{P} = \mathbf{U}\mathbf{V}^\top$, the matrix $\mathbf{Q}$, defined in (D.1), also admits a low-rank decomposition:

$$\mathbf{Q} = \mathbf{U}^*(\mathbf{V}^*)^\top,$$

where

$$\mathbf{U}^* = [\mathrm{diag}(\boldsymbol{\pi})]\mathbf{U}, \qquad \mathbf{V}^* = [\mathrm{diag}(\boldsymbol{\pi})]^{-1/2}\mathbf{V}.$$

The span of the right singular vectors $\mathbf{h}_1, \ldots, \mathbf{h}_r$ is the same as the column space of $\mathbf{V}^*$. It implies there exists a linear transformation $\mathbf{L} \in \mathbb{R}^{r \times r}$ such that

$$\mathbf{H} \equiv [\mathbf{h}_1, \ldots, \mathbf{h}_r] = \mathbf{V}^*\mathbf{L}. \tag{D.3}$$

Since $\mathbf{V}^*$ is a nonnegative matrix, each row of $\mathbf{H}$ is an affine combination of rows of $\mathbf{L}$. Furthermore, if $j$ is an anchor state, then the $j$-th row of $\mathbf{V}^*$ has exactly one nonzero entry, and so the $j$-th row of $\mathbf{H}$ is proportional to one row of $\mathbf{L}$. This gives rise to the following simplicial-cone geometry:

**Proposition 1** (Simplicial cone geometry). *Suppose* $\mathbf{P} = \mathbf{U}\mathbf{V}^\top$, *each meta-state has an anchor state, and* $\mathrm{rank}(\mathbf{U}) = r$. *Let* $\mathbf{H} = [\mathbf{h}_1, \ldots, \mathbf{h}_r]$ *contain the right singular vectors of* $\mathbf{Q} = \mathrm{diag}(\boldsymbol{\pi}) \mathbf{P} [\mathrm{diag}(\boldsymbol{\pi})]^{-1/2}$. *There exists a simplicial cone in* $\mathbb{R}^r$, *which has* $r$ *extreme rays, such that all rows of* $\mathbf{H}$ *are contained in this simplicial cone. Furthermore, for all anchor states* $j$ *of a meta-state, the* $j$-th *row of* $\mathbf{H}$ *lies exactly on one extreme ray of this simplicial cone.*

**Remark**. Similar simplicial-cone geometry has been discovered in the literature of nonnegative matrix factorization [2]. The simplicial cone there is associated with rows of the matrix that admits a nonnegative factorization, but the simplicial cone here is associated with singular vectors of the matrix. Since SVD is a linear projection, it is not surprising that the simplicial cone structure is retained in singular vectors.

However, in the real case, we have to apply SVD to the noisy matrix, then the simplicial cone is corrupted by noise and hardly visible. We hope to find a proper normalization of $\mathbf{H}$, so that the normalized rows are all contained in a simplex, where all points on the extreme ray of the previous simplicial cone (these points do not overlap) fall onto one vertex of the current simplex (these points now overlap). Such a simplex geometry is much more robust to noise corruption and is easier to estimate.

How to normalize $\mathbf{H}$ to obtain a simplex geometry is tricky. If all entries of $\mathbf{H}$ are nonnegative, we can normalize each row of $\mathbf{H}$ by the $\ell^1$-norm of that row, and rows of the resulting matrix are contained in a simplex. However, $\mathbf{H}$ consists of singular vectors and often has negative entries, so such a normalization doesn't work.

By Perron-Frobenius theorem in linear algebra, the leading right singular vector $\mathbf{h}_1$ have all positive coordinates. It turns out that normalizing each row of $\mathbf{H}$ by the corresponding coordinate of $\mathbf{h}_1$ is a proper normalizaiton that will produce a simplex geometry. This is the idea of SCORE [3, 4, 5]. See Figure 2 in the paper for illustration.

**Proposition 2** (Post-SCORE simplex geometry). *In the setting of Proposition 1, additionally, we assume $\mathbf{h}_1$ have all positive coordinates (e.g., $\mathbf{Q}^\top \mathbf{Q}$ is an irreducible matrix). Consider the $p \times (r-1)$ matrix $\mathbf{D} = [\mathrm{diag}(\mathbf{h}_1)]^{-1}[\mathbf{h}_2, \ldots, \mathbf{h}_r]$. Then, there exists a simplex $\mathcal{S}_0^* \subset \mathbb{R}^{r-1}$, which has $r$ vertices $\mathbf{b}_1, \ldots, \mathbf{b}_r$, such that all rows of $\mathbf{D}$ are contained in this simplex. Furthermore, for all anchor states $j$ of a same meta-state, the $j$-th row of $\mathbf{D}$ falls exactly onto one vertex of this simplex.*

Proposition 2 explains the rationale of the vertex hunting step. The vertex hunting step we used was borrowed from [4, 5]; see explanations therein.

Let $\mathbf{b}_1, \ldots, \mathbf{b}_r$ be the vertices of the simplex $\mathcal{S}_0^*$. By vertex hunting, we obtain estimates of these vertices. The next question is: How can we recover $\mathbf{V}$ from the simplex vertices $\mathbf{b}_1, \ldots, \mathbf{b}_r$?

Let $\mathbf{d}_j^\top$ be the $j$-th row of $\mathbf{D}$, for $j \in [p]$. By the nature of a simplex, each point in it can be uniquely expressed as a convex combination of the vertices. This means, for each $j \in [p]$, there exists a weight vector $\mathbf{w}_j$ from the standard simplex such that

$$\mathbf{d}_j = \sum_{k=1}^{r} \mathbf{w}_j(k)\mathbf{b}_k.$$

The next proposition shows that we can recover $\mathbf{V}$ from $\mathbf{w}_1, \ldots, \mathbf{w}_p$.

**Proposition 3** (Relation of simplex and matrix $\mathbf{V}$). *In the setting of Proposition 2, each row of $\mathbf{D}$ is a convex combination of the vertices of $\mathcal{S}_0^*$, i.e., for each $j \in [p]$, there exists $\mathbf{w}_j$ in the standard simplex such that $\mathbf{d}_j = \sum_{k=1}^{r} \mathbf{w}_j(k)\mathbf{b}_k$. Furthermore, consider the matrix $\mathbf{W} \in \mathbb{R}^{p \times r}$, whose $j$-th row equals to $\mathbf{w}_j^\top$. Then, $\mathbf{W}$ and $\mathbf{V}$ are connected by*

$$[\mathrm{diag}(\mathbf{h}_1)][\mathrm{diag}(\boldsymbol{\pi})]^{1/2}\mathbf{W} = \mathbf{V}[\mathrm{diag}(\mathbf{l}_1)],$$

*where $\mathbf{l}_1$ is the first column of $\mathbf{L}$ as defined in* (D .3).

By Proposition 3, each column of the matrix

$$[\mathrm{diag}(\mathbf{h}_1)][\mathrm{diag}(\boldsymbol{\pi})]^{1/2}\mathbf{W}$$

is proportional to the corresponding column of $\mathbf{V}$. Since each column of $\mathbf{V}$ has a unit $\ell^1$-norm, if we normalize each column of the above matrix by its $\ell^1$-norm, we can exactly recover $\mathbf{V}$.

Once we have obtained $\mathbf{V}$, we can immediately recover $\mathbf{U}$ from $(\mathbf{P}, \mathbf{V})$ by the relation:

$$\mathbf{U} = \mathbf{U}(\mathbf{V}^\top\mathbf{V})(\mathbf{V}^\top\mathbf{V})^{-1} = \mathbf{P}\mathbf{V}(\mathbf{V}^\top\mathbf{V})^{-1}.$$

The above gives the following theorem:

**Proposition 4** (Exact recovery of $\mathbf{U}$ and $\mathbf{V}$). *In the setting of Proposition 2, if we apply Algorithm 1 to the matrix $[\mathrm{diag}(\boldsymbol{\pi})]\mathbf{P}[\mathrm{diag}(\boldsymbol{\pi})]^{-1/2}$, it exactly outputs $\mathbf{U}$ and $\mathbf{V}$.*

### D .2 Proof of propositions

Proposition 1 follows from (D .3) and definition of simplicial cone. Proposition 4 is proved in Section 1.1. We now prove Propositions 2-3. Recall that by (D .3), for $k \in [r]$,

$$\mathbf{h}_k = \mathbf{V}^*\mathbf{l}_k,$$

where $\mathbf{l}_k$ is the $k$-th column of $\mathbf{L}$. When all the coordinates of $\mathbf{h}_1$ are strictly positive, the matrix $\mathbf{D}$ is well-defined. Additionally, for an anchor state $j$ of the $k$-th meta state, $\mathbf{h}_1(j) = V_{jk}^*\mathbf{l}_1(k)$, where $V_{jk}^* > 0$. Therefore, $\mathbf{l}_1(k) > 0$ for $k \in [r]$. We define a matrix

$$\mathbf{B} = [\mathrm{diag}(\mathbf{l}_1)]^{-1}[\mathbf{l}_2, \ldots, \mathbf{l}_r].$$

By definition,

$$[\mathbf{1}, \mathbf{D}] = [\mathrm{diag}(\mathbf{h}_1)]^{-1}\mathbf{H}, \tag{D .4}$$

and

$$[\mathbf{1}, \mathbf{B}] = [\mathrm{diag}(\mathbf{l}_1)]^{-1}\mathbf{L}. \tag{D .5}$$

Combining them with (D .3) gives

$$[\mathbf{1}, \mathbf{D}] = [\mathrm{diag}(\mathbf{h}_1)]^{-1}\mathbf{V}^*[\mathrm{diag}(\mathbf{l}_1)][\mathbf{1}, \mathbf{B}]. \tag{D .6}$$

Let
$$\mathbf{W} = [\mathrm{diag}(\mathbf{h}_1)]^{-1}\mathbf{V}^*[\mathrm{diag}(\mathbf{l}_1)].$$
The (D .6) implies
$$\mathbf{1} = \mathbf{W1}, \quad \mathbf{D} = \mathbf{WB}.$$
Since $\mathbf{W}$ is a nonnegative matrix, the first equation implies that each row of $\mathbf{W}$ is from the standard simplex, and the second equation implies that each row of $\mathbf{D}$ is a linear combination of the $r$ rows of $\mathbf{B}$, where the combination coefficients come from the corresponding row of $\mathbf{W}$. This has proved the simplex geometry stated in Proposition 2.

Note that the $j$-th row of $\mathbf{D}$ is located on one vertex of the simplex if and only if the $j$-th row of $\mathbf{W}$ is located on one vertex of the standard simplex. From the way we define $\mathbf{W}$, its $j$-th row equal to
$$\mathbf{w}_j^\top = \frac{1}{\mathbf{h}_1(j)\sqrt{\pi_j}}[V_{j1}\mathbf{l}_1(1), V_{j2}\mathbf{l}_1(2), \ldots, V_{jr}\mathbf{l}_1(r)].$$
Since $\mathbf{h}_1, \mathbf{l}_1$ and $\boldsymbol{\pi}$ are all positive vectors, $\mathbf{w}_j$ is located on one vertex of the standard simplex if and only if exactly one of $V_{j1}, \ldots, V_{jr}$ is nonzero, where the latter is true if and only if $j$ is an anchor state. This has proved Proposition 2.

Furthermore, from the way $\mathbf{W}$ is defined above, using the fact that $\mathbf{V}^* = [\mathrm{diag}(\boldsymbol{\pi})]^{-1/2}\mathbf{V}$, we immediately find that
$$\mathbf{W} = [\mathrm{diag}(\mathbf{h}_1)]^{-1}[\mathrm{diag}(\boldsymbol{\pi})]^{-1/2}\mathbf{V}[\mathrm{diag}(\mathbf{l}_1)],$$
which is equivalent to
$$[\mathrm{diag}(\mathbf{h}_1)][\mathrm{diag}(\boldsymbol{\pi})]^{1/2}\mathbf{W} = \mathbf{V}[\mathrm{diag}(\mathbf{l}_1)].$$
This has proved Proposition 3.

# E    Technical Proofs

*Proof of Lemma A .1.* Since $\boldsymbol{\pi}$ is a stationary distribution, the frequency matrix $\mathbf{F}$ satisfies
$$\sum_{i=1}^{p} F_{ij} = \sum_{i=1}^{p} \pi_i P_{ij} = \pi_j \tag{E .1}$$
for $j = 1, 2, \ldots, p$. Because $\sum_{j=1}^{p} P_{ij} = 1$ for $i = 1, 2, \ldots, p$,
$$\sum_{j=1}^{p} F_{ij} = \sum_{i=1}^{p} \pi_i P_{ij} = \pi_i. \tag{E .2}$$
It follows that
$$\|\mathbf{F}\|_1 = \max_{1\le j\le p} \sum_{i=1}^{p} F_{ij} = \max_{1\le j\le p} \pi_j = \pi_{\max}, \quad \|\mathbf{F}\|_\infty = \max_{1\le i\le p} \sum_{j=1}^{p} F_{ij} = \max_{1\le i\le p} \pi_i = \pi_{\max},$$
which further implies
$$\|\mathbf{F}\|_2 \le \sqrt{\|\mathbf{F}\|_\infty \|\mathbf{F}\|_1} = \pi_{\max} \le C_1 p^{-1}. \tag{E .3}$$
Therefore,
$$\sigma_1 = \|\mathbf{Q}\|_2 = \left\|\mathbf{F}[\mathrm{diag}(\boldsymbol{\pi})]^{-\frac{1}{2}}\right\|_2 \le \pi_{\min}^{-\frac{1}{2}}\|\mathbf{F}\|_2 \le C_1 c_1^{-\frac{1}{2}} p^{-\frac{1}{2}}.$$
As for the smallest singular value $\sigma_r$, by definition,
$$\sigma_r = \min_{\mathbf{x}\in\mathbb{S}^{p-1}}\left\|\mathbf{Q}^\top\mathbf{x}\right\|_2 = \min_{\mathbf{x}\in\mathbb{S}^{p-1}}\left\|[\mathrm{diag}(\boldsymbol{\pi})]^{-\frac{1}{2}}\mathbf{VU}^\top[\mathrm{diag}(\boldsymbol{\pi})]\mathbf{x}\right\|_2.$$
Since
$$\left\|[\mathrm{diag}(\boldsymbol{\pi})]^{-\frac{1}{2}}\mathbf{VU}^\top[\mathrm{diag}(\boldsymbol{\pi})]\mathbf{x}\right\|_2 \ge \sigma_{\min}\left([\mathrm{diag}(\boldsymbol{\pi})]^{-\frac{1}{2}}\mathbf{V}\right)\left\|\mathbf{U}^\top[\mathrm{diag}(\boldsymbol{\pi})]\mathbf{x}\right\|_2$$
and
$$\left\|\mathbf{U}^\top[\mathrm{diag}(\boldsymbol{\pi})]\mathbf{x}\right\|_2 \ge \sigma_{\min}\left(\mathbf{U}^\top[\mathrm{diag}(\boldsymbol{\pi})]\right)\|\mathbf{x}\|_2,$$
we have
$$\sigma_r \ge \sigma_{\min}\left([\mathrm{diag}(\boldsymbol{\pi})]^{-\frac{1}{2}}\mathbf{V}\right)\sigma_{\min}\left(\mathbf{U}^\top[\mathrm{diag}(\boldsymbol{\pi})]\right)$$
$$= \lambda_{\min}^{\frac{1}{2}}\left(\mathbf{V}^\top[\mathrm{diag}(\boldsymbol{\pi})]^{-1}\mathbf{V}\right)\lambda_{\min}^{\frac{1}{2}}\left(\mathbf{U}^\top[\mathrm{diag}(\boldsymbol{\pi})]^2\mathbf{U}\right) \ge c_2 p^{-\frac{1}{2}},$$
where the last inequality holds due to (A .6) ☐

*Proof of Lemma A .2.* We first consider the rows of right singular matrix $\mathbf{H}$. The columns of $[\mathrm{diag}(\boldsymbol{\pi})]^{-\frac{1}{2}}\mathbf{V}$ and $\mathbf{H}$ span the same linear space. Hence, there exists a nonsingular matrix $\mathbf{L} \in \mathbb{R}^{r \times r}$ such that

$$\mathbf{H} = [\mathrm{diag}(\boldsymbol{\pi})]^{-\frac{1}{2}}\mathbf{V}\mathbf{L}. \tag{E .4}$$

We plug (E .4) into $\mathbf{H}^\top\mathbf{H} = \mathbf{I}_r$ and obtain $\mathbf{L}^\top\mathbf{V}^\top[\mathrm{diag}(\boldsymbol{\pi})]^{-1}\mathbf{V}\mathbf{L} = \mathbf{I}_r$. Multiplying $\mathbf{L}$ on the left and $\mathbf{L}^\top$ on the right gives $\mathbf{L}\mathbf{L}^\top\mathbf{V}^\top[\mathrm{diag}(\boldsymbol{\pi})]^{-1}\mathbf{V}\mathbf{L}\mathbf{L}^\top = \mathbf{L}\mathbf{L}^\top$. Because $\mathbf{L}\mathbf{L}^\top$ is non-singular,

$$\mathbf{L}\mathbf{L}^\top = \left(\mathbf{V}^\top[\mathrm{diag}(\boldsymbol{\pi})]^{-1}\mathbf{V}\right)^{-1}.$$

As a result,

$$\|\mathbf{L}\|_2 = \lambda_{\min}^{-1/2}\left(\mathbf{V}^\top[\mathrm{diag}(\boldsymbol{\pi})]^{-1}\mathbf{V}\right) \le c_2^{-\frac{1}{2}}r^{-\frac{1}{2}}, \tag{E .5}$$

which implies that for any $j = 1, 2, \ldots, p$,

$$\begin{aligned}
\left\|\mathbf{e}_j^\top\mathbf{H}\right\|_2 &= \left\|\mathbf{e}_j^\top[\mathrm{diag}(\boldsymbol{\pi})]^{-\frac{1}{2}}\mathbf{V}\mathbf{L}\right\|_2 \le \pi_j^{-\frac{1}{2}}\left\|\mathbf{e}_j^\top\mathbf{V}\right\|_2\|\mathbf{L}\|_2 \\
&\le \pi_j^{-\frac{1}{2}}\left\|\mathbf{e}_j^\top\mathbf{V}\right\|_1\|\mathbf{L}\|_2 \le c_2^{-\frac{1}{2}}\pi_j^{-\frac{1}{2}}r^{-\frac{1}{2}} \cdot \left\|\mathbf{e}_j^\top\mathbf{V}\right\|_1.
\end{aligned} \tag{E .6}$$

It only remains to estimate $\left\|\mathbf{e}_j^\top\mathbf{V}\right\|_1$.

Note that the invariant distribution $\boldsymbol{\pi}$ satisfies $\boldsymbol{\pi}^\top\mathbf{P} = \boldsymbol{\pi}^\top$. For $j = 1, 2, \ldots, p$,

$$\pi_j = \left(\boldsymbol{\pi}^\top\mathbf{P}\right)\mathbf{e}_j = \boldsymbol{\pi}^\top\mathbf{U}\mathbf{V}^\top\mathbf{e}_j = \sum_{k=1}^r (\mathbf{U}^\top\boldsymbol{\pi})_k(\mathbf{V}^\top\mathbf{e}_j)_k \ge \left\|\mathbf{e}_j^\top\mathbf{V}\right\|_1 \cdot \min_{1 \le k \le r}\left(\mathbf{U}^\top\boldsymbol{\pi}\right)_k. \tag{E .7}$$

Under assumption (A .6),

$$\sum_{i=1}^p \pi_i^2 U_{ik}^2 = \mathbf{e}_k^\top\left(\mathbf{U}^\top[\mathrm{diag}(\boldsymbol{\pi})]^2\mathbf{U}\right)\mathbf{e}_k \ge c_2 p^{-1}r^{-1}, \qquad \text{for } k = 1, 2, \ldots, r.$$

It follows that

$$\left(\mathbf{U}^\top\boldsymbol{\pi}\right)_k = \sum_{i=1}^p \pi_i U_{ik} \ge \pi_{\max}^{-1}\sum_{i=1}^p \pi_i^2 U_{ik}^2 \ge C_1^{-1}c_2 r^{-1}. \tag{E .8}$$

Plugging (E .8) into (E .7) yields

$$\left\|\mathbf{e}_j^\top\mathbf{V}\right\|_1 \le C_1 c_2^{-1}\pi_j r. \tag{E .9}$$

We can further derive from (E .6) an upper bound for $\left\|\mathbf{e}_j^\top\mathbf{H}\right\|_2$.

As for the left singular matrix $\mathbf{G}$, we can estimate $\left\|\mathbf{e}_j^\top\mathbf{G}\right\|_2$ in a similar way. Analogous to the definition of $\mathbf{L}$, there exists a nonsingular matrix $\mathbf{R} \in \mathbb{R}^{r \times r}$ such that $\mathbf{G} = [\mathrm{diag}(\boldsymbol{\pi})]\mathbf{U}\mathbf{R}$ and $\|\mathbf{R}\|_2 = \lambda_{\min}^{-1/2}\left(\mathbf{U}^\top[\mathrm{diag}(\boldsymbol{\pi})]^2\mathbf{U}\right) \le c_2^{-\frac{1}{2}}\sqrt{pr}$. It follows that

$$\begin{aligned}
\left\|\mathbf{e}_j^\top\mathbf{G}\right\|_2 &= \left\|\mathbf{e}_j^\top[\mathrm{diag}(\boldsymbol{\pi})]\mathbf{U}\mathbf{R}\right\|_2 = \pi_j\left\|\mathbf{e}_j^\top\mathbf{U}\mathbf{R}\right\|_2 \\
&\le \pi_j\left\|\mathbf{e}_j^\top\mathbf{U}\right\|_2\|\mathbf{R}\|_2 \le \pi_j\left\|\mathbf{e}_j^\top\mathbf{U}\right\|_1\|\mathbf{R}\|_2 = c_2^{-\frac{1}{2}}\pi_j\sqrt{pr},
\end{aligned}$$

where we used $\left\|\mathbf{e}_j^\top\mathbf{U}\right\|_1 = 1$.

$\square$

*Proof of Lemma A .3.* We first show that $\mathbf{l}_1$ is the leading eigen vector of matrix

$$\boldsymbol{\Theta} = \left(\mathbf{U}^\top[\mathrm{diag}(\boldsymbol{\pi})]^2\mathbf{U}\right)\left(\mathbf{V}^\top[\mathrm{diag}(\boldsymbol{\pi})]^{-1}\mathbf{V}\right).$$

Note that by definition, $\mathbf{Q} = [\mathrm{diag}(\boldsymbol{\pi})]\mathbf{U}\mathbf{V}^\top[\mathrm{diag}(\boldsymbol{\pi})]^{-\frac{1}{2}}$, thus $\mathbf{Q}^\top\mathbf{Q}$ and $\boldsymbol{\Theta}$ share the same eigen values. Recall that $\mathbf{h}_1$ is the leading right singular vector of $\mathbf{Q}$,

$$\sigma_1^2\mathbf{h}_1 = \mathbf{Q}^\top\mathbf{Q}\mathbf{h}_1 = [\mathrm{diag}(\boldsymbol{\pi})]^{-\frac{1}{2}}\mathbf{V}\mathbf{U}^\top[\mathrm{diag}(\boldsymbol{\pi})]^2\mathbf{U}\mathbf{V}^\top[\mathrm{diag}(\boldsymbol{\pi})]^{-\frac{1}{2}}\mathbf{h}_1. \tag{E .10}$$

Plugging $\mathbf{h}_1 = [\mathrm{diag}(\boldsymbol{\pi})]^{-\frac{1}{2}}\mathbf{V}\mathbf{l}_1$ into (E .10) and multiplying $\mathbf{V}^\top[\mathrm{diag}(\boldsymbol{\pi})]^{-\frac{1}{2}}$ on the left, we have

$$\sigma_1^2\mathbf{V}^\top[\mathrm{diag}(\boldsymbol{\pi})]^{-1}\mathbf{V}\mathbf{l}_1 = \mathbf{V}^\top[\mathrm{diag}(\boldsymbol{\pi})]^{-1}\mathbf{V}\left(\mathbf{U}^\top[\mathrm{diag}(\boldsymbol{\pi})]^2\mathbf{U}\right)\mathbf{V}^\top[\mathrm{diag}(\boldsymbol{\pi})]^{-1}\mathbf{V}\mathbf{l}_1.$$

It can be reduced to

$$\boldsymbol{\Theta}\mathbf{l}_1 = \big(\mathbf{U}^\top[\mathrm{diag}(\boldsymbol{\pi})]^2\mathbf{U}\big)\big(\mathbf{V}^\top[\mathrm{diag}(\boldsymbol{\pi})]^{-1}\mathbf{V}\big)\mathbf{l}_1 = \sigma_1^2\mathbf{l}_1.$$

The entries of $\boldsymbol{\Theta}$ are lower bounded by

$$\pi_{\min}^2\pi_{\max}^{-1}\min_{k,l}\big[(\mathbf{U}^\top\mathbf{U})(\mathbf{V}^\top\mathbf{V})\big]_{kl} = \pi_{\min}^2\pi_{\max}^{-1}\min_{k,l}\big(\mathbf{U}^\top\mathbf{P}\mathbf{V}\big)_{kl},$$

and upper bounded by

$$\pi_{\max}^2\pi_{\min}^{-1}\max_{k,l}\big[(\mathbf{U}^\top\mathbf{U})(\mathbf{V}^\top\mathbf{V})\big]_{kl} = \pi_{\max}^2\pi_{\min}^{-1}\max_{k,l}\big(\mathbf{U}^\top\mathbf{P}\mathbf{V}\big)_{kl}.$$

Condition (A .5) ensures that $\mathbf{U}^\top\mathbf{P}\mathbf{V}$ is a positive matrix, hence $\boldsymbol{\Theta}$ is also positive. According to Perron-Frobenius Theorem, all components of $\mathbf{l}_1$ are non-zero and have the same sign. Without loss of generality, we assume that the entries of $\mathbf{l}_1$ are all positive. Accoring to Theorem 3.1 in [6],

$$\frac{\max_{1\le k\le r}\mathbf{l}_1(k)}{\min_{1\le k\le r}\mathbf{l}_1(k)} \le \max_{s,t,k}\left\{\frac{\boldsymbol{\Theta}_{sk}}{\boldsymbol{\Theta}_{tk}}\right\} \le \frac{\max_{k,l}\boldsymbol{\Theta}_{kl}}{\min_{k,l}\boldsymbol{\Theta}_{kl}} \le \frac{\pi_{\max}^2\pi_{\min}^{-1}\max_{k,l}\big(\mathbf{U}^\top\mathbf{P}\mathbf{V}\big)_{kl}}{\pi_{\min}^2\pi_{\max}^{-1}\min_{k,l}\big(\mathbf{U}^\top\mathbf{P}\mathbf{V}\big)_{kl}} \le C_1^3c_1^{-3}C_4,$$

(E .11)

where we used assumptions (A .1) and (A .5). Recall that in (E .5), $\|\mathbf{l}_1\|_2 \le \|\mathbf{L}\|_2 \le c_2^{-\frac{1}{2}}r^{-\frac{1}{2}}$, therefore, $\min_{1\le k\le r}\mathbf{l}_1(k) \le r^{-\frac{1}{2}}\|\mathbf{l}_1\|_2 \le c_2^{-\frac{1}{2}}r^{-1}$. (E .11) then implies

$$\max_{1\le k\le r}\mathbf{l}_1(k) \le C_1^3c_1^{-3}C_4\min_{1\le k\le r}\mathbf{l}_1(k) \le \widetilde{C}r^{-1}$$

for some constant $\widetilde{C} > 0$.

Consider $\big\|\mathbf{L}^{-1}\big\|_2$. Since $[\mathrm{diag}(\boldsymbol{\pi})]^{-\frac{1}{2}}\mathbf{V} = \mathbf{H}\mathbf{L}^{-1}$ and $\mathbf{H}$ is orthonormal,

$$\big\|\mathbf{L}^{-1}\big\|_2 = \max_{\mathbf{x}\in\mathbb{S}^{r-1}}\big\|\mathbf{L}^{-1}\mathbf{x}\big\|_2 = \max_{\mathbf{x}\in\mathbb{S}^{r-1}}\big\|\mathbf{H}\mathbf{L}^{-1}\mathbf{x}\big\|_2$$

$$= \max_{\mathbf{x}\in\mathbb{S}^{r-1}}\big\|[\mathrm{diag}(\boldsymbol{\pi})]^{-\frac{1}{2}}\mathbf{V}\mathbf{x}\big\|_2 = \big\|[\mathrm{diag}(\boldsymbol{\pi})]^{-\frac{1}{2}}\mathbf{V}\big\|_2.$$

By (E .9), $0 \le [\mathrm{diag}(\boldsymbol{\pi})]^{-1}\mathbf{V}\mathbf{1}_r \le C_1c_2^{-1}r$, thus

$$\big\|\mathbf{V}^\top[\mathrm{diag}(\boldsymbol{\pi})]^{-1}\mathbf{V}\big\|_1 = \big\|\mathbf{V}^\top[\mathrm{diag}(\boldsymbol{\pi})]^{-1}\mathbf{V}\mathbf{1}_r\big\|_\infty$$

$$\le C_1c_2^{-1}r\big\|\mathbf{V}^\top\mathbf{1}_p\big\|_\infty = C_1c_2^{-1}r\|\mathbf{1}_r\|_\infty = C_1c_2^{-1}r.$$

We have

$$\big\|\mathbf{L}^{-1}\big\|_2 = \big\|[\mathrm{diag}(\boldsymbol{\pi})]^{-\frac{1}{2}}\mathbf{V}\big\|_2 \le \big\|\mathbf{V}^\top[\mathrm{diag}(\boldsymbol{\pi})]^{-1}\mathbf{V}\big\|_1^{\frac{1}{2}} \le C_1^{\frac{1}{2}}c_2^{-\frac{1}{2}}\sqrt{r}.$$

Therefore, $\|\mathbf{l}_1\|_2 \ge \big\|\mathbf{L}^{-1}\big\|_2^{-1} \ge C_1^{-\frac{1}{2}}c_2^{\frac{1}{2}}r^{-\frac{1}{2}}$ and

$$\max_{1\le k\le r}\mathbf{l}_1(k) \ge r^{-\frac{1}{2}}\|\mathbf{l}_1\|_2 \ge C_1^{-\frac{1}{2}}c_2^{\frac{1}{2}}r^{-1}.$$

We can conclude from (E .11) that

$$\min_{1\le k\le r}\mathbf{l}_1(k) \ge C_1^{-3}c_1^3C_4^{-1}\max_{1\le k\le r}\mathbf{l}_1(k) \ge \widetilde{c}r^{-1}$$

for some $\widetilde{c} > 0$. □

*Proof of Lemma A .4.* Recall that by definition

$$\mathbf{h}_1 = [\mathrm{diag}(\boldsymbol{\pi})]^{-\frac{1}{2}}\mathbf{V}\mathbf{l}_1,$$

and Lemma A .3 provides an estimate of the entries in $\mathbf{l}_1$. Therefore,

$$cr^{-1}\cdot\pi_j^{-\frac{1}{2}}\big\|\mathbf{e}_j^\top\mathbf{V}\big\|_1 \le \mathbf{h}_1(j) \le Cr^{-1}\cdot\pi_j^{-\frac{1}{2}}\big\|\mathbf{e}_j^\top\mathbf{V}\big\|_1,$$

where $c, C > 0$ are the constants in Lemma A .3.

Note that in (E.9), there is an upper bound for $\left\|\mathbf{e}_j^\top \mathbf{V}\right\|_1$. Hence,

$$\mathbf{h}_1(j) \leq C\pi_j^{-\frac{1}{2}}r^{-1} \cdot C_1 c_2^{-1}\pi_j r = CC_1 c_2^{-1}\sqrt{\pi_j}. \tag{E.12}$$

We now derive a lower bound for $\left\|\mathbf{e}_j^\top \mathbf{V}\right\|_1$ using assumption (A.2). Because the stationary distribution $\boldsymbol{\pi}$ satisfies $\boldsymbol{\pi}^\top \mathbf{P} = \boldsymbol{\pi}^\top$, for each $j = 1, 2, \ldots, p$ we have

$$\pi_j = \boldsymbol{\pi}^\top \mathbf{P} \mathbf{e}_j = \boldsymbol{\pi}^\top \mathbf{U}\mathbf{V}^\top \mathbf{e}_j = \sum_{k=1}^{r}\left(\mathbf{U}^\top \boldsymbol{\pi}\right)_k \left(\mathbf{e}_j^\top \mathbf{V}\right)_k$$

$$\leq \bar{C}_1 r^{-1}\sum_{k=1}^{r}\left(\mathbf{e}_j^\top \mathbf{V}\right)_k = \bar{C}_1 r^{-1}\left\|\mathbf{e}_j^\top \mathbf{V}\right\|_1.$$

It follows that

$$\mathbf{h}_1(j) \geq c\pi_j^{-\frac{1}{2}}r^{-1}\left\|\mathbf{e}_j^\top \mathbf{V}\right\|_1 \geq c\pi_j^{-\frac{1}{2}}r^{-1} \cdot \bar{C}_1^{-1}\pi_j r = \bar{C}_1^{-1}c\sqrt{\pi_j}. \tag{E.13}$$

As for $\mathbf{g}_1$, by the definition of singular value decomposition,

$$\mathbf{g}_1 = \sigma_1^{-1}\mathbf{Q}\mathbf{h}_1,$$

thus $\mathbf{g}_1$ is nonnegative. For any $j = 1, 2, \ldots, p$,

$$\mathbf{g}_1(j) = \sigma_1^{-1}\mathbf{e}_j^\top \mathbf{Q}\mathbf{h}_1 = \sigma_1^{-1}\mathbf{e}_j^\top \mathbf{F}[\mathrm{diag}(\boldsymbol{\pi})]^{-\frac{1}{2}}\mathbf{h}_1.$$

By (E.12), $[\mathrm{diag}(\boldsymbol{\pi})]^{-\frac{1}{2}}\mathbf{h}_1 \leq CC_1 c_2^{-1}\mathbf{1}_p$, thus

$$\mathbf{g}_1(j) \leq CC_1 c_2^{-1}\sigma_1^{-1}\mathbf{e}_j^\top \mathbf{F}\mathbf{1}_p = CC_1 c_2^{-1}\sigma_1^{-1}\pi_j. \tag{E.14}$$

Here, we used $\mathbf{F}\mathbf{1}_p = \boldsymbol{\pi}$. Since $\left\|[\mathrm{diag}(\boldsymbol{\pi})]^{\frac{1}{2}}\mathbf{1}\right\|_2 = \sqrt{\sum_{i=1}^p \pi_i} = 1$,

$$\sigma_1 = \max_{\mathbf{x}\in\mathbb{R}^p}\left\|\mathbf{Q}\mathbf{x}\right\|_2 \geq \left\|\mathbf{Q}[\mathrm{diag}(\boldsymbol{\pi})]^{\frac{1}{2}}\mathbf{1}_p\right\|_2, = \left\|\mathbf{F}\mathbf{1}_p\right\|_2 = \left\|\boldsymbol{\pi}\right\|_2 \geq p^{-\frac{1}{2}}\left\|\boldsymbol{\pi}\right\|_1 = p^{-\frac{1}{2}}. \tag{E.15}$$

Plugging (E.15) into (E.14), we obtain an upper bound for $\mathbf{g}_1(j)$. $\qquad\square$

# F    Numerical Experiments

## F.1    Explanation of Simulation Settings

We test our new approach on simulated sample transitions. For a $p$-state Markov chain with $r$ meta-states, we first randomly create two matrices $\mathbf{U}, \mathbf{V} \in \mathbb{R}_+^{p\times r}$ such that each meta-state has the same number of anchor states. After assembling a transition matrix $\mathbf{P} = \mathbf{U}\mathbf{V}^\top$, we generate random walk data $\{X_0, X_1, \ldots, X_n\}$. For each data point in the figures, we conduct 5 independent experiments and plot their mean and standard deviation.

In Figure 3 (a), we run experiments with $p = 1000$, $r = 6$ and the number of anchor states equal to $25, 50, 75, 100$ for each meta-state. When $p$ is fixed and $n$ varies, the log-total-variation error in $\widehat{\mathbf{V}}$ scales linearly with $\log(n)$, with a fitted slope $\approx -0.5$. This is consistent with conclusion of Theorem 3 in paper which indicates that the error bound decreases with $n$ at the speed of $n^{-1/2}$. In Figure 3 (b), we carry out experiments with $n/p = 1000$, $r = 6$ and the number of anchor states equal to $\lfloor 0.025p\rfloor, \lfloor 0.050p\rfloor, \lfloor 0.075p\rfloor, \lfloor 0.100p\rfloor$ for each meta-state. When both $(n, p)$ vary while $n/p$ is fixed, the the log-total-variation error in $\widehat{\mathbf{V}}$ remains almost constant, with a fitted slope $\approx 0$. This validates the scaling of $\sqrt{p/n}$ in the error bound of $\widehat{\mathbf{V}}$. In both figures, we observe that having multiple anchor states per each metastate makes the estimation error slightly smaller.

In Figure 3 (c), we consider estimating the transition matrix $\mathbf{P}$ by $\widehat{\mathbf{U}}\widehat{\mathbf{V}}^\top$, and compare it with the the the spectral estimator in [8]. Our method has a slightly better performance. Note that our method not only estimates $\mathbf{P}$ but also estimates $(\mathbf{U}, \mathbf{V})$, while the spectral method cannot estimate $(\mathbf{U}, \mathbf{V})$.

## F.2 More Results on Manhattan Taxi-trip Data

**Distributions of Pick-up and Drop-off Locations.**

Figure F.1: Distributions of pick-up (L) and drop-off (R) location, illustrated as heatmaps.

**Columns of Singular Vectors.**

We conduct SVD to matrix $\widetilde{\mathbf{N}} = \mathbf{N}\left[\mathrm{diag}(\mathbf{N}^{\top}\mathbf{1}_p)\right]^{-\frac{1}{2}}$. Denote the right singular vectors as $\widehat{\mathbf{h}}_1, \widehat{\mathbf{h}}_2, \ldots, \widehat{\mathbf{h}}_r$ and singular values $\widehat{\sigma}_1, \widehat{\sigma}_2, \ldots, \widehat{\sigma}_r$. In the following, we illustrate $\widehat{\mathbf{h}}_1, \widehat{\mathbf{h}}_2, \ldots, \widehat{\mathbf{h}}_r$ with heat maps. It turns out that the figures do not have clear patterns.

$$\widehat{\sigma}_1 = 2.3787 \qquad \widehat{\sigma}_2 = 1.1028 \qquad \widehat{\sigma}_3 = 0.9148$$

$$\widehat{\sigma}_4 = 0.8154 \qquad \widehat{\sigma}_5 = 0.6880 \qquad \widehat{\sigma}_6 = 0.5076$$

$$\widehat{\sigma}_7 = 0.4557 \qquad \widehat{\sigma}_8 = 0.4352. \qquad \widehat{\sigma}_9 = 0.4076$$

$$\widehat{\sigma}_{10} = 0.3818$$

Figure F.2: Singular vectors of $\widetilde{\mathbf{N}}$ illustrated as heat maps.

**Aggregation and Disaggregation Distributions.**

We apply state aggregation learning to $\mathbf{N}$ with the number of meta-states equal to $r = 10$. The columns of estimated $\widehat{\mathbf{U}}$ and $\widehat{\mathbf{V}}$ are illustrated as heat maps. We can tell from the figures that aggregation likelihoods and disaggregation distributions have practical meanings in real life. For example, the heat map of $\widehat{\mathbf{V}}$ has two red points which correspond to New York Penn. Station & New York Ferry Waterway. It reveals the behavior of a certain group of passengers.

$$\widehat{\mathbf{U}}_1 \qquad \widehat{\mathbf{V}}_1$$

$$\left(\widehat{\boldsymbol{\pi}}^{\top}\widehat{\mathbf{U}}\right)_1 = 0.0770$$

$$\left(\widehat{\boldsymbol{\pi}}^{\top}\widehat{\mathbf{U}}\right)_2 = 0.1133$$

$\widehat{\mathbf{U}}_2$     $\widehat{\mathbf{V}}_2$

$$\left(\widehat{\boldsymbol{\pi}}^{\top}\widehat{\mathbf{U}}\right)_3 = 0.0503$$

$\widehat{\mathbf{U}}_3$     $\widehat{\mathbf{V}}_3$

$$\left(\widehat{\boldsymbol{\pi}}^{\top}\widehat{\mathbf{U}}\right)_4 = 0.1022$$

$\widehat{\mathbf{U}}_4$     $\widehat{\mathbf{V}}_4$

Right: The two red points correspond to New York Penn. Station & New York Ferry Waterway.

$\widehat{\mathbf{U}}_5$                    $\widehat{\mathbf{V}}_5$

$$\left(\widehat{\boldsymbol{\pi}}^{\top}\widehat{\mathbf{U}}\right)_5 = 0.1135$$

$\widehat{\mathbf{U}}_6$                    $\widehat{\mathbf{V}}_6$

$$\left(\widehat{\boldsymbol{\pi}}^{\top}\widehat{\mathbf{U}}\right)_6 = 0.0757$$

$\widehat{\mathbf{U}}_7$                    $\widehat{\mathbf{V}}_7$

$$\left(\widehat{\boldsymbol{\pi}}^{\top}\widehat{\mathbf{U}}\right)_7 = 0.1288$$

$$\left(\widehat{\boldsymbol{\pi}}^{\top}\widehat{\mathbf{U}}\right)_8 = 0.0899$$

$$\left(\widehat{\boldsymbol{\pi}}^{\top}\widehat{\mathbf{U}}\right)_9 = 0.1385$$

$$\left(\widehat{\boldsymbol{\pi}}^{\top}\widehat{\mathbf{U}}\right)_{10} = 0.1106$$

Figure F.3: Columns of $\widehat{\mathbf{U}}$ and $\widehat{\mathbf{V}}$ illustrated as heat maps.

**Anchor Regions and Partitions for Different $r$.**

Figure F.4: Partition of New York City by rounding the estimated disaggregation distributions to the closest vertices in the low-dimensional simplex.