[Reviews · NeurIPS 2019]

Reviewer 1



This paper studies the problem of learning soft state aggregation of a Markov model, where there are r hidden meta states, each corresponds to a distribution over the observed state of the Markov model. Under the anchor state assumption, the authors propose an algorithm that provably learns the state aggregation model from the Markov chain’s trajectory. They evaluated their algorithm on a Manhattan taxi-trip dataset which yields interesting discoveries. There has been lots of work on estimating the low rank transition matrix itself and on matrix factorization in the topic modelling setting, and this work seems to be connecting the two problems. The paper is presented well and easy to follow. I have the following questions regarding the novelty and impact of this paper. Questions/concerns: 1. As mentioned in line 97, there are lots of work on estimating the transition matrix of a Markov model. Will similar result as in this paper hold if one first applies such an algorithm, e.g [28], and then simply runs the algorithm for topic modeling/non-negative matrix factorization, e.g. [4]? If no theoretical guarantee can be show, maybe this can serve as a benchmark in the experiment? 2. The authors emphasize in line 273 that the major difference between this word and topic modeling is that the observations from a Markov chain trajectory are correlated while these are independent in the topic modelling setting. However, one could instead sample from the trajectory once in a while (only when the chain mixes). It seems to me that the problem reduces to a topic modelling problem while the sample size only shrink by a factor of tau_*. Can the authors comment on that? 3. Certain minimax lower bound is known for the factorization problem in the topic modelling setting [22], is it possible to obtain similar result here? I appreciate the author's clarification. My questions are addressed. I've updated my score to 6.

Reviewer 2



Overall the paper has strong and interesting results to share. However, I would be happier if the authors improved the exposition of the material.

Reviewer 3



The authors could do a better job pointing out the novelty of their techniques -- i.e., where these depart from standard spectral methods.

[Author Response · NeurIPS 2019]

We thank three reviewers for the constructive comments. We are especially happy that they think the paper contains
strong and interesting results. In what follows we make a few clarifications and itemized responses.

**Novelty of our work:**

• We are the first to bring the perspective of NMF/topic modeling to Markov state aggregation. Even though the idea
is not hard to understand, *no one has investigated it*. Prior to our work, there are no existing methods that can directly
estimate the aggregation/disaggregation distributions with theoretical guarantees. Our work opens the door of bridging
two areas. We hope our work would inspire future work and more efficient methods may be developed.

• Making this idea work, both in theory and in applications, requires substantial efforts and nontrivial analysis. The
devil is in the detail. In theory, the entry-wise eigenvector analysis, particularly needed for state aggregation learning, is
very challenging. This result per se is significant and provides a technical tool that may be useful for the analysis of a
broader class of spectral/NMF methods. In application, we not only obtain encouraging results on Manhattan taxi data
but also demonstrate the method is useful for accelerating policy learning.

**Response to Reviewer #1**

• *"Will similar results hold for a straightforward combination of existing estimators of transition matrix (e.g., [28])*
*and existing topic modeling methods (e.g. [4])?"*
**RE:** Good point. In fact, straightforward application of [4] would yield a slower rate of convergence. The reason is that
"translating" our problem to a topic model would result in a special case where the dictionary size is equal to the number
of documents. Unfortunately, in this case, [4] has a sub-optimal rate of convergence (see [22]). Combining [4] with the
estimator in [28] is a good idea, but whether it resolves the sub-optimality issue remains unclear. Additionally, our
method has a practical advantage: It operates on the projected data by PCA and is computationally fast. In contrast, a
combination of [28] and [4] would require handling data in high dimensions, which is computationally more intensive.
Besides, this is only a potential proposal, not an existing method. We agree that it is very interesting to explore all kinds
of possibilities in the context of our framework, but it is beyond the scope of this paper.

• *"Why not consider sampling from the trajectory and reduce it to a standard topic model?"*
**RE:** Downsampling the trajectory was exactly what we did in the previous version of this paper. However, this
simplified approach received many criticisms. By resampling from the trajectory, we lose the sample size by a constant
factor. In practice, the sample size is often limited compared to the dimension, and discarding even a fraction of samples
can significantly deteriorate the accuracy. Additionally, the downsampling approach requires knowledge of the mixing
time or at least its lower bound, which is often not known and becomes an additional tuning parameter. Resampling
from the trajectory is only a way to avoid technical difficulty of theorem proving. In practice, people would almost
always use all the data without downsampling. We prefer not to have such a gap between theory and application.

• *"It there any minimax lower bound?"*
**RE:** As mentioned in Lines 264-266, there exists a lower bound for $r = 1$. To obtain a lower bound for $r > 1$ is very
interesting, but it is beyond the scope of this paper. This paper aims to provide an algorithm with provable guarantees.

• *"Better explanation of the connection to related work."*
**RE:** Thanks for the nice suggestion. In the submission, we summarized the connection to related works in state
aggregation, spectral methods, estimating transition matrix, learning mixtures of discrete distributions, topic modeling,
and NMF. See Section 1 and the end of Section 4. We will follow your suggestion to re-arrange and expand them.

**Response to Reviewer #3**

• *"Improvement on writing, such as to expand the section of "connection to literatures", to shorten "our contributions",*
*to re-arrange Sections 3 and 4, and to mention some proof ideas)"*
**RE.** Thank you for these great suggestions! We will follow them to improve the writing.

• *"More discussions on Assumptions (a)-(e)."*
**RE.** Thank you. We kept the discussions short due to space limit. We used to have extensive discussions on these
assumptions in a previous version of the paper, and we will add them back. We are glad that you see merits in our paper
and we will improve the writing and organization as you suggested.

**Response to Reviewer #4**

• *"Difference from standard spectral methods."*
**RE:** Thank you for seeing the merit in our paper. The state aggregation model has richer structure than just spectral
decomposition (eg., polytope structure, anchor states and nonnegativity). In Lines 153-162, we show that each left/right
singular vector is a linear combination of multiple disaggregation/aggregation distributions, however they cannot be
used to immediately identify the disaggregation/aggregation distributions. This is why we need to use anchor states to
help us identify the simplex structure of the state space. The key idea of our method is to leverage the anchor structure
and "combine" multiple singular vectors to get a valid estimate of an individual aggregation/disaggregation distribution.
As a result, our method needs to perform several non-trivial steps after performing singular value decomposition.
Experiments with Manhattan taxi data also clearly shows the comparison between our method and standard spectral
method.

[Meta-Review · NeurIPS 2019]

The referees are unanimous in recommending acceptance, two of them giving high marks. I have read the paper and judge it to be a worthy contribution.